# Optimal and Adaptive Monteiro-Svaiter Acceleration

Yair Carmon[*]    Danielle Hausler[*]    Arun Jambulapati[†]    Yujia Jin[†]    Aaron Sidford[†]

## Abstract

We develop a variant of the Monteiro-Svaiter (MS) acceleration framework that removes the need to solve an expensive implicit equation at every iteration. Consequently, for any $p \geq 2$ we improve the complexity of convex optimization with Lipschitz $p$th derivative by a logarithmic factor, matching a lower bound. We also introduce an MS subproblem solver that requires no knowledge of problem parameters, and implement it as either a second- or first-order method via exact linear system solution or MinRes, respectively. On logistic regression our method outperforms previous second-order acceleration schemes, but under-performs Newton's method; simply iterating our first-order adaptive subproblem solver performs comparably to L-BFGS.

## 1 Introduction

We consider the problem of minimizing a convex function $f : \mathcal{X} \to \mathbb{R}$ over closed convex set $\mathcal{X} \subseteq \mathbb{R}^d$, given access to an oracle $\mathcal{O} : \mathcal{X} \to \mathcal{X}$ that minimizes a local model of $f$ around a given query point. A key motivating example of such an oracle is the cubic-regularized Newton step

$$\mathcal{O}_{\mathsf{cr}}(y) = \operatorname*{argmin}_{x \in \mathcal{X}} \left\{ f(y) + \nabla f(y)^\top (x - y) + \frac{1}{2}(x-y)^\top \nabla^2 f(y)(x-y) + \frac{M}{6}\|x - y\|^3 \right\}, \quad (1)$$

i.e., minimizing the second-order Taylor approximation of $f$ around $y$ plus a cubic regularization term. However, our results apply to additional oracles including a simple gradient step, regularized higher-order Taylor expansions [5, 19, 7, 23, 8, 35, 21, 41, 36, 26], ball-constrained optimization [12], and new adaptive oracles that we develop.

Seminal work by Monteiro and Svaiter [32] (MS) shows how to accelerate the basic oracle iteration $x_{t+1} = \mathcal{O}(x_t)$. Their algorithm is based on the fact that many oracles, including $\mathcal{O}_{\mathsf{cr}}$, implicitly approximate proximal points. That is, for every $y$ and $x = \mathcal{O}(y)$, there exists $\lambda_{x,y} > 0$ such that $x \approx \operatorname{argmin}_{x' \in \mathcal{X}} \{ f(x') + \frac{1}{2}\lambda_{x,y}\|x' - y\|^2 \}$, with the approximation error controlled by a specific condition they define. MS prove that, under this condition, the accelerated proximal point method [22, 40] (with dynamic regularization parameter) maintains its rate of convergence. Applying their framework to $\mathcal{O}_{\mathsf{cr}}$ and assuming $\nabla^2 f$ is Lipschitz, they achieve error bounds that decay as $O(t^{-7/2}\log t)$ after $t$ oracle calls, improving the $O(t^{-2})$ rate of the basic $\mathcal{O}_{\mathsf{cr}}$ iteration [37] and the $O(t^{-3})$ rate of an earlier accelerated method [33]. Subsequent works apply variations of the MS framework to different oracles, obtaining improved theoretical guarantees for functions with continuous higher-order derivatives [19, 7, 23, 41, 2], parallel optimization [6], logistic and $\ell_\infty$ regression [8, 12], minimizing functions with Hölder continuous higher derivatives [41], and distributionally-robust optimization [13, 11].

However, all of these algorithms based on the MS framework share a common drawback: the iterate $y_t$ used to produce $x_{t+1} = \mathcal{O}(y_t)$ depends on the proximal parameter $\lambda_{t+1} = \lambda_{x_{t+1}, y_t}$, which itself depends on both $x_{t+1}$ and $y_t$. This circular dependence necessitates solving an implicit equation for

---

[*]Tel Aviv University, ycarmon@tauex.tau.ac.il, hausler@mail.tau.ac.il

[†]Stanford University, {jmblpati,yujiajin,sidford}@stanford.edu

36th Conference on Neural Information Processing Systems (NeurIPS 2022).

$\lambda_{t+1}$; MS (and many subsequent results based upon it) propose bisection procedures for doing so using a number of oracle calls logarithmic in the problem parameters. From a theoretical perspective, the additional bisection complexity introduces a logarithmic gap between the upper bounds due to MS-based algorithms and the best known lower bounds [3, 21] in a number of settings.

From a practical perspective, the use of bisection in the MS framework is undesirable as it potentially discards the optimization progress made by oracle calls during each bisection. In his textbook, Nesterov [34, §4.3.3] argues that the logarithmic cost of bisection likely renders the MS scheme for accelerating $\mathcal{O}_{\mathsf{cr}}$ inferior in practice to algorithms whose error decays at the asymptotically worse rate $O(t^{-3})$ but do not require bisection; he notes that removing the bisection from the MS algorithm is an "open and challenging question in Optimization Theory." Carmon et al. [13] also point out bisection as one of the main hurdles in making their theoretical scheme practical, while Song et al. [41] note this limitation and propose a heuristic alternative to bisection. (See Appendix A for extended discussion of related work, including a concurrent and independent result by Kovalev and Gasnikov [27].)

## 1.1   Our contributions

We settle this open question, providing a variant of MS acceleration that does not require bisection (Section 2). When combined with certain existing MS oracles (Section 3.1), our algorithm obtains complexity bounds that are optimal up to constant factors, improving over prior art by a logarithmic factor (see Table 1). In addition, our algorithm has no parameters sensitive to tuning.

We then go a step further and (in Section 3.2) develop an adaptive alternative to $\mathcal{O}_{\mathsf{cr}}$ (Equation (1)). Our oracle does not require tuning the parameter $M$, which in theory should be proportional to the (difficult to estimate) Lipschitz constant of $\nabla^2 f$. Using our oracle, we obtain the optimal Hessian evaluation complexity $O(t^{-(4+3\nu)/2})$ for functions with order-$\nu$ Hölder Hessian (Lipschitz Hessian is the $\nu = 1$ special case), without requiring any knowledge of the Hölder constant and order $\nu$. Our oracle is also efficient: while existing complexity bounds for computing $\mathcal{O}_{\mathsf{cr}}$ require a logarithmic number of linear system solutions per call, our oracle requires a double-logarithmic number. Moreover, when used with our acceleration method, the number of linear system solves per iteration is essentially constant.

We also provide a first-order implementation of our adaptive oracle (Section 3.3). It approximately solves linear systems via first-order operations (Hessian-vector products) using MinRes/Conjugate Residuals [42, 18] with a simple, adaptive, stopping criterion lifted directly from our analysis. Our oracle attains the optimal first-order evaluation complexity for smooth functions up to an *additive* logarithmic term, without knowledge of the gradient Lipschitz constant or any parameter tuning. Moreover, it maintains an optimal outer iteration complexity for Hölder Hessian of any order.

Finally, we report empirical results (Section 4).[3] On logistic regression problems, combining our optimal acceleration scheme with our adaptive oracle outperforms previously proposed accelerated second-order methods. However, we also show that (while somewhat helpful for $\mathcal{O}_{\mathsf{cr}}$ with a conservative choice of $H$), adding momentum to well-tuned or adaptive second-order methods is *harmful* in logistic regression: simply iterating our oracle—or, better yet, applying Newton's method—dramatically outperforms all "accelerated" algorithms. This important fact seems to have gone unobserved in the literature on accelerated second-order methods, despite logistic regression appearing in many related experiments [41, 16, 30, 24]. Simply iterating our adaptive oracle outperforms the classical accelerated gradient descent, and performs comparably to L-BFGS.

## 1.2   Limitations and outlook

While our algorithms resolve an enduring theoretical open problem in convex optimization, and are free of sensitive parameters that typically hinder theoretically-optimal methods, practical performance remains a limitation. On logistic regression, Newton's method is remarkably fast, and our acceleration scheme does not seem to help our adaptive oracle. We do not fully understand why this is so, but we suspect that it has to do with additional structure in logistic regression, which Newton's method can automatically exploit but momentum cannot. We believe that future research should identify the structure that makes Newton's method so efficient, and modifying momentum schemes to leverage it.

---

[3]The code for our experiments is available at `https://github.com/danielle-hausler/ms-optimal`.

Scalability is another important limitation. While our first-order oracle significantly improves scalability over the second-order oracle from which it is built, it still relies on exact gradient and Hessian-vector products. Therefore, it will have difficulty scaling up to very large datasets. Nevertheless, we hope that further scalability improvements may be possible by building an oracle that utilizes cheap stochastic gradient estimates instead of exact gradients, bringing with it the exciting prospect of a new and powerful adaptive stochastic gradient method. The alternative, probabilistic approximation condition we propose in Appendix C might be helpful in this regard.

## 2 Removing bisection from the Monteiro-Svaiter framework

**Algorithm 1:** Optimal MS Acceleration

**Input:** Initial $x_0$, function $f$, oracle $\mathcal{O}$
**Parameters:** Initial $\lambda'_0$, multiplicative adjustment factor $\alpha > 1$

1 Set $v_0 = x_0$, $A_0 = 0$
2 $\tilde{x}_1, \lambda_1 = \mathcal{O}(x_0; \lambda'_0)$ , $\lambda'_1 = \lambda_1$
3 **for** $t = 0, 1, \ldots,$ **do**
4    $a'_{t+1} = \frac{1}{2\lambda'_{t+1}}\left(1 + \sqrt{1 + 4\lambda'_{t+1}A_t}\right)$
5    $A'_{t+1} = A_t + a'_{t+1}$
6    $y_t = \frac{A_t}{A'_{t+1}}x_t + \frac{a'_{t+1}}{A'_{t+1}}v_t$
7    **if** $t > 0$ **then** $\tilde{x}_{t+1}, \lambda_{t+1} = \mathcal{O}(y_t; \lambda'_{t+1})$
8    **if** $\lambda_{t+1} \leq \lambda'_{t+1}$ **then**
9      $a_{t+1} = a'_{t+1}$, $A_{t+1} = A'_{t+1}$
10      $x_{t+1} = \tilde{x}_{t+1}$
11      $\lambda'_{t+2} = \frac{1}{\alpha}\lambda'_{t+1}$
12    **else**
13      $\gamma_{t+1} = \frac{\lambda'_{t+1}}{\lambda_{t+1}}$
14      $a_{t+1} = \gamma_{t+1}a'_{t+1}$, $A_{t+1} = A_t + a_{t+1}$
15      $x_{t+1} = \frac{(1-\gamma_{t+1})A_t}{A_{t+1}}x_t + \frac{\gamma_{t+1}A'_{t+1}}{A_{t+1}}\tilde{x}_{t+1}$
16      $\lambda'_{t+2} = \alpha\lambda'_{t+1}$
17    $v_{t+1} = v_t - a_{t+1}\nabla f(\tilde{x}_{t+1})$

**Algorithm 0:** MS Acceleration

**Input:** Initial $x_0$, function $f$, oracle $\mathcal{O}$
**Parameters:** Bisection limits $\lambda^\ell, \lambda^h$, and tolerance $\rho > 1$

1 Set $v_0 = x_0$, $A_0 = 0$
2 **for** $t = 0, 1, \ldots,$ **do**
3    $\lambda^\ell_{t+1}, \lambda^h_{t+1} = \lambda^\ell, \lambda^h$
4    $\lambda'_{t+1} = \frac{\lambda^\ell_{t+1} + \lambda^h_{t+1}}{2}$
5    $a'_{t+1} = \frac{1}{2\lambda'_{t+1}}\left(1 + \sqrt{1 + 4\lambda'_{t+1}A_t}\right)$
6    $A'_{t+1} = A_t + a'_{t+1}$
7    $y_t = \frac{A_t}{A'_{t+1}}x_t + \frac{a'_{t+1}}{A'_{t+1}}v_t$
8    $\tilde{x}_{t+1}, \lambda_{t+1} = \mathcal{O}(y_t; \lambda'_{t+1})$
9    **if** $\lambda_{t+1} \in \left[\frac{1}{\rho}\lambda'_{t+1}, \lambda'_{t+1}\right]$ **then**
10      $a_{t+1} = a'_{t+1}$, $A_{t+1} = A'_{t+1}$
11      $x_{t+1} = \tilde{x}_{t+1}$
12    **else if** $\lambda_{t+1} < \frac{1}{\rho}\lambda'_{t+1}$ **then**
13      $\lambda^h_{t+1} = \lambda'_{t+1}$
14      Go to line 4
15    **else**
16      $\lambda^l_{t+1} = \lambda'_{t+1}$
17      Go to line 4
18    $v_{t+1} = v_t - a_{t+1}\nabla f(\tilde{x}_{t+1})$

In this section we present our acceleration algorithm (Algorithm 1) which removes bisection from the MS method (shown in stylized form as Algorithm 0) and thereby attains optimal rates of convergence. For simplicity, in this section and the next we focus on unconstrained optimization ($\mathcal{X} = \mathbb{R}^d$) and assume that $f$ is continuously differentiable, so that $\nabla f$ exists. In Appendix C we extend our framework to general closed and convex domains and non-differentiable convex objectives.

The key object in both the original MS algorithm and our new variant is an oracle $\mathcal{O}$ that approximately minimizes a local model of $f$ at a query point $y$. In particular, $\mathcal{O}$ satisfies the following approximation error bound, adapted from Monteiro and Svaiter [32, eq. (3.3)] ($\lambda$ in [32] is $1/\lambda$ in our notation).

**Definition 1** (MS oracle)**.** *An oracle* $\mathcal{O} : \mathbb{R}^d \times \mathbb{R}_+ \to \mathbb{R}^d \times \mathbb{R}_+$ *is a* $\sigma$-MS oracle *for function* $f : \mathbb{R}^d \to \mathbb{R}$ *if for every* $y \in \mathbb{R}^d$ *and* $\lambda' > 0$*, the points* $(x, \lambda) = \mathcal{O}(y; \lambda')$ *satisfy*

$$\left\|x - \left(y - \tfrac{1}{\lambda}\nabla f(x)\right)\right\| \leq \sigma\|x - y\|. \tag{2}$$

Definition 1 endows the oracle with an additional output $\lambda$ and an additional input $\lambda'$. The value of $\lambda$ has the following simple interpretation: any point $x$ satisfying (2) approximately minimizes $F(x') = f(x') + \frac{\lambda}{2}\|x' - y\|^2$ in the sense that $\|\nabla F(x)\| \leq \lambda\sigma\|x - y\|$. In particular, computing an exact proximal point $x_\lambda = \text{argmin}_{x'} F(x')$ and outputting $(x, \lambda)$ implements a 0-MS oracle. The input $\lambda'$ is optional: oracle implementations in prior work do not require it, but our new adaptive

oracles (described in the next section) use it for improved efficiency. In Appendix C we provide a slightly more general approximation condition for MS oracles that handles non-smooth objectives and bounded domains, as well as a different, stochastic condition similar to that of [4, 11].

Let us discuss the key differences between our algorithm (Algorithm 1) and the stylized MS algorithm (Algorithm 0). At every iteration, Algorithm 0 searches for a value $\lambda'_{t+1}$ such that $\tilde{x}_{t+1}, \lambda_{t+1} = \mathcal{O}(y_t; \lambda'_{t+1})$ satisfies $\lambda_{t+1} \approx \lambda'_{t+1}$ (note that $y_t$ depends on $\lambda'_{t+1}$). This is done via a bisection procedure iteratively shrinking an interval that contains a successful choice of $\lambda'_{t+1}$.[4] This bisection process is inefficient in the sense that every time we reach lines 14 and 17 (highlighted in red) all of the optimization progress made by the last oracle call is discarded.

In contrast, even though our algorithm queries $\mathcal{O}$ in the same way (with $y_t$ computed based on a guess $\lambda'_{t+1}$), it makes use of the oracle output even if $\lambda_{t+1}$ is very far from $\lambda'_{t+1}$, thus never discarding progress made by the oracle. Instead of performing a bisection, we compare $\lambda'_{t+1}$ and $\lambda_{t+1}$ to guide our next guess $\lambda'_{t+2}$. When $\lambda'$ overshoots $\lambda$, we decrease it by a factor $\alpha$ (line 11, highlighted in green) and set $x_t$ and $A_t$ as in Algorithm 0. When it undershoots, we multiply it by $\alpha$ (line 16). In this case, we perform an additional key algorithmic modification which we call the *momentum damping mechanism*: we scale down the growth of the parameter $A_{t+1}$ and replace the next iterate with a convex combination of $x_t$ and $\tilde{x}_{t+1}$ to ensure that our overly optimistic guess for $\lambda_{t+1}$ does not destabilize the algorithm.[5] In Appendix E.6 we demonstrate empirically that this mechanism is important for stabilizing Algorithm 1.

Different MS oracles attain different rates of convergence when accelerated via the MS framework. In the following definition, we distill a key property that determines this rate.

**Definition 2** (Movement bound). *For $s \geq 1$, $c, \lambda > 0$, and $x, y \in \mathbb{R}^d$ we say that $(x, y, \lambda)$ satisfy a $(s, c)$-movement bound if*

$$\|x - y\| \geq \begin{cases} (\lambda/c^s)^{1/(s-1)} & s < \infty \\ 1/c & s = \infty, \end{cases} \tag{3}$$

*where a $(1, c)$-movement bound simply means that $\lambda \leq c$.*

In the next section, we will show how to build MS oracles that, given query $y$, output $(x, \lambda)$ such that $(x, y, \lambda)$ always satisfy a $(s, c)$-movement bound, for certain $s$ and $c$ depending on the oracle type and function structure (e.g., level of smoothness). For example, when $f$ has $H$-Lipschitz Hessian, the cubic-regularized Newton step with $M = 2H$ is a $\frac{1}{2}$-MS oracle that guarantees a $(2, \sqrt{H})$-movement bound. With the necessary definitions in hand, we are ready to state our main result: the iteration (and MS oracle query) complexity of Algorithm 1.

**Theorem 1.** *Let $f : \mathbb{R}^d \to \mathbb{R}$ be convex and differentiable, and consider Algorithm 1 with parameters $\alpha > 1$, $\lambda' > 0$, and a $\sigma$-MS oracle (Definition 1) for $f$ with $\sigma \in [0, 0.99)$. Let $s \geq 1$ and $c > 0$, and suppose that for all $t$ such that $\lambda_t > \lambda'_t$ or $t = 1$, the iterates $(\tilde{x}_t, y_{t-1}, \lambda_t)$ satisfy a $(s, c)$-movement bound (Definition 2). There exist $C_{\alpha,s} = O\left(\frac{s}{\min\{s, \ln \alpha\}} \alpha^{\frac{s+1}{3s+1}}\right)$ and $K_\alpha = O\left(\frac{1}{\ln \alpha} \alpha^{1/3}\right)$ such that[6] for any $x_\star \in \mathbb{R}^d$ and $\epsilon > 0$, we have $f(x_T) - f(x_\star) \leq \epsilon$ when*

$$T \geq \begin{cases} C_{\alpha,s} \cdot \left(\frac{c^s \|x_0 - x_\star\|^{s+1}}{\epsilon}\right)^{\frac{2}{3s+1}} & s < \infty \\[2mm] K_\alpha \cdot (c\|x_0 - x_\star\|)^{\frac{2}{3}} \log \frac{\lambda_1 \|x_0 - x_\star\|^2}{\epsilon} & s = \infty. \end{cases}$$

**Proof sketch.** The remainder of this section is an overview of the proof of Theorem 1, which we provide in full in Appendix B. To simplify this proof sketch, we treat $\alpha$, $c$, and $1/(1 - \sigma)$ as $O(1)$,

---

[4]Algorithm 0 simplifies the bisection routine of Monteiro and Svaiter [32] and implicitly assumes that an initial interval $[\lambda^\ell, \lambda^h]$ always contains a valid solution. One can guarantee such an interval exists by selecting very small $\lambda^\ell$ and very large $\lambda^h$. Alternatively, one may construct a valid initial interval via a bracketing procedure, as we do in the empirical comparison. Either way, the cost is logarithmic in problem parameters.

[5]It is also possible to set $x_{t+1} = \arg\min_{x \in \{\tilde{x}_{t+1}, x_t\}} f(x)$ instead of the convex combination in line 15 and maintain our theoretical guarantees.

[6]For a fixed $s \geq 1$, the value of $\alpha$ minimizing our complexity bound is $\alpha^\star = e^{\frac{3s+1}{s+1}}$. In practice, performance is not sensitive to the choice of $\alpha$ (see Appendix E.3).

and focus on $s < \infty$. To highlight the novel aspects of the proof, let us first briefly recall the analysis of Algorithm 0 [32, 19, 7, 23, 12]. For every $t \leq T$ let

$$E_t := f(x_t) - f(x_\star) \ , \quad D_t := \frac{1}{2}\|v_t - x_\star\|^2 \ \text{ and } \ M_t = \frac{1}{2}\|\tilde{x}_t - y_{t-1}\|^2.$$

The key facts about the standard MS iterations are

$$E_T \leq \frac{D_0}{A_T} \ , \quad \sum_{t \in [T]} \lambda_t A_t M_t \leq O(D_0) \ \text{ and } \ \sqrt{A_T} \geq \Omega(1) \sum_{t \in [T]} \frac{1}{\sqrt{\lambda'_t}}. \tag{4}$$

The first fact implies that the optimality gap at iteration $T$ is inversely proportional to $A_T$, while the latter two facts imply that $A_T$ grows rapidly. More specifically, substituting the movement bound $M_t \geq \Omega\big((\lambda_t)^{2/(s-1)}\big)$ and $\lambda'_t \geq \Omega(\lambda_t)$ (thanks to the bisection) yields $\sum_{t \in [T]} \lambda'^{\frac{s+1}{s-1}}_t A_t = O(D_0)$. Combining this with the third fact in (4) and using the reverse Hölder inequality allows one to conclude that, for $k = \frac{s+1}{3s+1}$ and $k' = \frac{s-1}{3s+1}$, we have $A_T^k \geq \Omega(D_0^{-k'}) \sum_{t \in [T]} A_i^{k'}$, which, upon further algebraic manipulation, yields $A_T \geq \Omega(T^{(3s+1)/2} D_0^{-(s-1)/2})$. Plugging this back to to the first fact in (4) gives the claimed convergence rate.

Having described the standard MS analysis, we move on to our algorithm. Our *first challenge* is re-establishing the facts (4). The difficult case is $\lambda_t > \lambda'_t$, where the standard cancellation that occurs in the MS analysis may fail. This is where the momentum damping mechanism (lines 14 and 15 of our algorithm) comes into play, allowing us to show that (See Proposition 1 in the appendix)

$$E_T \leq \frac{D_0}{A_T} \ , \quad \sum_{t \in \mathcal{S}_{\overline{T}}^\geq \cup \{1\}} \lambda'_t A_t M_t \leq O(D_0) \ \text{ and } \ \sqrt{A_T} \geq \Omega(1) \sum_{t \in \mathcal{S}_{\overline{T}}^\leq} \frac{1}{\sqrt{\lambda'_t}}, \tag{5}$$

where $\mathcal{S}_{\overline{T}}^\geq := \{t \in [T] \mid \lambda_t \geq \lambda'_t\}$ and $\mathcal{S}_{\overline{T}}^>, \mathcal{S}_{\overline{T}}^\leq, \mathcal{S}_{\overline{T}}^<$ and $\mathcal{S}_{\overline{T}}^=$ are analogously defined.

Comparing (4) and (5), the price of removing the bisection becomes evident: at each iteration (except the first) only one of the terms forcing the growth of $A_t$ receives a contribution. The *second challenge* of our proof is establishing a lower bound on $\sqrt{A_T}$ in terms of the $1/\sqrt{\lambda'_t}$ values for $t \in \mathcal{S}_{\overline{T}}^> \cup \{1\}$, where the movement bound holds for $M_t$. This is where the multiplicative $\lambda'$ update rule (lines 11 and 16 of the algorithm) comes into play: it allows us to "credit" the contribution of every "down iterate" (in $\mathcal{S}_{\overline{T}}^\leq$) to an adjacent "up" iterate ($\mathcal{S}_{\overline{T}}^> \cup \{1\}$) and furthermore argue that the contribution gets an exponential bonus based on the distance between the two. Consequently, we are able to identify a set $\mathcal{Q}_T \subseteq \mathcal{S}_{\overline{T}}^> \cup \{1\}$ of iterates, and a sequence $\{r_t\}$ such that (see Lemma 1) $\sqrt{A_T} \geq \Omega(1) \sum_{t \in \mathcal{Q}_T} \sqrt{\frac{\alpha^{r_t-1}}{\lambda'_t}}$ and $\sum_{t \in [T]} r_t = \frac{T-1}{2}$.

Repeating the reverse Hölder argument of prior work, we obtain the recursive bound

$$A_T^k \geq \Omega(D_0^{-k'}) \sum_{t \in \mathcal{Q}_T} A_t^{k'} \alpha^{kr_t} \geq \Omega(D_0^{-k'}) \sum_{t \in \mathcal{Q}_T} A_t^{k'} r_t \tag{6}$$

with $k = \frac{s+1}{3s+1}$ and $k' = \frac{s-1}{3s+1}$ as before. The *final challenge* of our proof is to show that such recursion implies sufficient growth of $A_t$. This is where careful algebra comes into play; we show that (6) implies that $A_T \geq \Omega\big((\sum_{t \in [T]} r_t)^{(3s+1)/2} D_0^{-(s-1)/2}\big)$ (see Lemmas 3 and 4) which establishes our result since $\sum_{t \in [T]} r_t = \frac{T-1}{2}$.

## 3 MS oracle implementations

In this section we describe several oracles that satisfy both Definition 1 (the MS condition) and Definition 2 (movement bounds) and may therefore be used by Algorithm 1. Section 3.1 briefly reviews oracles that have appeared in prior work, while Section 3.2 and Section 3.3 describe our new adaptive oracle implementations. We summarize the key oracle properties and resulting complexity bounds in Table 1.

### 3.1 Oracles from prior work

Here we consider several previously-studied oracles of the form $(x, \lambda) = \mathcal{O}(y)$, where we omit the second argument $\lambda'$ since prior work does not leverage it to improve implementation efficiency.

| Assumption | Oracle | Complexity with Algorithm 1 | Lower bound |
|---|---|---|---|
| $\nabla^p f$ is $(1,\nu)$-Hölder [*] | $\mathcal{O}_{p,\nu\text{-reg}}$ | $O\left(\epsilon^{-\frac{2}{3(p+\nu)-2}}\right)$ evals of $\nabla^p f$ | $\Omega\left(\epsilon^{-\frac{2}{3(p+\nu)-2}}\right)$ [3, 21] |
| $\nabla^3 f$ is 1-Lipschitz | $\mathcal{O}_{3\text{-reg-so}}$ | $O\left(\epsilon^{-\frac{1}{5}}\right)$ Hessian evals | $\Omega\left(\epsilon^{-\frac{1}{5}}\right)$ [3, 21] |
| N/A | $\mathcal{O}_{r\text{-ball}}$ | $O\left(r^{-\frac{2}{3}}\log\frac{1}{\epsilon}\right)$ oracle calls | $\Omega\left(r^{-\frac{2}{3}}\right)$ [12] |
| Stable Hessian | $\mathcal{O}_{r\text{-BaCoN}}$ | $O\left(r^{-\frac{2}{3}}\log\frac{1}{\epsilon}\right)$ Hessian evals | - |
| $\nabla^2 f$ is $(1,\nu)$-Hölder [†] | $\mathcal{O}_{\text{aMSN}}$ (Alg. 2) | $O\left(\epsilon^{-\frac{2}{4+3\nu}}\right)$ Hessian evals | $\Omega\left(\epsilon^{-\frac{2}{4+3\nu}}\right)$ [3, 21] |
| | | $O\left(\epsilon^{-\frac{2}{4+3\nu}}\right) + \widetilde{O}(1)$ linear systems | - |
| $\nabla f$ is $\ell$-Lipschitz and $\nabla^2 f$ is $(1,\nu)$-Hölder [†] | $\mathcal{O}_{\text{aMSN-fo}}$ (Alg. 3) | $O\left(\left(\frac{\epsilon}{\ell}\right)^{-\frac{1}{2}}\right) + \widetilde{O}(1)$ first-order evals | $\Omega\left(\left(\frac{\epsilon}{\ell}\right)^{-\frac{1}{2}}\right)$ [34] |
| | | $O\left(\min\left\{\left(\frac{\epsilon}{\ell}\right)^{-\frac{1}{2}}, \epsilon^{-\frac{2}{4+3\nu}}\right\}\right)$ iterations | - |

**Table 1.** Complexity bounds for finding $x$ such that $f(x) - f(x_\star) \leq \epsilon$ assuming $\|x - x_\star\| \leq 1$, attained by MS oracles from the literature (top 4 rows, described in Section 3.1) and oracles we develop (bottom two rows). In all cases we improve on prior work by a logarithmic factor. [*]We require $p + \nu \geq 2$. [†]Our adaptive oracles do not require knowledge of continuity constants or even the Hölder order $\nu \in [0, 1]$.

**Gradient descent step [e.g., 34].** As a gentle start, consider the oracle $\mathcal{O}_{\text{gd}}(y) = (y - \eta\nabla f(y), \frac{1}{\eta})$, i.e., an oracle that returns $x$ by taking standard gradient step with size $\eta$ and $\lambda = 1/\eta$. Obviously, the oracle always satisfies a $(1, \eta^{-1})$-movement bound. Moreover, if we assume that $\nabla f$ is $L$-Lipschitz, then $\|x - (y - \frac{1}{\lambda}\nabla f(x))\| = \eta\|\nabla f(x) - \nabla f(y)\| \leq \eta L\|x - y\|$. Therefore, when $\eta^{-1} \geq L/\sigma$ the oracle is a $\sigma$-MS oracle.

**Taylor descent step [5, 35, 19, 7, 23, 41].** Generalizing both $\mathcal{O}_{\text{gd}}$ and the cubic-regularized Newton step oracle $\mathcal{O}_{\text{cr}}$, we define for every integer $p \geq 1$ and $\nu \in [0, 1]$ the oracle $\mathcal{O}_{p,\nu\text{-reg}}$, that, for parameter $C$ and input $y$ returns $(x, \lambda) = \mathcal{O}_{p,\nu\text{-reg}}(y)$ where

$$x = \operatorname*{argmin}_{x' \in \mathbb{R}^d}\left\{\tilde{f}_p(x'; y) + \frac{M}{p!(p+\nu)}\|x' - y\|^{p+\nu}\right\} , \quad \lambda = \frac{M}{p!}\|x - y\|^{p+\nu-2} \tag{7}$$

and $\tilde{f}_p(x; y) := \sum_{i=0}^{p} \frac{1}{i!}\nabla^i f(y)[(x - y)^{\otimes i}]$ is the Taylor expansion of $f$ around $y$ evaluated at $x$. Oracles $\mathcal{O}_{\text{gd}}$ and $\mathcal{O}_{\text{cr}}$ correspond to the special cases $\mathcal{O}_{1,1\text{-reg}}$ (with $\eta = M^{-1}$) and $\mathcal{O}_{2,1\text{-reg}}$, respectively. Clearly, by definition, the oracle always satisfies a $(p + \nu - 1, (M/p!)^{1/(p+\nu-1)})$-movement bound. Moreover, it is easy to show that

$$\left\|x - \left(y - \frac{1}{\lambda}\nabla f(x)\right)\right\| = \frac{1}{\lambda}\|\nabla f(x) - \nabla\tilde{f}_p(x; y)\| = \frac{p!}{M}\frac{\|\nabla f(x) - \nabla\tilde{f}_p(x; y)\|}{\|x - y\|^{p+\nu-2}}.$$

For any $p \geq 1$ and $\nu \in [0, 1]$ we say that

$\nabla^p f$ is $(H, \nu)$-Hölder if for all $x, y$ we have $\|\nabla^p f(x) - \nabla^p f(y)\|_{\text{op}} \leq H\|x - y\|^\nu$.

(An $(H, 1)$-Hölder derivative is $H$-Lipschitz.) If $\nabla^p f$ is $(H, \nu)$-Hölder, Taylor's theorem gives $\|\nabla f(x) - \nabla\tilde{f}_p(x; y)\| \leq \frac{H}{p!}\|x - y\|^{p+\nu-1}$ [41, Lemma 2.5], and so $\|x - (y - \frac{1}{\lambda}\nabla f(x))\| \leq \frac{H}{M}\|x - y\|$. Therefore, when $M \geq H/\sigma$ the oracle is a $\sigma$-MS oracle.

**Exploiting third-order smoothness with a second order oracle [36, 26].** For $p > 2$, computing $\mathcal{O}_{p,\nu\text{-reg}}$ is typically intractable due to the need to compute the high-order derivative tensors $\nabla^3 f(y), \nabla^4 f(y), \ldots, \nabla^p f(y)$. Nevertheless for $p = 3$ Nesterov [36] designs an approximate solver for (7), which we denote $\mathcal{O}_{3\text{-reg-so}}$, using only $\nabla^2 f(y)$ and a logarithmic number of gradient evaluations. When $\nabla^3 f$ is $(L_3, 1)$-Hölder, [36] shows that $\mathcal{O}_{3\text{-reg-so}}$ is a valid MS-oracle satisfying a $(3, O(L_3))$-movement bound, on par with the movement bound of $\mathcal{O}_{3,1\text{-reg}}$.

**Exact ball optimization oracle [12].** For a given query $y$, consider the exact minimizer of $f$ constrained to a ball of radius $r$ around $y$, i.e., consider an oracle $\mathcal{O}_{r\text{-ball}}$ such that $(x, \lambda) = \mathcal{O}_{r\text{-ball}}(y)$ satisfy $x \in \operatorname{argmin}_{x': \|x'-y\| \leq r} f(x')$ and $\lambda = \frac{\|\nabla f(x)\|}{\|x-y\|}$. One may easily verify that (unless $\lambda = \|\nabla f(x)\| = 0$) we have $x = y - \frac{1}{\lambda} \nabla f(x)$, and therefore the oracle is a 0-MS oracle. Moreover, when $f$ is convex, we have either $\|x - y\| = r$ or $x$ is a global minimizer of $f$, and so we may assume without loss of generality that the oracle satisfies an $(\infty, 1/r)$ movement bound.

**Ball-Constrained Newton (BaCoN) oracle [12].** Exactly implementing $\mathcal{O}_{r\text{-ball}}$ is generally intractable. Nevertheless, Carmon et al. [12, Alg. 3] describe a method $\mathcal{O}_{r\text{-BaCoN}}$ based on solving a sequence of $\widetilde{O}(1)$ trust-region problems (ball-constrained Newton steps), which we call that, for functions that are $O(1)$-Hessian stable in a ball of radius $r$ (or $1/r$-quasi-self-concordant) and have a finite condition number, outputs $(x, \lambda)$ satisfying the $\frac{1}{2}$-MS oracle condition and an $(\infty, O(1/r))$-movement bound. Implementing $\mathcal{O}_{r\text{-BaCoN}}$ requires only a single Hessian evaluation and a number of linear system solutions that is polylogarithmic in problem parameters. Subsequent works implementing ball oracles [13, 4, 11] satisfy an approximation guarantee different than the MS condition, similar to the one we describe in Appendix C.

### 3.2 An adaptive Monteiro-Svaiter-Newton oracle

The oracle implementations in Section 3.1 satisfy movement bounds by design and the MS condition (2) by assumption. For example, the cubic-regularized Newton step oracle $\mathcal{O}_{cr}$ is guaranteed to satisfy the MS condition only when the regularization parameter $M$ is sufficiently larger than the Lipschitz constant of $\nabla^2 f$. This suggests that $M$ must be carefully tuned to ensure good performance. Prior work attempt to dynamically adjust $M$ in order to meet certain approximation conditions [14, 20, 21, 24]. However, even computing a single cubic-regularized Newton step entails searching for $\lambda$ that satisfies $\|[\nabla^2 f(y) + \lambda I]^{-1} \nabla f(y)\| = \frac{M\lambda}{2}$. Therefore, such a search over $M$ is essentially a (potentially) redundant double search over $\lambda$.

We propose a more direct and more adaptive MS oracle recipe: *search for the smallest $\lambda$ for which the regularized Newton step $x = y - [\nabla^2 f(y) + \lambda I]^{-1} \nabla f(y)$ satisfies the MS condition* (2).[7] This yields valid MS oracle by construction, independently of any assumption. Moreover, it is simple to argue that when $\nabla^2 f$ is $(H, \nu)$-Hölder continuous for some $\nu \in [0, 1]$, such oracle would guarantee the same movement bound as $\mathcal{O}_{2, \nu\text{-reg}}$ with the best choice parameters $M$ and $\nu$ (see Appendix D.1)—even though our recipe requires neither of these parameters!

Exactly fulfilling this recipe, i.e., finding the ideal minimal $\lambda^\star$ that satisfies the MS condition, is difficult. Fortunately, to adaptively guarantee movement bounds, it suffices to find a value $\lambda$ such the corresponding regularized Newton step satisfies the MS condition, while the step corresponding to $\lambda/2$ does not; Algorithm 2 finds precisely such a $\lambda$.

Let us describe the operation of Algorithm 2. If the input $\lambda'$ is invalid (i.e., its corresponding regularized Newton step does not satisfy the MS condition so that $\text{CHECKMS}(\lambda'; y, \sigma)$ evaluates to False), we set $\lambda_{\text{invld}} \leftarrow \lambda'$ and test a double-exponentially increasing series of $\lambda$'s, until reaching a valid $\lambda_{\text{vld}}$ (line 11). If $\lambda'$ is valid and the LAZY flag is set, we return it immediately. Otherwise (if LAZY is not set) we set $\lambda_{\text{vld}} = \lambda'$ and decrease it at a double-exponential rate until finding an invalid $\lambda_{\text{invld}}$ (line 5). In either case (so long as LAZY is not set) we obtain an (invalid,valid) pair $(\lambda_{\text{invld}}, \lambda_{\text{vld}})$ such that $\lambda_{\text{vld}}/\lambda_{\text{invld}} = 2^{2^{k^\star}}$ at the cost of $2 + k^\star$ linear system solutions. We then perform precisely $k^\star$ log-scale bisection steps in order to shrink $\lambda_{\text{vld}}/\lambda_{\text{invld}}$ down to 2 while maintaining the invariant that $\lambda_{\text{vld}}$ is valid and $\lambda_{\text{invld}}$ is invalid (line 15).

The following theorem bounds the complexity of Algorithm 2 in terms of linear-system solution number, and establishes a movement bound for its output assuming that $\nabla^2 f$ is locally Hölder around the query point. We defer the proof of the theorem and its following corollary to Appendix D.2.

**Theorem 2.** *Algorithm 2 with parameter $\sigma$ is a $\sigma$-MS oracle $\mathcal{O}_{\text{aMSN}}$. For any $y \in \mathbb{R}^d$, computing $(x, \lambda) = \mathcal{O}_{\text{aMSN}}(y)$ requires at most $2 + 2\log_2\left(1 + \left|\log_2 \frac{\lambda}{\lambda'}\right|\right)$ linear systems solutions. If LAZY is False or $\lambda > \lambda'$, and if $\nabla^2 f$ is $(H, \nu)$-Hölder in a ball of radius $2\|x - y\|$ around $y$, then $(x, y, \lambda)$ satisfy a $\left(1 + \nu, (2H/\sigma)^{1/(1+\nu)}/\sigma\right)$-movement bound.*

---

[7] The prior works [30, 17] also directly consider quadratically-regularized Newton steps, but employ approximation conditions other than (2) to select the parameter $\lambda$.

**Algorithm 2:** $\mathcal{O}_{\text{aMSN}}$

**Input:** Query $y \in \mathbb{R}^d$, $\lambda' > 0$. Flag LAZY.
**Parameters:** MS factor $\sigma \in (0,1)$.

1. **if** CHECKMS$(\lambda'; y, \sigma)$ **then**
2.     **if** LAZY **then return**
    $y - [\nabla^2 f(y) + \lambda' I]^{-1} \nabla f(y)$, $\lambda'$
3.     **else**
4.         $\lambda_{\text{vld}} \leftarrow \lambda'$ , $k \leftarrow 0$
5.         **while** CHECKMS$(\lambda_{\text{vld}}/2^{2^k}; y, \sigma)$ **do**
6.             $\lambda_{\text{vld}} \leftarrow \lambda_{\text{vld}}/2^{2^k}$
7.             $k \leftarrow k + 1$
8.         $k^\star \leftarrow k$ , $\lambda_{\text{invld}} \leftarrow \lambda_{\text{vld}}/2^{2^{k^\star}}$
9. **else**
10.     $\lambda_{\text{invld}} \leftarrow \lambda'$ , $k \leftarrow 0$
11.     **while not** CHECKMS$(\lambda_{\text{invld}} 2^{2^k}; y, \sigma)$ **do**
12.         $\lambda_{\text{invld}} \leftarrow \lambda_{\text{invld}} 2^{2^k}$
13.         $k \leftarrow k + 1$
14.     $k^\star \leftarrow k$ , $\lambda_{\text{vld}} \leftarrow \lambda_{\text{invld}} 2^{2^{k^\star}}$
15. **while** $\lambda_{\text{invld}} < \lambda_{\text{vld}}/2$ **do**
16.     $\lambda \leftarrow \sqrt{\lambda_{\text{invld}} \lambda_{\text{vld}}}$
17.     **if** CHECKMS$(\lambda; y, \sigma)$ **then** $\lambda_{\text{vld}} \leftarrow \lambda$
18.     **else** $\lambda_{\text{invld}} \leftarrow \lambda$
19. **return** $y - [\nabla^2 f(y) + \lambda_{\text{vld}} I]^{-1} \nabla f(y)$, $\lambda_{\text{vld}}$

20. **function** CHECKMS$(\lambda; y, \sigma)$
21.     $x = y - [\nabla^2 f(y) + \lambda I]^{-1} \nabla f(y)$
22.     **if** $\|x - (y - \frac{1}{\lambda}\nabla f(x))\| \leq \sigma \|x - y\|$
    **then return** True
23.     **else return** False

---

**Algorithm 3:** $\mathcal{O}_{\text{aMSN-fo}}$

**Input:** $y \in \mathbb{R}^d$, $\lambda' > 0$. Flag LAZY.
**Parameters:** MS factor $\sigma \in (0,1)$.

1. $\lambda \leftarrow \lambda'$ , FAILEDCHECK $\leftarrow$ False
2. **Repeat**
3.     $A \leftarrow \nabla^2 f(y) + \lambda I$ , $b \leftarrow -\nabla f(y)$
    ▷ Apply MinRes/Conjugate Residuals [18]
    until obtaining $w$ s.t. $\|Aw - b\| \leq \frac{\lambda \sigma}{2}\|w\|$
4.     $x \leftarrow y + $ CONJRES$(A, b, \lambda\sigma)$
5.     **if** $\|x - (y - \frac{1}{\lambda}\nabla f(x))\| \leq \sigma\|x - y\|$
    **then**
6.         **if** LAZY or FAILEDCHECK **then**
7.             **return** $x, \lambda$
8.         **else** $\lambda \leftarrow \lambda/2$
9.     **else**
10.         FAILEDCHECK $\leftarrow$ True
11.         $\lambda \leftarrow 2\lambda$

12. **function** CONJRES$(A, b, \lambda\sigma)$
13.     $w_0 \leftarrow 0$
14.     $p_0 \leftarrow r_0 \leftarrow Aw_0 - b$   ▷ $r_i = Aw_i - b$
15.     $s_0 \leftarrow q_0 \leftarrow Ar_0$   ▷ $q_i = Ap_i$
16.     $i \leftarrow 0$
17.     **while** $\|r_i\| > \frac{\lambda\sigma}{2}\|w_i\|$ **do**
18.         $w_{i+1} \leftarrow w_i - \frac{\langle r_i, s_i \rangle}{\|q_i\|^2} p_i$
19.         $r_{i+1} \leftarrow r_i - \frac{\langle r_i, s_i \rangle}{\|q_i\|^2} q_i$
20.         $s_{i+1} \leftarrow Ar_{i+1}$
21.         $p_{i+1} \leftarrow \frac{\langle r_{i+1}, s_{i+1} \rangle}{\langle r_i, s_i \rangle} p_i + r_{i+1}$
22.         $q_{i+1} \leftarrow \frac{\langle r_{i+1}, s_{i+1} \rangle}{\langle r_i, s_i \rangle} q_i + s_{i+1}$
23.         $i \leftarrow i + 1$
24.     **return** $w_i$

---

To understand the LAZY option of Algorithm 2, note that when $\lambda'$ is valid we will necessarily output $\lambda \leq \lambda'$. In such case Theorem 1 does not require a movement bound (except for the first iteration). Therefore, we might as well save on computation and return $\lambda'$. The following Corollary 3 gives the overall complexity bound for the combination of Algorithm 1 and $\mathcal{O}_{\text{aMSN}}$, leveraging "lazy" oracle calls to show that the number of linear system solves per iteration is essentially constant.

**Corollary 3.** *Consider Algorithm 1 with initial point $x_0$, parameters $\alpha$ satisfying $1.1 \leq \alpha = O(1)$ and $\lambda'_0$, and $\sigma$-MS oracle $\mathcal{O}_{\text{aMSN}}$ (with LAZY $= True$ in all but the first iteration) with $\sigma \in (0.01, 0.99)$. For any $H, \epsilon > 0$, $\nu \in [0,1]$ and any $x_\star \in \mathbb{R}^d$, if $f$ is convex with $(H, \nu)$-Hölder Hessian, the algorithm produces an iterate $x_T$ such that $f(x_T) \leq f(x_\star) + \epsilon$ using $T = O\left(\left(H\|x_0 - x_\star\|^{2+\nu}/\epsilon\right)^{2/(4+3\nu)}\right)$ Hessian evaluations and $O\left(T + \log\log\max\left\{\frac{HR^\nu}{\lambda'_0}, \frac{\lambda'_0 R^2}{\epsilon}\right\}\right)$ linear system solutions, where $R$ is the distance between $x_0$ and $\operatorname{argmin}_{x'} f(x')$.*

Note that as long as $\lambda'_0$ is in the range $\left[2^{-2^T}HR^\nu, 2^{2^T}\epsilon R^{-2}\right]$, the double logarithmic term in our bound on linear system solution number is $O(T)$. Therefore, the overall bound is $O(T)$ for an extremely large range of $\lambda'_0$ values.

### 3.3 First-order implementation via MinRes/Conjugate Residuals

We now present a first-order implementation of our adaptive oracle, $\mathcal{O}_{\text{aMSN-fo}}$ (Algorithm 3), which replaces exact linear system solutions with approximations obtained via Hessian-vector products and the MinRes/Conjugate Residuals method [42, 18]. Similar to Algorithm 2, the algorithm searches

for $\lambda$ such that $x_\lambda \approx y - [\nabla^2 f(y) + \lambda I]^{-1}\nabla f(y)$ satisfies the MS condition, but $x_{\lambda/2}$ does not. Departing from the double-exponential scheme of Algorithm 2, here we adopt the following doubling scheme that allows us to control the cost of the $x_\lambda$ approximation. If $\lambda'$ is such that $x_{\lambda'}$ does not satisfy the MS condition, we repeatedly test $\lambda = 2\lambda', 4\lambda', 8\lambda', \ldots$ and return the first one for which $x_\lambda$ satisfies the MS condition. If $x_{\lambda'}$ satisfies the MS condition and the algorithm is LAZY, we immediately return it. Otherwise, we repeatedly test $\lambda = \frac{1}{2}\lambda', \frac{1}{4}\lambda', \frac{1}{8}\lambda', \ldots$ until reaching $\lambda$ for which $x_\lambda$ does not satisfy the MS condition, and return $x_{2\lambda}$.

The subroutine CONJRES of Algorithm 3 takes as input a matrix $A$, a vector $b$, and accuracy parameter $\lambda\sigma$, and iteratively generates $\{w_i\}$ that approximate $A^{-1}b$. The construction of the MinRes/Conjugate Residuals method guarnatees that $w_i$ minimizes the norm of the residual $r_i = Aw_i - b$ in the Krylov subspace $\text{span}\{b, Ab, \ldots, A^{i-1}b\}$. The key algorithmic decision here is when to stop the iterations: stop too early, and the approximation for the Newton step might not be accurate enough to guarantee a movement bound; stop too late, and incur a high Hessian-vector product complexity. We introduce a simple stopping condition (line 17) that strikes a balance. On the one hand, we show that whenever the condition $\|r_i\| \leq \frac{\lambda\sigma}{2}\|w_i\|$ holds, the resulting point $x$ can certify roughly the same movement bounds as exact Newton steps. On the other hand, by invoking the complexity bounds in [28] and using the the optimality of $\|r_i\|$, we guarantee that the stopping condition is met within a number of iterations proportional to $1/\sqrt{\lambda}$. The structure of our doubling scheme for $\lambda$ then allows us to relate the overall first-order complexity to the lowest value of $\lambda$ queried, obtaining the following guarantees. See proofs in Appendix D.3.

**Theorem 4.** *Algorithm 3 with parameter $\sigma$ is a $\sigma$-MS oracle $\mathcal{O}_{\text{aMSN-fo}}$. For any $y \in \mathbb{R}^d$, computing $(x, \lambda) = \mathcal{O}_{\text{aMSN-fo}}(y)$ requires at most $O\left(\sqrt{1 + \frac{\|\nabla^2 f(y)\|_{\text{op}}}{\sigma\min\{\lambda,\lambda'\}}}\right)$ Hessian-vector product and $O\left(\left|\log\frac{\lambda}{\lambda'}\right|\right)$ gradient computations. If LAZY is False or $\lambda > \lambda'$, and if $\nabla^2 f$ is $(H, \nu)$-Hölder, then $(x, y, \lambda)$ satisfy a $\left(1 + \nu, (6H/\sigma)^{1/(1+\nu)}\right)$-movement bound.*

**Corollary 5.** *Consider Algorithm 1 with initial point $x_0$, parameters $\alpha$ satisfying $1.1 \leq \alpha = O(1)$ and $\lambda'_0$, and $\sigma$-MS oracle $\mathcal{O}_{\text{aMSN-fo}}$ with LAZY set to True in all but the first iteration and $\sigma \in (0.01, 0.99)$. For any $L, H, \epsilon > 0$, $\nu \in [0, 1]$ and any $x_\star \in \mathbb{R}^d$, if $f$ is convex with $(H, \nu)$-Hölder Hessian and $L$-Lipschitz gradient, the algorithm produces an iterate $x_T$ such that $f(x_T) \leq f(x_\star) + \epsilon$ within $T = O\left(\left(\frac{H\|x_0 - x_\star\|^{2+\nu}}{\epsilon}\right)^{2/(4+3\nu)}\right)$ iterations and at most $O\left(\left(\frac{L\|x_0 - x_\star\|^2}{\epsilon}\right)^{1/2} + \sqrt{\frac{L}{\lambda'_0}} + \log\frac{\lambda'_0}{L}\right)$ gradient and Hessian-vector product evaluations.*

Note that the $L$-Lipschitz gradient assumption implies an $(L, 0)$-Hölder Hessian assumption, giving the iteration complexity bound we state in Table 1. Moreover, note that our algorithm has the optimal $O(\sqrt{L\|x_0 - x_\star\|^2/\epsilon})$ complexity for any $\lambda'_0$ in the range $\Omega(\epsilon/\|x_0 - x_\star\|^2)$ to $L\exp\left\{O(\sqrt{L\|x_0 - x_\star\|^2/\epsilon})\right\}$. By choosing a large $\lambda'_0$ (say $10^6$) we may guarantee that only the logarithmic term is added to the optimal first-order evaluation complexity.

## 4 Experiments

We conduct three sets of experiments. First, we consider $\mathcal{O}_{\text{cr}}$ with a fixed parameter $M$ and compare previous acceleration schemes to Algorithm 1. Second, we combine Algorithm 1 with our adaptive $\mathcal{O}_{\text{aMSN}}$ and test it against previous adaptive accelerated (second-order) methods and Newton's method. Finally, we compare Algorithm 1 with our first-order adaptive oracle $\mathcal{O}_{\text{aMSN-fo}}$ to other first-order methods. We provide full implementation details in Appendix E.1. Figure 1 summarizes our results for logistic regression on the 'a9a' dataset [15]; see Appendix E.2 for similar results on three additional datasets. These experiments were conducted with *no tuning* of Algorithm 1: the parameters $\sigma$ and $\alpha$ were simply set to $\frac{1}{2}$ and 2, respectively. An additional experiment, reported in Appendix E.3, shows that the algorithm is indeed insensitive to that choice.

**Non-adaptive methods.** We use the non-adaptive oracle $\mathcal{O}_{\text{cr}}$ (1), and take $M$ to be $0.2\bar{H}$ where, for feature vectors $\phi_1, \ldots, \phi_n$, $\bar{H} = \|\frac{1}{n}\sum_{i=1}^n \phi_i\phi_i^T\|_{\text{op}}\max_{i\in[n]}\|\phi_i\|$ is an upper bound on $6\sqrt{3} \approx 10$ times the Lipschitz constant of the logistic regression Hessian [see, e.g., 41]. Fixing the MS oracle

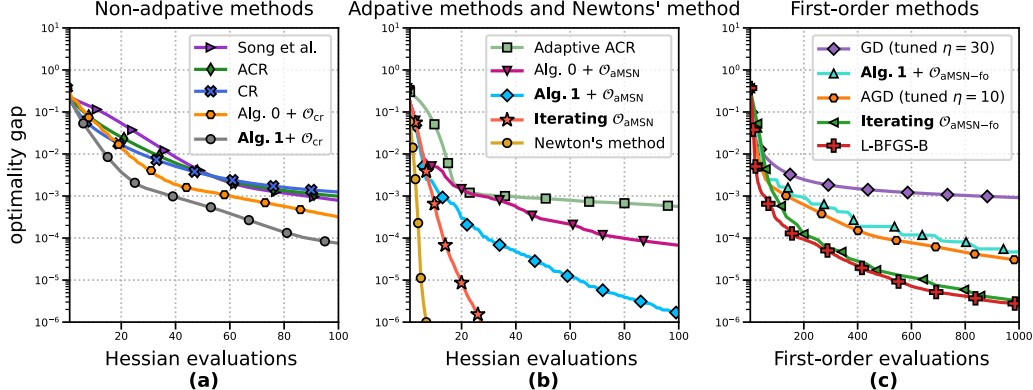

**Figure 1.** Empirical results for logistic regression on the "a9a" dataset. See Section 4 for description, and Appendix E.2 for additional datasets. Boldface legend entries denote methods we contribute.

allows for a controlled comparison of different acceleration schemes: Figure 1(a) shows that standard MS acceleration with a carefully-implemented bisection outperforms standard cubic regularization (CR) and its accelerated counterpart (ACR) [33, Alg. 4.8], and removing the bisection via Algorithm 1 yields the best results. We also implemented the heuristic suggested by Song et al. [41], where instead of a bisection in Algorithm 0 we select a sequence $\lambda'_t$ such that $A_t = \frac{1}{M\|x_0 - x_\star\|}(t/3)^{7/2}$. In Appendix E.4 we tune the $M$ parameter for each method separately, finding that the optimal $M$ for CR is near 0, so that $\mathcal{O}_{\mathsf{cr}}$ is nearly a Newton step (and not a valid MS oracle).

**Adaptive methods and Newton's method.** We compare the following adaptive accelerated second-order methods (which do not require an estimate of the Hessian Lipschitz constant): Adaptive ACR [21, Algorithm 4] (which adaptively sets $M$ in $\mathcal{O}_{\mathsf{cr}}$), standard MS acceleration (Algorithm 0) with $\mathcal{O}_{\mathsf{aMSN}}$ (Algorithm 2, with LAZY = False) and Algorithm 1 with $\mathcal{O}_{\mathsf{aMSN}}$ (with LAZY = True in all but the first iteration). Figure 1(b) shows that the latter converges significantly faster than the other adaptive acceleration schemes. However, the classical "unaccelerated" Newton iteration $x_{t+1} = -(\nabla^2 f(x_t))^{-1}\nabla f(x_t)$ strongly outperforms all "accelerated" methods, indicating that momentum mechanisms might actually be slowing down convergence in logistic regression problems. To test this, we consider the following simple iteration of (the non-lazy variant) of our oracle: $x_{t+1}, \lambda_{t+1} = \mathcal{O}_{\mathsf{aMSN}}(x_t; \lambda_t/2)$; it significantly improves over Algorithm 1.

These results beg the question: is momentum ever useful for second-order methods? In Appendix E.5 we test different schemes on the lower bound construction [3, 21]. We find momentum is helpful for $\mathcal{O}_{\mathsf{cr}}$, but not for the adaptive oracle $\mathcal{O}_{\mathsf{aMSN}}$. What makes Newton's method perform so well on logistic regression, and whether simply iterating $\mathcal{O}_{\mathsf{aMSN}}$ is worst-case optimal, are important questions for future work.

**First-order methods.** We compare our first-order adaptive $\mathcal{O}_{\mathsf{aMSN\text{-}fo}}$ (Algorithm 3) to the following baselines: gradient descent and accelerated gradient descent [38] with a tuned step size $\eta$, and L-BFGS-B from SciPy [10, 44, 43]. In light of the above comparison with Newton's method, we also test the following simple iteration of (the lazy variant) of our oracle: $x_{t+1}, \lambda_{t+1} = \mathcal{O}_{\mathsf{aMSN\text{-}fo}}(x_t; \lambda_t/2)$. Figure 1(c) shows that forgoing (second-order) momentum is better for the first-order oracle, too: Algorithm 1 performs comparably to tuned AGD (without tuning a single parameter), and the equally adaptive $\mathcal{O}_{\mathsf{aMSN\text{-}fo}}$ iteration performs comparably to with L-BFGS-B.

## Acknowledgments

We thank the anonymous reviewers for helpful questions and suggestions, leading to important writing clarifications, a simplification of the proof of Lemma 3, and an improved complexity bound for second-order methods under third-order smoothness (due to the observation that our framework is compatible with the oracle of [36]). YC and DH were supported in part by the Israeli Science Foundation (ISF) grant no. 2486/21, the Len Blavatnik and the Blavatnik Family foundation and the Adelis Foundation. YJ was supported in part by a Stanford Graduate Fellowship and the Dantzig-Lieberman Fellowship. AS was supported in part by a Microsoft Research Faculty Fellowship, NSF CAREER Award CCF-1844855, NSF Grant CCF-1955039, a PayPal research award, and a Sloan Research Fellowship.

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
