| 10 $\qquad x_{t+1} = \tilde{x}_{t+1}$ |
| 11 $\qquad \lambda'_{t+2} = \frac{1}{\alpha}\lambda'_{t+1}$ |
| 12 $\quad$ **else** |
| 13 $\qquad \gamma_{t+1} = \frac{\lambda'_{t+1}}{\lambda_{t+1}}$ |
| 14 $\qquad a_{t+1} = \gamma_{t+1}a'_{t+1}, \ A_{t+1} = A_t + a_{t+1}$ |
| 15 $\qquad x_{t+1} = \frac{(1-\gamma_{t+1})A_t}{A_{t+1}}x_t + \frac{\gamma_{t+1}A'_{t+1}}{A_{t+1}}\tilde{x}_{t+1}$ |
| 16 $\qquad \lambda'_{t+2} = \alpha\lambda'_{t+1}$ |
| 17 $\quad v_{t+1} = v_t - a_{t+1}\nabla f(\tilde{x}_{t+1})$ |

| **Algorithm 0:** MS Acceleration |
| --- |
| **Input:** Initial $x_0$, function $f$, oracle $\mathcal{O}$ |
| **Parameters:** Bisection limits $\lambda^\ell, \lambda^h$, and tolerance $\rho > 1$ |
| 1 Set $v_0 = x_0$, $A_0 = 0$ |
| 2 **for** $t = 0, 1, \ldots,$ **do** |
| 3 $\quad \lambda^\ell_{t+1}, \lambda^h_{t+1} = \lambda^\ell, \lambda^h$ |
| 4 $\quad \lambda'_{t+1} = \frac{\lambda^\ell_{t+1}+\lambda^h_{t+1}}{2}$ |
| 5 $\quad a'_{t+1} = \frac{1}{2\lambda'_{t+1}}\left(1 + \sqrt{1 + 4\lambda'_{t+1}A_t}\right)$ |
| 6 $\quad A'_{t+1} = A_t + a'_{t+1}$ |
| 7 $\quad y_t = \frac{A_t}{A'_{t+1}}x_t + \frac{a'_{t+1}}{A'_{t+1}}v_t$ |
| 8 $\quad \tilde{x}_{t+1}, \lambda_{t+1} = \mathcal{O}(y_t; \lambda'_{t+1})$ |
| 9 $\quad$ **if** $\lambda_{t+1} \in \left[\frac{1}{\rho}\lambda'_{t+1}, \lambda'_{t+1}\right]$ **then** |
| 10 $\qquad a_{t+1} = a'_{t+1}, \ A_{t+1} = A'_{t+1}$ |
| 11 $\qquad x_{t+1} = \tilde{x}_{t+1}$ |
| 12 $\quad$ **else if** $\lambda_{t+1} < \frac{1}{\rho}\lambda'_{t+1}$ **then** |
| 13 $\qquad \lambda^h_{t+1} = \lambda'_{t+1}$ |
| 14 $\qquad$ Go to line 4 |
| 15 $\quad$ **else** |
| 16 $\qquad \lambda^l_{t+1} = \lambda'_{t+1}$ |
| 17 $\qquad$ Go to line 4 |

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

# A    Additional related work

**Bisection-free methods for variational inequalities.**    Monteiro and Svaiter also proposed second-order methods for solving variational inequalities for monotone operators with continuous derivatives [31], and subsequent work provided improved rates for variational inequalities with continuous higher-order derivatives via tensor methods [9, 25]. These works also feature an implicit equation over a scalar regularization/step-size parameter, that necessitates a bisection and increases complexity by a logarithmic factor. In recent papers, Lin and Jordan [29] and Adil et al. [1] remove that logarithmic factor by developing bisection-free methods for variational inequalities. However, applying these methods directly to convex optimization with Lipschitz $p$th derivatives yields a rate of $O(t^{-(p+1)/2})$ rather than the optimal $O(t^{-(3p+1)/2})$ rate of our method. Moreover, these works remove bisections using techniques fundamentally different from ours. In particular, they do not apply a damping scheme on the $A_t$ sequence, nor do they apply a multiplicative update for the regularization parameter.

**Adaptive Newton and tensor methods.**    A number of works consider adaptive variants of the cubic-regularized Newton method and its tensor counterparts. Cartis et al. [14], Gould et al. [20] propose adaptive variants of cubic regularization for non-convex optimization. For convex optimization, Mishchenko [30] provides a simple adaptive scheme converging at rate $O(t^{-2})$, followed by an improvement in its guarantee to $O(t^{-3})$ [17]. For tensor methods, Jiang et al. [24] proposes an adaptive regularization scheme for convex functions with Lipschitz-continuous $p^{th}$ derivatives which achieves the rate $O(t^{-p-1})$. In addition, Grapiglia and Nesterov [21] gives analogous results under $\nu$-Hölder continuity of the derivatives.

Even for the case second-order methods with Lipschitz-continuous Hessian, an adaptive scheme with optimal rate $O(t^{-3.5})$ remained open prior to this work. Beyond removing the bisection from the MS framework, our key algorithmic techniques include directly considering quadratically-regularized Newton step (similar to [30, 17] and different from [21, 24]), and using the original MS approximation condition for selecting an appropriate regularization parameter, which is new in the context of adaptive methods. These techniques allow us to adapt to both the constant and order of Hessian Hölder continuity simultaneously.

**Comparison to [27].**    In concurrent and independent theoretical work, Kovalev and Gasnikov [27] propose an algorithm that also attains the optimal $p$th derivative evaluation complexity for convex optimization with Lipschitz $p$th derivatives. While also inspired by MS acceleration, the algorithm of [27] is quite different from ours: they replace the implicitly defined regularization parameters inherent to MS oracles by approximating proximal points with explicit, predetermined regularization parameters. To obtain these proximal points they apply a tensor-extragradient method, and stop it when the MS condition is met. By careful analysis, they show that—even though an individual outer acceleration step requires multiple derivative evaluations—the overall complexity of their method is optimal. In contrast, our method makes a single oracle call (high-order derivative evaluation) per step. To obtain optimal complexity, our method relies on a dynamic sequence of "guessed" regularization parameters and a momentum damping schemes that handles cases where these guesses overshoot. The two works offer complementary viewpoints of the algorithmic innovation required for removing bisection from the MS framework.

In comparison to Kovalev and Gasnikov [27], we believe that our algorithm offers advantages in terms of generality and adaptivity. From a generality perspective, our algorithm applies to every setting where the original MS framework applies. In addition to functions with Lipschitz $p$th derivatives, that includes functions with Hölder derivatives [41], ball minimization oracles [12], and a second-order oracles for functions with Lipschitz third derivative [36, 26]. While extending [27] to these settings may be possible, it would require additional work in formulating and analyzing an appropriate subproblem solver. Regarding adaptivity, like the original MS framework, our algorithm is agnostic to both the order of the Lipschitz derivative and its corresponding Lipschitz constant. In contrast, Kovalev and Gasnikov [27] require the derivative order for determining the regularization parameters, and the Lipschitz constant for the subproblem solver.

# B Proof of Theorem 1

In this section, we provide a complete proof for Theorem 1. We begin in Appendix B.1 with establishing a potential decrease result analogous to the standard analyses of accelerated proximal methods (Proposition 1). In Appendix B.2 we then provide a lower bound of $A_T$ in terms of the values of $\lambda'_t$ at a subset of "up" iterations where $\lambda'_t > \lambda_t$ (Lemma 1). We apply these results along with the reverse Hölder inequality to obtain Theorem 1 in Appendix B.3. The last part of the analysis relies on two technical lemmas on the growth rates of sequences satisfying certain recursive inequalities (Lemmas 3 and 4), which we prove at the end of the section in Appendix B.4. Beyond its utility for proving Theorem 1, Lemma 1 includes additional lower bounds on $A_T$ in terms of $\lambda'_t$ which we use in the analysis of adaptive oracle implementations in Appendix D.

We use the following notation for the set of "down" iterations:

$$\mathcal{S}_{T}^{\leq} := \{t \in [T] \mid \lambda_t \leq \lambda'_t\}$$

and analogously define $\mathcal{S}_{T}^{>}$ ("up" iterations), $\mathcal{S}_{T}^{\geq}$, $\mathcal{S}_{T}^{<}$ and $\mathcal{S}_{T}^{=}$.

## B.1 Potential decrease

**Proposition 1.** *Under the assumptions of Theorem 1, let $E_t := f(x_t) - f(x_\star)$, $D_t := \frac{1}{2}\|v_t - x_\star\|^2$, and $N_{t+1} := \frac{1}{2}\|\tilde{x}_{t+1} - y_t\|^2$ for all $t \geq 0$. Then, for all $t \geq 0$*

$$A_{t+1}E_{t+1} + D_{t+1} + (1 - \sigma^2)A'_{t+1}\min(\lambda_{t+1}, \lambda'_{t+1})N_{t+1} \leq A_t E_t + D_t. \qquad (8)$$

*Consequently, for all $T \geq 1$, $\sqrt{A_T} \geq \frac{1}{2}\sum_{t \in \mathcal{S}_{T}^{\leq}} 1/\sqrt{\lambda'_t}$,*

$$E_T \leq \frac{D_0}{A_T} \quad, \text{ and } \quad (1 - \sigma^2)\sum_{t \in \mathcal{S}_{T}^{\geq}} A_t \lambda'_t N_t \leq D_0 - A_T E_T. \qquad (9)$$

*Proof.* By definition of $D_t$ and the definition of $v_{t+1}$ in line 17, we have

$$D_{t+1} = \frac{1}{2}\|v_{t+1} - x_\star\|^2 = \frac{1}{2}\|(v_t - a_{t+1}\nabla f(\tilde{x}_{t+1})) - x_\star\|^2$$

$$= D_t + a_{t+1}\langle \nabla f(\tilde{x}_{t+1}), x_\star - v_t\rangle + \frac{a_{t+1}^2}{2}\|\nabla f(\tilde{x}_{t+1})\|^2. \qquad (10)$$

Also, by the definition $y_t$ in line 6 and $A'_{t+1} := A_t + a'_{t+1}$ in line 5, we have

$$a'_{t+1}v_t = A'_{t+1}y_t - A_t x_t = a'_{t+1}\tilde{x}_{t+1} + A'_{t+1}(y_t - \tilde{x}_{t+1}) - A_t(x_t - \tilde{x}_{t+1}).$$

Subtracting $a'_{t+1}x_\star$ from both sides and considering the inner product with $\nabla f(\tilde{x}_{t+1})$ then yields

$$a'_{t+1}\langle \nabla f(\tilde{x}_{t+1}), x_\star - v_t\rangle$$
$$= \nabla f(\tilde{x}_{t+1})^\top \left[a'_{t+1}(x_\star - \tilde{x}_{t+1}) + A'_{t+1}(\tilde{x}_{t+1} - y_t) + A_t(x_t - \tilde{x}_{t+1})\right]$$
$$\overset{(i)}{\leq} a'_{t+1}[f(x_\star) - f(\tilde{x}_{t+1})] + A'_{t+1}\langle \nabla f(\tilde{x}_{t+1}), \tilde{x}_{t+1} - y_t\rangle + A_t[f(x_t) - f(\tilde{x}_{t+1})]$$
$$\overset{(ii)}{\leq} A_t E_t - A'_{t+1}[f(\tilde{x}_{t+1}) - f(x_\star)] + A'_{t+1}\langle \nabla f(\tilde{x}_{t+1}), \tilde{x}_{t+1} - y_t\rangle.$$

where we used $(i)$ convexity of $f$ and $(ii)$ again that $A'_{t+1} = A_t + a'_{t+1}$ (line 5).

Next, note that we can upper bound $\langle \nabla f(\tilde{x}_{t+1}), x_\star - v_t\rangle$ as

$$\lambda_{t+1}\langle \nabla f(\tilde{x}_{t+1}), \tilde{x}_{t+1} - y_t\rangle$$
$$= \frac{1}{2}\|\nabla f(\tilde{x}_{t+1}) + \lambda_{t+1}(\tilde{x}_{t+1} - y_t)\|^2 - \frac{1}{2}\|\nabla f(\tilde{x}_{t+1})\|^2 - \frac{\lambda_{t+1}^2}{2}\|\tilde{x}_{t+1} - y_t\|^2$$
$$\leq -\lambda_{t+1}^2(1 - \sigma^2)N_{t+1} - \frac{1}{2}\|\nabla f(\tilde{x}_{t+1})\|^2,$$

where for the inequality we used that $\mathcal{O}$ is a $\sigma$-MS oracle for $f$ (definition 1) and the definition of $N_t$. Substituting back gives

$$A'_{t+1}[f(\tilde{x}_{t+1}) - f(x_\star)] \leq$$

$$A_t E_t + a'_{t+1} \langle \nabla f(\tilde{x}_{t+1}), v_t - x_\star \rangle - (1 - \sigma^2) A'_{t+1} \lambda_{t+1} N_{t+1} - \frac{A'_{t+1}}{2\lambda_{t+1}} \|\nabla f(\tilde{x}_{t+1})\|^2. \quad (11)$$

We separately consider the cases $\lambda_{t+1} \leq \lambda'_{t+1}$ and $\lambda_{t+1} > \lambda'_{t+1}$. First, when $\lambda_{t+1} \leq \lambda'_{t+1}$, by definition in the algorithm $x_{t+1} = \tilde{x}_{t+1}$, $a_{t+1} = a'_{t+1}$, $A_{t+1} = A'_{t+1}$ and by line 4 we have $A_{t+1} = \lambda'_{t+1} a^2_{t+1}$. Consequently, we can combine (10) and (11) to conclude that

$$A_{t+1} E_{t+1} + D_{t+1} + \lambda_{t+1} A_{t+1} (1 - \sigma^2) N_{t+1} \leq A_t E_t + D_t + \left( \frac{a^2_{t+1}}{2} - \frac{A_{t+1}}{2\lambda_{t+1}} \right) \|\nabla f(\tilde{x}_{t+1})\|^2$$

$$\leq A_t E_t + D_t. \tag{12}$$

On the other hand, when $\lambda_{t+1} > \lambda'_{t+1}$, by the definition of $\gamma_{t+1} = \lambda'_{t+1}/\lambda_{t+1}$ in line 13, $a_{t+1}, A_{t+1}$ in line 14, and $x_{t+1}$ in line 15, we have $A_{t+1} = (1 - \gamma_{t+1}) A_t + \gamma_{t+1} A'_{t+1}$, and therefore convexity of $f$ implies that

$$f(x_{t+1}) \leq \frac{(1 - \gamma_{t+1}) A_t}{A_{t+1}} f(x_t) + \frac{\gamma_{t+1} A'_{t+1}}{A_{t+1}} f(\tilde{x}_{t+1}).$$

Subtracting $f(x_\star)$, multiplying by $A_{t+1}$, combining with (11) to bound $f(\tilde{x}_{t+1})$ and noting that $\gamma_{t+1} a'_{t+1} = a_{t+1}$ yields

$$A_{t+1} E_{t+1} \leq (1 - \gamma_{t+1}) A_t E_t + \gamma_{t+1} A'_{t+1} [f(\tilde{x}_{t+1}) - f(x_\star)]$$

$$\leq A_t E_t + a_{t+1} \langle \nabla f(\tilde{x}_{t+1}), x_\star - v_t \rangle - (1 - \sigma^2) A'_{t+1} \lambda'_{t+1} N_{t+1} - \frac{\gamma_{t+1} A'_{t+1}}{2\lambda_{t+1}} \|\nabla f(\tilde{x}_{t+1})\|^2.$$

Noting that $A'_{t+1} = \lambda'_{t+1} (a'_{t+1})^2 = \frac{\lambda_{t+1}}{\gamma_{t+1}} a^2_{t+1}$ by definition and further substituting (10) into the above display yields

$$A_{t+1} E_{t+1} \leq A_t E_t + D_t - D_{t+1} - (1 - \sigma^2) A'_{t+1} \lambda'_{t+1} N_{t+1}, \tag{13}$$

which, when combined with (12) yields (8).

The bound on $A_T$ follows from standard argument for Monteiro-Svaiter acceleration restricting to the proper set, i.e. $\mathcal{S}_T^\leq$, see e.g. Lemma 27 in [12], we include here for completeness.

$$\sqrt{A_T} = \sqrt{A_T} - \sqrt{A_0} = \sum_{t=0}^{T-1} \frac{A_{t+1} - A_t}{\sqrt{A_{t+1}} + \sqrt{A_t}} \geq \sum_{t+1 \in \mathcal{S}_T^\leq} \frac{a'_{t+1}}{\sqrt{A'_{t+1}} + \sqrt{A_t}}$$

$$= \sum_{t+1 \in \mathcal{S}_T^\leq} \frac{\sqrt{A'_{t+1}/\lambda'_{t+1}}}{\sqrt{A'_{t+1}} + \sqrt{A_t}} \geq \frac{1}{2} \sum_{t \in \mathcal{S}_T^\leq} \sqrt{1/\lambda'_t}.$$

For the second line we used that $\lambda'_{t+1}(a'_{t+1})^2 = A'_{t+1}$ and that $A'_t$ is increasing in $t$. Finally, the conclusions in (9) follow from inductively applying (8) and using $A_0 = 0$. $\qquad \square$

## B.2  Lower bounding $A_T$ using "up" iterates

Next, we provide more fine-grained bounds on the growth of $A_t$, implied by the adaptive scheme for updating $\lambda'$ in line 11 and 16.

**Lemma 1.** *In the setting of Theorem 1, for any $\widehat{T} \geq 1$, there exists a non-empty set $\mathcal{Q}_{\widehat{T}} \subseteq \mathcal{S}_{\widehat{T}}^> \cup \{1\}$ and positive numbers $r_t$ for each $t \in \mathcal{Q}_{\widehat{T}}$ such that*

$$\sum_{t \in \mathcal{Q}_{\widehat{T}}} r_t = \frac{\widehat{T} - 1}{2}, \tag{14}$$

*and*

$$\sqrt{A_{\widehat{T}}} \geq \frac{1}{4\sqrt{\alpha}} \sum_{t \in \mathcal{Q}_{\widehat{T}}} \sqrt{\frac{\alpha^{r_t - 1}}{\lambda'_t}}. \tag{15}$$

*Further, the definition is consistent in the sense that for any $T \geq 1$ and defined $\mathcal{Q}_T$, for any $\widehat{T}_1, \widehat{T}_2 \in \mathcal{Q}_T$, suppose $\widehat{T}_1 < \widehat{T}_2$ and $r_{t,\widehat{T}}$ are the numbers when applying previous argument to $\widehat{T}$, then $\mathcal{Q}_{\widehat{T}_1} \subseteq \mathcal{Q}_{\widehat{T}_2}$ and $r_{t,\widehat{T}_1} = r_{t,\widehat{T}_2}$ for any $t \leq \widehat{T}_1$. Thus, we omit the second subscript in defining $r_t$ when clear from context.*

*Furthermore, for $T \geq 1$,*

$$\sqrt{A_T} \geq \frac{\sqrt{\alpha} - 1}{4\alpha} \sum_{t \in [T]} \sqrt{\frac{1}{\lambda'_t}}. \tag{16}$$

*Proof.* We define $\mathcal{Q}_{\widehat{T}}$ to be the set of "up-down" iterates, i.e., iterates $t$ for which $\lambda_t > \lambda'_t$ but $\lambda_{t+1} \leq \lambda'_{t+1}$; we also add to $\mathcal{Q}_{\widehat{T}}$ the first iterate and, if $\widehat{T} \in \mathcal{S}^{\geq}_{\widehat{T}}$, the iterate $\widehat{T}$. Formally, we have

$$\mathcal{Q}_{\widehat{T}} := (\mathcal{S}^{>}_{\widehat{T}} \cap \{t \mid t+1 \in \mathcal{S}^{\leq}_{\widehat{T}} \text{ or } t = \widehat{T}\}) \cup \{1\}.$$

We let $1 = \tau_1 < \tau_2 < \cdots < \tau_S \leq \widehat{T}$ denote the $S = |\mathcal{Q}_{\widehat{T}}|$ distinct elements of $\mathcal{Q}_{\widehat{T}}$ in increasing order. For notational convenience, we also let $\tau_{S+1} := \widehat{T}$.

For every $i \in [S]$, we let $n_i$ be the index of the last "down" iterate between $\tau_i$ and $\tau_{i+1}$ (and $\widehat{T}$ if $i = S$), that is

$$n_i := \begin{cases} \max\{t \in \mathcal{S}^{\leq}_{\widehat{T}} | \tau_i \leq t < \tau_{i+1}\} & \text{if } i < S \\ \widehat{T} & \text{otherwise} \end{cases}. \tag{17}$$

As an immediate consequence of the definition of $\tau_i$ and $n_i$, we have for all $i < S$, $n_i \in [\tau_i, \tau_{i+1})$. We also have that the set of $n_i$ are distinct, i.e. $n_i \neq n_j$ for all $i, j \in [S]$ with $i \neq j$.

Note that between any two "up-down" iterates $\tau_i$ and $\tau_{i+1}$ we have a sequence of "down" iterates (ending at $n_i$) followed by a sequence of "up iterates" (ending at $\tau_{i+1}$). In other words, for all $i < S$ and $k \in (n_i, \tau_{i+1}]$ we have $k \in \mathcal{S}^{\geq}_{\widehat{T}}$. Consequently, $\lambda'_{k+1} = \alpha \lambda'_k$ for all $k \in (n_i, \tau_{i+1})$ (since these are "up" iterates). Since $\lambda'_{n_i+1} = \alpha^{-1} \lambda'_{n_i}$ (because $n_i$ is a "down" iterate), we conclude that $\lambda'_{n_i} = \alpha^{2-(\tau_{i+1}-n_i)} \lambda'_{\tau_{i+1}}$. Combining this with Proposition 1 implies the following lower bound on $\sqrt{A_{\widehat{T}}}$:

$$\sqrt{A_{\widehat{T}}} \geq \frac{1}{2} \sum_{t \in \mathcal{S}^{\leq}_{\widehat{T}}} \frac{1}{\sqrt{\lambda'_t}} \geq \frac{1}{2} \sum_{i \in [S-1]} \frac{1}{\sqrt{\lambda'_{n_i}}} \geq \frac{1}{2} \sum_{i \in [S-1]} \sqrt{\frac{\alpha^{\tau_{i+1}-n_i-2}}{\lambda'_{\tau_{i+1}}}}. \tag{18}$$

Further, as argued above, for $k \in (\tau_i, n_i]$ we have $k \in \mathcal{S}^{\leq}_{\widehat{T}}$ for all and therefore $\lambda'_{k+1} = \lambda'_k/\alpha$. Consequently, when $\tau_i < \widehat{T}$ we have $\lambda'_{n_i} = \alpha^{2-(n_i-\tau_i)} \lambda'_{\tau_i}$. When $\tau_i = \widehat{T}$ the inequality also holds since $\tau_i = n_i$. Together with the conclusion of Proposition 1, this implies the following lower bound on $\sqrt{A_{\widehat{T}}}$:

$$\sqrt{A_{\widehat{T}}} \geq \frac{1}{2} \sum_{t \in \mathcal{S}^{\leq}_{\widehat{T}}} \frac{1}{\sqrt{\lambda'_t}} \geq \frac{1}{2} \sum_{i \in [S]} \frac{1}{\sqrt{\lambda'_{n_i}}} \geq \frac{1}{2} \sum_{i \in [S]} \sqrt{\frac{\alpha^{n_i-\tau_i-2}}{\lambda'_{\tau_i}}}. \tag{19}$$

We now define $r_i$ as follows

$$r_{\tau_i} = \begin{cases} \frac{1}{2}(n_1 - 1) & \text{if } i = 1 \\ \frac{1}{2}(n_i - n_{i-1}) & \text{if } 1 < i \leq S. \end{cases}$$

Clearly we have $r_t \geq 0$ for all $t \in \mathcal{Q}_{\widehat{T}}$ and $\sum_{t \in \mathcal{Q}_{\widehat{T}}} r_t = \sum_{i \in [S]} r_{\tau_i} = \frac{\widehat{T}-1}{2}$, which proves (14).

To show (15), note that for any $\alpha \geq 1$, $\frac{1}{2}\sqrt{\alpha^a} + \frac{1}{2}\sqrt{\alpha^b} \geq \sqrt{\alpha^{\frac{1}{2}a+\frac{1}{2}b}}$ due to the arithmetic and geometric mean (AM-GM) inequality. Averaging our two lower bounds on $\sqrt{A_{\widehat{T}}}$, (18) and (19), we conclude that

$$\sqrt{A_{\widehat{T}}} \geq \frac{1}{4}\left(\sum_{i=2}^{S}\sqrt{\frac{\alpha^{\tau_i-n_{i-1}-2}}{\lambda'_{\tau_i}}} + \sum_{i\in[S]}\sqrt{\frac{\alpha^{n_i-\tau_i-2}}{\lambda'_{\tau_i}}}\right) \geq \frac{1}{4}\sum_{i\in[S]}\sqrt{\frac{\alpha^{r_{\tau_i}-2}}{\lambda'_{\tau_i}}} = \frac{1}{4}\sum_{t\in\mathcal{Q}_{\widehat{T}}}\sqrt{\frac{\alpha^{r_t-2}}{\lambda'_t}}.$$

Here the first term on the RHS bound comes purely from $\sqrt{\frac{\alpha^{n_i-\tau_i-2}}{\lambda'_{\tau_i}}}$ when $i=1$ since $n_1 - \tau_1 > \frac{1}{2}(n_1-\tau_1) = \frac{1}{2}r_{\tau_1}$ which leads to the coefficient of $1/4$ on RHS.

Now for the consistency arguments, note by definition of $\mathcal{Q}$ and $r_t$ we have $\mathcal{Q}_{\widehat{T}_1} \subseteq \mathcal{Q}_{\widehat{T}_2}$ and $r_{t,\widehat{T}_1} = r_{t,\widehat{T}_2}$ for any $\widehat{T}_1 < \widehat{T}_2 \in \mathcal{Q}_{\widehat{T}}$.

To show the second inequality (16), we start again with $\widehat{T} = T$. From the conclusion of Proposition 1 and the observation that $k \in \mathcal{S}_T^{\leq}$ for any $k \in (\tau_i, n_i]$, giving

$$\sqrt{A_T} \geq \frac{1}{2}\sum_{t\in\mathcal{S}_T^{\leq}}\frac{1}{\sqrt{\lambda'_t}} = \frac{1}{2}\sum_{t=1}^{n_1}\sqrt{\frac{1}{\lambda'_t}} + \frac{1}{2}\sum_{i=2}^{S}\sum_{t=\tau_i+1}^{n_i}\sqrt{\frac{1}{\lambda'_t}}. \tag{20}$$

Moreover, since for $i < S$ and $k \in (n_i, \tau_{i+1}]$ we have $\lambda'_k = \alpha^{(k-n_i-2)}\lambda'_{n_i}$, and

$$\sum_{t\in(n_i,\tau_{i+1}]}\sqrt{\frac{1}{\lambda'_t}} = \left(\sum_{j=1}^{\tau_{i+1}-n_i}\frac{\alpha}{\alpha^{j/2}}\right)\sqrt{\frac{1}{\lambda'_{n_i}}} \leq \frac{\alpha}{\sqrt{\alpha}-1}\sqrt{\frac{1}{\lambda'_{n_i}}}. \tag{21}$$

Combining (20) and (21) with $\sqrt{A_T} \geq \frac{1}{2}\sum_{i\in[S-1]}\sqrt{\frac{1}{\lambda'_{n_i}}}$ yields (16) since

$$\sqrt{A_T} \geq \frac{1}{4}\left(\sum_{t=1}^{n_1}\sqrt{\frac{1}{\lambda'_t}} + \sum_{i=2}^{S}\sum_{t=\tau_i+1}^{n_i}\sqrt{\frac{1}{\lambda'_t}}\right) + \frac{1}{4}\sum_{i=1}^{S-1}\sqrt{\frac{1}{\lambda'_{n_i}}}$$

$$\geq \frac{1}{4}\left(\sum_{t=1}^{n_1}\sqrt{\frac{1}{\lambda'_t}} + \sum_{i=2}^{S}\sum_{t=\tau_i+1}^{n_i}\sqrt{\frac{1}{\lambda'_t}}\right) + \frac{\sqrt{\alpha}-1}{4\alpha}\sum_{i=1}^{S-1}\sum_{t\in(n_i,\tau_{i+1}]}\sqrt{\frac{1}{\lambda'_t}}$$

$$\geq \frac{\sqrt{\alpha}-1}{4\alpha}\sum_{t\in[T]}\sqrt{\frac{1}{\lambda'_t}}.$$

$\square$

## B.3 Completing the proof of Theorem 1

We now show how to use Proposition 1 and Lemma 1 to obtain optimal acceleration, considering the cases $s \in (1, \infty)$, $s = \infty$, and $s = 1$ in turn.

**The $s \in (1, \infty)$ case.** If $E_T \leq 0$, the result $f(x_T) - f(x_\star) \leq 0$ follows immediately. Therefore, it suffices to consider the case when $E_T > 0$. For any $\widehat{T} \in \mathcal{Q}_T$, applying Proposition 1 and Lemma 1 (using that movement bounds hold for all iterations in $\mathcal{Q}_T$ including the first iterate $t = 1$ by assumption) yields

$$D_0 \geq D_0 - A_{\widehat{T}}E_{\widehat{T}} \geq \sum_{t\in\mathcal{Q}_{\widehat{T}}}A_t\lambda'_t(1-\sigma^2)M_t \geq \frac{1-\sigma^2}{2}c^{-\frac{2s}{s-1}}\sum_{t\in\mathcal{Q}_{\widehat{T}}}A_t(\lambda'_t)^{\frac{s+1}{s-1}} \geq 0. \tag{22}$$

This implies $E_{\widehat{T}} \leq D_0/A_{\widehat{T}}$ where $\sqrt{A_{\widehat{T}}} \geq \frac{1}{4\sqrt{\alpha}}\sum_{t\in\mathcal{Q}_{\widehat{T}}}\sqrt{\frac{\alpha^{r_t-1}}{\lambda'_t}}$ for $\sum_{t\in\mathcal{Q}_{\widehat{T}}}r_t = \frac{\widehat{T}-1}{2}$.

The reverse Hölder inequality (which is a standard technique in analyzing MS acceleration [19, 7, 23, 41, 2]) states that, for all $q > 1$, and any two vectors $u, v$ with positive elements,

$$\sum_i u_i v_i \geq \left(\sum_i u_i^{1/q}\right)^q \left(\sum_i v_i^{1/(1-q)}\right)^{1-q}.$$

We set $q = \frac{3s+1}{2(s+1)}$ and apply the reverse Hölder inequality to obtain

$$
4\sqrt{\alpha A_{\widehat{T}}} \geq \sum_{t \in \mathcal{Q}_{\widehat{T}}} \sqrt{\frac{\alpha^{r_t-1}}{\lambda_t'}} = \sum_{t \in \mathcal{Q}_{\widehat{T}}} \left( A_t^{q-1}\sqrt{\alpha^{r_t-1}} \right) \left( \frac{A_t^{1-q}}{\sqrt{\lambda_t'}} \right)
$$

$$
\overset{(i)}{\geq} \left( \sum_{t \in \mathcal{Q}_{\widehat{T}}} A_t^{1-\frac{1}{q}} \alpha^{\frac{r_t-1}{2q}} \right)^q \left( \sum_{t \in \mathcal{Q}_{\widehat{T}}} A_t (\lambda_t')^{\frac{s+1}{s-1}} \right)^{1-q}
$$

$$
\overset{(ii)}{\geq} \left( \sum_{t \in \mathcal{Q}_{\widehat{T}}} A_t^{1-\frac{1}{q}} \left( \alpha^{\frac{1}{2q}} \right)^{r_t-1} \right)^q \left( \frac{2}{1-\sigma^2} D_0 c^{\frac{2s}{s-1}} \right)^{1-q}
$$

$$
\overset{(iii)}{\geq} \left( \sum_{t \in \mathcal{Q}_{\widehat{T}}} A_t^{1-\frac{1}{q}} r_t \cdot c_{\alpha,q} \right)^q \left( \frac{2D_0}{(1-\sigma^2)c^{-\frac{2s}{s-1}}} \right)^{1-q} \quad \text{for } c_{\alpha,q} := \min\left( 1, \frac{1}{2q}\ln\alpha \right)
$$

$$\tag{23}$$

where we used $(i)$ the reverse Hölder inequality with $u_t = A_t^{q-1}\sqrt{\alpha^{r_t-1}}$ and $v_t = A_t^{1-q}/\sqrt{\lambda_t'}$ (for $t \in \mathcal{Q}_{\widehat{T}}$) and $-\frac{1}{2} \cdot \frac{1}{1-q} = \frac{s+1}{s-1}$, $(ii)$ the bound (22), and $(iii)$ the following lemma (proved in the next subsection) with $a \leftarrow \alpha^{1/2q}$ and $b \leftarrow r_t \geq 0$.

**Lemma 2.** *For all $a \geq 0$ and $b \geq 1$, we have $a^{b-1} \geq \min\{1, \ln a\} \cdot b$.*

Substituting the definitions

$$
B_t := A_t^{1-\frac{1}{q}} \quad \text{and} \quad \beta := c_{\alpha,q}\left( \frac{1}{4\sqrt{\alpha}} \right)^{\frac{1}{q}} \left( \frac{2D_0}{(1-\sigma^2)c^{-\frac{2s}{s-1}}} \right)^{\frac{1-q}{q}},
$$

the bound (23) can be rewritten as

$$
B_{\widehat{T}}^{\frac{s+1}{s-1}} = B_{\widehat{T}}^{\frac{1}{2(1-q)}} \geq \beta \cdot \sum_{\tau \in \mathcal{Q}_{\widehat{T}}} B_\tau \cdot r_\tau \quad \text{for all } \widehat{T} \in \mathcal{Q}_T.
$$

To deduce the growth rate of $B_t$, we give the following lemma generalizing the analyses in prior work [7, 23] (see proof in the next Appendix B.4).

**Lemma 3.** *Let $B_1, ..., B_k \in \mathbb{R}_{>0}$, $r_1, ..., r_k \in \mathbb{R}_{\geq 0}$ and $\beta > 0$. Further, suppose that for some $m > 1$ and all $i \in [k]$ it is the case that $B_i^m \geq \beta \sum_{j \in [i]} B_j \cdot r_j$. Then for all $i \in [k]$ we have that $B_i \geq \left( \frac{m-1}{m}\beta \cdot \sum_{j \in [i]} r_j \right)^{1/(m-1)}$.*

Applying the lemma with $m = \frac{s+1}{s-1}$ and recalling that $\sum_{t \in \mathcal{Q}_T} r_t = \frac{T-1}{2}$, we obtain (for $T' = \max \mathcal{Q}_T$),

$$
B_T \geq B_{T'} \geq \left( \frac{\frac{s+1}{s-1}-1}{\frac{s+1}{s-1}} \cdot \beta \cdot \sum_{t \in \mathcal{Q}_T} r_t \right)^{\frac{1}{\frac{s+1}{s-1}-1}} = \left( \frac{2}{s+1} \cdot \beta \cdot \frac{T-1}{2} \right)^{\frac{s-1}{2}} \quad \text{for any } T \in \mathbb{Z}_{>0}.
$$

Since $A_T = B_T^{\frac{q}{q-1}}$ and $\frac{q}{q-1} = \frac{3s+1}{s-1}$ this gives the desired growth rate of

$$
A_T \geq \left( \frac{2}{s+1} \cdot \beta \cdot \frac{T-1}{2} \right)^{\frac{3s+1}{2}} \quad \text{for any } T \in \mathbb{Z}_{>0},
$$

where, substituting back, we have

$$
\beta = \min\left( 1, \frac{s+1}{3s+1}\ln\alpha \right) \cdot \left( \frac{1}{4\sqrt{\alpha}} \right)^{\frac{2s+2}{3s+1}} c^{-\frac{2s}{3s+1}} \left( \frac{2}{(1-\sigma^2)}D_0 \right)^{-\frac{s-1}{3s+1}}.
$$

Thus, for

$$T = \Omega\left(\frac{\left(\frac{\frac{1}{2}\|x_0 - x_\star\|^2}{\epsilon}\right)^{\frac{2}{3s+1}}}{\frac{1}{s+1}\beta}\right) = \Omega\left(\frac{\alpha^{\frac{s+1}{3s+1}}}{\min\left(1, \frac{1}{s}\ln\alpha\right)} \cdot \left(\frac{c^s\|x_0 - x_\star\|^{s+1}}{\epsilon}\right)^{\frac{2}{3s+1}}\right),$$

we have $A_T \geq \frac{1}{2}\|x_0 - x_\star\|^2/\epsilon$, and consequently

$$f(x_T) - f(x_\star) = E_T \leq \frac{\frac{1}{2}\|x_0 - x_\star\|^2}{A_T} \leq \epsilon.$$

The case for $s \in (1, \infty)$ follows immediately.

**The $s = \infty$ case.** Considering $s = \infty$ and $q = \frac{3}{2}$ in (23) yields for any $T \in \mathbb{Z}_{>0}$ and $\widehat{T} \in \mathcal{Q}_T$,

$$4\sqrt{\alpha A_{\widehat{T}}} \geq \left(\sum_{t \in \mathcal{Q}_{\widehat{T}}} A_t^{\frac{1}{3}} \min\left(1, \frac{1}{3}\ln\alpha\right) \cdot r_t\right)^{\frac{3}{2}} \left(\frac{2c^2 D_0}{1 - \sigma^2}\right)^{-\frac{1}{2}}.$$

Defining

$$B_t := A_t^{1/3} \quad \text{and} \quad \beta := \min\{1, \tfrac{1}{3}\ln\alpha\} \cdot \left(2^5 c^2 \alpha D_0/(1 - \sigma^2)\right)^{-1/3}$$

we have

$$B_{\widehat{T}} \geq \beta \sum_{\tau \in \mathcal{Q}_{\widehat{T}}} B_\tau \cdot r_\tau, \quad \text{for all } \widehat{T} \in \mathcal{Q}_T.$$

We deduce an exponential rate of growth for $B_t$ using the following lemma, inspired by the analysis in [12] (and proved in the next subsection).

**Lemma 4.** *Let $B_1, ..., B_k \in \mathbb{R}_{>0}$ be non-decreasing and let $r_1, ..., r_k \in \mathbb{R}_{\geq 0}$ and $R_i := \sum_{j \in [i]} r_j$ for $i \in [k]$. Further, suppose that for some $\beta > 0$, and all $i \in [k]$ it is the case that $B_i \geq \beta \cdot \sum_{j \in [i]} B_j \cdot r_j$. Then $B_i \geq \exp(\beta R_i - 1)B_1$ for all $i \in [k]$.*

Applying the lemma and substituting back the definition of $B_t$, we obtain,

$$A_T^{1/3} \geq \exp\left(\beta \sum_{t \in \mathcal{Q}_T} r_t - 1\right) A_1^{1/3} = \exp\left(\beta \cdot \frac{T-1}{2} - 1\right) A_1^{1/3},$$

where we let $\beta := \min\left(1, \frac{1}{3}\ln\alpha\right) \cdot \left(\frac{2^5 c^2 \alpha D_0}{(1 - \sigma^2)}\right)^{-\frac{1}{3}}$.

Thus, for

$$T = \Omega\left(\frac{\log\frac{\|x_0 - x_\star\|^2}{\epsilon A_1}}{\beta}\right) = \Omega\left(\frac{\alpha^{\frac{1}{3}}}{\min\left(1, \frac{1}{3}\ln\alpha\right)} \cdot (c\|x_0 - x_\star\|)^{\frac{2}{3}} \log\frac{\|x_0 - x_\star\|^2}{\epsilon A_1}\right),$$

we have $A_T \geq \frac{1}{2}\|x_0 - x_\star\|^2/\epsilon$, and consequently

$$f(x_T) - f(x_\star) = E_T \leq \frac{\frac{1}{2}\|x_0 - x_\star\|^2}{A_T} \leq \epsilon,$$

which proves the case for $s = \infty$.

**The $s = 1$ case.** This case corresponds to the standard analysis of Nesterov acceleration. The $(1, c)$-movement bound guarantees that $\lambda_t \leq c$ for all $t$. Recalling that $\lambda_t' \leq \lambda_t$ for all $t \in \mathcal{Q}_T$, the bound (15), yields

$$4\sqrt{\alpha A_T} \geq \sum_{t \in \mathcal{Q}_T} \sqrt{\frac{\alpha^{r_t - 1}}{c}} \geq \frac{\min\{1, \frac{1}{2}\ln\alpha\}}{\sqrt{c}} \sum_{t \in \mathcal{Q}_T} r_t = \frac{\min\{1, \frac{1}{2}\ln\alpha\}}{\sqrt{c}} \cdot \frac{T-1}{2},$$

where the final bound uses Lemma 1. Consequently, the error bound (9) we have

$$E_T \leq \frac{D_0}{A_T} = O\left(\frac{\alpha c D_0}{\min\{1, \ln^2\alpha\}T^2}\right),$$

yielding the claimed result for $s = 1$.

### B.4 Helper lemmas

**Lemma 2.** *For all $a \geq 0$ and $b \geq 1$, we have $a^{b-1} \geq \min\{1, \ln a\} \cdot b$.*

*Proof.* Define the difference function $f(x) := a^{x-1} - \min(1, \ln a) \cdot x$. We note that clearly $f(1) \geq 0$ and the first-order derivative $f'(x) = (\ln a) \cdot a^{x-1} - \min(1, \ln a) \geq 0$ for all $x \geq 1$. Consequently, by the integral formula $f(x) = \int_1^x f'(z)dz$ we have that $f(x) \geq 0$ for all $x \geq 1$. $\qquad\square$

**Lemma 3.** *Let $B_1, ..., B_k \in \mathbb{R}_{>0}$, $r_1, ..., r_k \in \mathbb{R}_{\geq 0}$ and $\beta > 0$. Further, suppose that for some $m > 1$ and all $i \in [k]$ it is the case that $B_i^m \geq \beta \sum_{j \in [i]} B_j \cdot r_j$. Then for all $i \in [k]$ we have that*
$$B_i \geq \left( \tfrac{m-1}{m} \beta \cdot \sum_{j \in [i]} r_j \right)^{1/(m-1)}.$$

*Proof.* Without loss of generality we take $\beta = 1$, since otherwise we may redefine $r_i$ to be $\beta r_i$. Furthermore, we assume $r_1 > 0$ as otherwise we can divide into the following two cases:

1. if all $r_i = 0$ the desired inequality naively holds;

2. if there exists some $i_0$ such that $r_{i_0} > 0$ and $r_i = 0$ for all $i < i_0$, then it suffices to consider the sequence starting from $i_0$.

First, for $i = 1$, note that $B_1^m \geq B_1 \cdot r_1$ and consequently $B_1 \geq (r_1)^{1/(m-1)} \geq (\tfrac{m-1}{m} \cdot r_1)^{1/(m-1)}$ as desired.

Next, for any $j > 1$, we have

$$\sum_{j' \in [j+1]} B_{j'} r_{j'} - \sum_{j' \in [j]} B_{j'} r_{j'} = B_{j+1} r_{j+1} \geq \left( \sum_{j' \in [j+1]} B_{j'} r_{j'} \right)^{1/m} r_{j+1},$$

and consequently

$$r_{j+1} \leq \frac{\sum_{j' \in [j+1]} B_{j'} r_{j'} - \sum_{j' \in [j]} B_{j'} r_{j'}}{\left( \sum_{j' \in [j+1]} B_{j'} r_{j'} \right)^{1/m}} \leq \int_{\sum_{j' \in [j]} B_{j'} r_{j'}}^{\sum_{j' \in [j+1]} B_{j'} r_{j'}} \frac{1}{t^{1/m}} dt$$

$$= \frac{m}{m-1} \left( \left( \sum_{j' \in [j+1]} B_{j'} r_{j'} \right)^{\frac{m-1}{m}} - \left( \sum_{j' \in [j]} B_{j'} r_{j'} \right)^{\frac{m-1}{m}} \right).$$

Summing the above inequality for all $j \in [i]$, noting that for $i = 1$ we have $r_1 \leq \frac{m}{m-1} (B_1 r_1)^{\frac{m-1}{m}}$, and rearranging terms yields

$$\left( \sum_{j \in [i+1]} B_j r_j \right)^{\frac{m-1}{m}} \geq \frac{m-1}{m} \sum_{j \in [i+1]} r_j.$$

Combining this with the condition $B_{i+1}^m \geq \sum_{j \in [i+1]} B_j r_j$ concludes the proof. $\qquad\square$

**Lemma 4.** *Let $B_1, ..., B_k \in \mathbb{R}_{>0}$ be non-decreasing and let $r_1, ..., r_k \in \mathbb{R}_{\geq 0}$ and $R_i := \sum_{j \in [i]} r_j$ for $i \in [k]$. Further, suppose that for some $\beta > 0$, and all $i \in [k]$ it is the case that $B_i \geq \beta \cdot \sum_{j \in [i]} B_j \cdot r_j$. Then $B_i \geq \exp(\beta R_i - 1) B_1$ for all $i \in [k]$.*

*Proof.* Let $k_0$ denote the largest element of $[k]$ for which $R_{k_0} < \beta^{-1}$ and let $k_0 = 0$ if there is no such element. Note that $\exp(\beta(R_{k_0} - \beta^{-1}) \leq 1$. Since $B_k$ increases monotonically in $k$ this implies that $B_i \geq \exp\left(\beta(R_i - \beta^{-1})\right) B_1$ for all $i \in [k_0]$.

For $i = k_0 + 1$, we have

$$\sum_{j' \in [i]} \beta B_{j'} r_{j'} - B_1 \geq \sum_{j' \in [i]} \beta B_{j'} r_{j'} - \sum_{j' \in [i-1]} \beta B_{j'} r_{j'} = \beta \cdot B_i r_i \geq \left( \sum_{j' \in [i]} \beta B_{j'} r_{j'} \right) \beta r_i,$$

where the first inequality is due to the definition of $k_0$ and that $B_i$ is non-increasing. Consequently,

$$\beta \cdot r_{k_0+1} \leq \frac{\sum_{j' \in [k_0+1]} \beta \cdot B_{j'} r_{j'} - B_1}{\left( \beta \cdot \sum_{j' \in [k_0+1]} B_{j'} r_{j'} \right)} \leq \int_{B_1}^{\sum_{j' \in [k_0+1]} \beta \cdot B_{j'} r_{j'}} \frac{1}{t} \, dt$$

$$= \log \left( \frac{\sum_{j' \in [k_0+1]} \beta \cdot B_{j'} r_{j'}}{B_1} \right).$$

For any $i + 1$ such that $2 \leq i + 1 \leq k$ and $j \in [i]$, we have

$$\sum_{j' \in [j+1]} \beta B_{j'} r_{j'} - \sum_{j' \in [j]} \beta B_{j'} r_{j'} = \beta \cdot B_{j+1} r_{j+1} \geq \left( \sum_{j' \in [j+1]} \beta B_{j'} r_{j'} \right) \beta r_{j+1},$$

and consequently

$$\beta \cdot r_{j+1} \leq \frac{\sum_{j' \in [j+1]} \beta \cdot B_{j'} r_{j'} - \sum_{j' \in [j]} \beta \cdot B_{j'} r_{j'}}{\left( \beta \cdot \sum_{j' \in [j+1]} B_{j'} r_{j'} \right)} \leq \int_{\sum_{j' \in [j]} \beta \cdot B_{j'} r_{j'}}^{\sum_{j' \in [j+1]} \beta \cdot B_{j'} r_{j'}} \frac{1}{t} \, dt$$

$$= \log \left( \frac{\beta \cdot \sum_{j' \in [j+1]} B_{j'} r_{j'}}{\beta \cdot \sum_{j' \in [j]} B_{j'} r_{j'}} \right).$$

Summing up above inequalities yields that for any $i \in [k_0 + 1, k]$ it is the case that

$$\beta \sum_{j=k_0+1}^{i} r_j \leq \log \left( \frac{1}{B_1} \sum_{j \in [i+1]} B_i r_i \right) \leq \log \left( \frac{B_i}{B_1} \right)$$

Since $\beta \sum_{j=k_0+1}^{i} r_j = \beta(R_i - R_{k_0}) \geq \beta(R_i - \beta^{-1})$, where we define $R_0 := 0$, we obtain that $B_i \geq \exp(\beta R_i - 1) B_1$ holds for all $i > k_0$, and hence for all $i \geq 1$. $\qquad \square$

## C  Generalized oracle notions

In this section, we consider a setting where the convex function $f$ may be non-differentiable (i.e., $\nabla f$ might not exist everywhere) and the problem may be constrained (i.e., the convex closed domain $\mathcal{X}$ may be different from $\mathbb{R}^d$). We consider two types of oracles: a slight generalization of the MS oracle for the non-differentiable and/or constrained setting, and a fairly different notion of a "stochastic proximal oracle" similar to the one considered in [4]. We also provide a slight variation of Algorithm 1 that makes use of these oracles and also allows more flexibility in choosing some of the iterates, and prove convergence rate bounds for this algorithm combined with either oracle.

To generalize Definition 1 of a MS oracle, we consider mappings that return, in addition to $x \in \mathcal{X}$ and $\lambda > 0$, a vector $g \in \mathbb{R}^d$ that replaces $\nabla f(x)$. More specifically, recall the definition of the subdifferential of $f$ at $x \in \mathcal{X}$:

$$\partial f(x) := \{ g \in \mathbb{R}^d \mid \langle g, x' - x \rangle \leq f(x') - f(x) \text{ for all } x' \in \mathcal{X} \}.$$

An element of $\partial f(x)$ is called a subgradient of $f$ at $x$. Note that when $f$ is differentiable at $x$ we have $\nabla f(x) \in \partial f(x)$. However, on the boundary of $\mathcal{X}$ there may additional elements in the subdifferential even when $f$ is differentiable. Our generalized MS oracle (that originally appeared in [32]) returns $g$, a subgradient of $f$ at $x$, such that the MS condition holds with $g$ instead of $\nabla f(x)$.

**Definition 3** (Generalized MS oracle). *An oracle $\mathcal{O} : \mathcal{X} \times \mathbb{R}_+ \to \mathcal{X} \times \mathbb{R}^d \times \mathbb{R}_+$ is a $\sigma$-Generalized MS oracle for function $f : \mathcal{X} \to \mathbb{R}$ if for every $y \in \mathcal{X}$ and $\lambda' > 0$, the points $(x, g, \lambda) = \mathcal{O}(y; \lambda')$ satisfy the following:*

$$g \in \partial f(x) \ \ and \ \ \left\| x - \left( y - \tfrac{1}{\lambda} g \right) \right\| \leq \sigma \| x - y \|. \tag{24}$$

We remark that considering subgradients instead of gradients is essential for handling constrained optimization even when $f$ is differentiable, because even exact proximal point do not necessarily satisfy the simple MS condition (2). That is, letting $F_\lambda(x) = f(x) + \frac{\lambda}{2} \| x - y \|^2$, the point $x_\lambda = \operatorname{argmin}_{x \in \mathcal{X}} F_\lambda(x)$ does not necessarily satisfy $x_\lambda = y - \frac{1}{\lambda} \nabla f(x_\lambda)$. Nevertheless, the first-order optimality conditions of characterizing $x_\lambda$ guarantee that $\lambda(y - x_\lambda) \in \partial f(x_\lambda)$. Therefore, exact proximal points are 0-Generalized MS oracles.

We now present a different oracle, with a probabilistic approximation condition that relates directly to the exact proximal point $x_\lambda$. The advantage of approximation conditions of this kind is that they can be efficiently satisfied using stochastic first-order methods in certain non-smooth problems where certifying (24) is hard [4].

**Definition 4** (Stochastic proximal oracle). *A (randomized) oracle $\mathcal{O} : \mathcal{X} \times \mathbb{R}_+ \to \mathcal{X} \times \mathbb{R}^d \times \mathbb{R}_+$ is a $\sigma$-stochastic proximal oracle for function $f : \mathbb{R}^d \to \mathbb{R}$ if for every $y \in \mathcal{X}$ and $\lambda' > 0$, the points $(x, g, \lambda) = \mathcal{O}(y; \lambda')$ satisfy the following:*

$$\mathbb{E} F_\lambda(x) \leq \min_{x' \in \mathcal{X}} F_\lambda(x') + \frac{\lambda \sigma^2}{4} \mathbb{E} \| x - y \|^2 \ , \ \ \mathbb{E} g = g_\lambda \ \ and \ \ \operatorname{Var}(g) \leq \frac{\sigma^2}{2} \mathbb{E} \| x - y \|^2, \tag{25}$$

*where $F_\lambda(x') := f(x') + \frac{\lambda}{2} \| x' - y \|^2$, $x_\lambda = \operatorname{argmin}_{x' \in \mathcal{X}} F_\lambda(x')$ and $g_\lambda = \lambda(y - x_\lambda)$, and all expectations are conditional on $y, \lambda'$ and $\lambda$.*

Three remarks are in order. First, note that exact proximal points are also 0-stochastic proximal oracles. Second, the condition $\mathbb{E} g = g_\lambda = \lambda(y - x_\lambda)$ implies that if the stochastic proximal oracle outputs a deterministic $g$ then it also computes $x_\lambda$ exactly. Third, a $\sigma$-Generalized MS oracle output $x, g, \lambda$ satisfies $g + \lambda(x - y) \in \partial F_\lambda(x)$ and therefore, by $\lambda$-strong convexity of $F_\lambda$,

$$F_\lambda(x) - \min_{x' \in \mathcal{X}} F_\lambda(x') \leq \frac{1}{2\lambda} \| g + \lambda(x - y) \|^2 \leq \frac{\lambda \sigma^2}{2} \| x - y \|^2.$$

Therefore, up to a replacing $\sigma$ with $\sigma/\sqrt{2}$, the generalized MS-condition (24) implies the first part of (25). Moreover, when $g$ is deterministic its variance is zero, giving the third part of the condition. The second part of the condition, however, is not directly implied by (24). Nevertheless, given a procedure that for any $\delta \geq 0$ outputs a point $x^\delta$ such that $F_\lambda(x^\delta) - \min_{x' \in \mathcal{X}} F_\lambda(x') \leq \frac{\lambda \delta^2}{2}$ (e.g., an $\sigma$-MS oracle with appropriate value of $\sigma$), it is possible to generically obtain an estimator $\hat{x}_\lambda$ that is unbiased for $x_\lambda$ via multilevel Monte Carlo [see 4, 11], thereby obtaining $g = \lambda(y - \hat{x}_\lambda)$ satisfying the second part of (25) as well as the variance bound in the third part of (25).

Algorithm 4 uses either the generalized MS oracle (Definition 3) or the stochastic proximal oracle (Definition 4). The differences between it and Algorithm 1 are highlighted in blue. There are two differences addition to the obvious one in the oracle interface (which now returns an additional vector $g_{t+1}$). First, we use the vector $g_{t+1}$ to update $v_t$ using a projected mirror descent step (here $\operatorname{Proj}_{\mathcal{X}}$ denotes the Euclidean projection unto $\mathcal{X}$). Second, we allow the algorithm to replace the point $\bar{x}_t$ output from the oracle with any other point $\tilde{x}_t$ that has a lower function value. We note that such option exists also in the original proposal by Monteiro and Svaiter [32] and is independent of the other generalizations studied in this section.

**Theorem 6.** *Let $f : \mathcal{X} \to \mathbb{R}$ be convex and differentiable, let $\mathcal{X}$ be closed and convex, and consider Algorithm 1 with parameters $\alpha > 1$, $\lambda' > 0$, and a $\sigma$-Generalized MS oracle (Definition 3) or $\sigma$-stochastic proximal oracle (Definition 4) for $f$ with $\sigma \in [0, 0.99)$. Let $p \geq 1$ and $c > 0$, and suppose that for all $t$ such that $\lambda_t > \lambda'_t$ or $t = 1$, the iterates $(\bar{x}_t, y_{t-1}, \lambda_t)$ satisfy a $(s, c)$-movement bound (Definition 2) with probability 1. There exist $C_{\alpha,s} = O\left( \frac{s}{\min\{s, \ln \alpha\}} \alpha^{\frac{s+1}{3s+1}} \right)$ and $K_\alpha = O\left( \frac{1}{\ln \alpha} \alpha^{1/3} \right)$ such that the following holds. Let $x_\star \in \mathcal{X}$; if $\mathcal{O}$ is a stochastic proximal oracle then let $x_\star$ be a minimizer of $f$. For any $\epsilon > 0$, when*

$$T \geq \begin{cases} C_{\alpha,s} \left( \frac{c^s \| x_0 - x_\star \|^{s+1}}{\epsilon} \right)^{\frac{2}{3s+1}} & s < \infty \\ K_\alpha (c \| x_0 - x_\star \|)^{\frac{2}{3}} \log \frac{\lambda_1 \| x_0 - x_\star \|^2}{\epsilon} & s = \infty, \end{cases}$$

---

**Algorithm 4:** Generalized Optimal MS Acceleration

**Input:** Initial $x_0$, generalized oracle $\mathcal{O}$

**Parameters:** Initial $\lambda'_0$, multiplicative adjustment factor $\alpha > 1$

1  Set $v_0 = x_0$, $A_0 = 0$

2  $\bar{x}_1, g_1, \lambda_1 = \mathcal{O}(x_0; \lambda'_0)$ , $\lambda'_1 = \lambda_1$

3  **for** $t = 0, 1, \ldots,$ **do**

4      $a'_{t+1} = \frac{1}{2\lambda'_{t+1}}\left(1 + \sqrt{1 + 4\lambda'_{t+1}A_t}\right)$

5      $A'_{t+1} = A_t + a'_{t+1}$

6      $y_t = \frac{A_t}{A'_{t+1}}x_t + \frac{a'_{t+1}}{A'_{t+1}}v_t$

7      **if** $t > 0$ **then** $\bar{x}_{t+1}, g_{t+1}, \lambda_{t+1} = \mathcal{O}(y_t; \lambda'_{t+1})$

8      Let $\tilde{x}_{t+1} \in \mathcal{X}$ satisfy $f(\tilde{x}_{t+1}) \leq f(\bar{x}_{t+1})$

9      **if** $\lambda_{t+1} \leq \lambda'_{t+1}$ **then**

10          $a_{t+1} = a'_{t+1}$, $A_{t+1} = A_t + a_t$

11          $x_{t+1} = \tilde{x}_{t+1}$

12          $\lambda'_{t+2} = \frac{1}{\alpha}\lambda'_{t+1}$

13      **else**

14          $\gamma_{t+1} = \frac{\lambda'_{t+1}}{\lambda_{t+1}}$

15          $a_{t+1} = \gamma_{t+1}a'_{t+1}$, $A_{t+1} = A_t + a_t$

16          $x_{t+1} = \frac{(1-\gamma_{t+1})A_t}{A_{t+1}}x_t + \frac{\gamma_{t+1}A'_{t+1}}{A_{t+1}}\tilde{x}_{t+1}$

17          $\lambda'_{t+2} = \alpha\lambda'_{t+1}$

18      $v_{t+1} = \text{argmin}_{v \in \mathcal{X}}\left\{\langle g_{t+1}, v\rangle + \frac{1}{2a_{t+1}}\|v - v_t\|^2\right\} = \text{Proj}_{\mathcal{X}}(v_t - a_{t+1}g_{t+1})$

---

*we have $f(x_T) - f(x_\star) \leq \epsilon$ with probability at least $2/3$.*

Before providing the proof, we make three more remarks. First, the success probability is relevant only for the stochastic proximal oracle; the bound for the generalized MS oracle holds with probability 1. Second, the need to assume that $x_\star$ is a minimizer of $f$ is also due to a technical issue with the analysis of the stochastic proximal oracle, pointed out in the proof below. Finally, we note that for stochastic proximal oracle we may require the movement bounds to hold on either $(\tilde{x}_t, y_{t-1}, \lambda_t)$ as stated in the theorem, or on $(x_t^\star, y_{t-1}, \lambda_t)$, where $x_t^\star$ is the exact $\lambda_t$ proximal point of $y_{t-1}$.

We recommend reading the proof of Theorem 1 (in Appendix B) before reading the following proof.

*Proof of Theorem 6.* The proof consists of showing that a version of Proposition 1 holds under the conditions of Theorem 6; from there on the arguments on the analysis of the growth rate of $A_T$ is identical. For generalized MS oracles, the proof of Proposition 1 goes through unchanged, except for $\bar{x}_t$ replacing $\tilde{x}_t$, the subgradient $g_t \in \partial f(\bar{x}_t)$ replacing $\nabla f(\tilde{x}_t)$, and using $f(\tilde{x}_t) \leq f(\bar{x}_t)$ to show that $A_{t+1}E_{t+1} \leq (1 - \gamma_{t+1})A_t E_t + \gamma_{t+1}A'_{t+1}[f(\bar{x}_{t+1}) - f(x_\star)]$.

Next, we consider stochastic proximal oracles and adapt [4, Lemma 5], which considers a very similar oracle, to account for our momentum damping scheme (lines 14 to 17). Beginning with some notation, we define the filtration

$$\mathcal{F}_t := \sigma(\lambda_1, \bar{x}_1, g_1, \ldots, \lambda_t, \bar{x}_t, g_t, \lambda_{t+1})$$

so that $a_{t+1}, a'_{t+1}, A_{t+1} \in \mathcal{F}_t$. In addition, we let

$$\hat{x}_t = \underset{x \in \mathcal{X}}{\text{argmin}}\left\{f(x) + \frac{\lambda_t}{2}\|x - y_{t-1}\|^2\right\}$$

and note that $\hat{x}_{t+1} \in \mathcal{F}_t$ and moreover that

$$\hat{g}_{t+1} := \lambda_{t+1}(y_t - \hat{x}_{t+1}) = \mathbb{E}[g_{t+1} \mid \mathcal{F}_t]$$

by the second part of (25). We also define $E_t := f(x_t) - f(x_\star)$, $D_t := \frac{1}{2}\|v_t - x_\star\|^2$, and $M_{t+1} := \frac{1}{2}\|\bar{x}_{t+1} - y_t\|^2$ as in Proposition 1 (except with $\bar{x}_t$ instead of $\tilde{x}_t$).

The update formula for $v_t$ gives us

$$D_{t+1} = \frac{1}{2}\|\mathsf{Proj}_{\mathcal{X}}(v_t - a_{t+1}g_{t+1}) - x_\star\|^2$$

$$\leq \frac{1}{2}\|(v_t - a_{t+1}g_{t+1}) - x_\star\|^2 = D_t + a_{t+1}\langle g_{t+1}, x_\star - v_t\rangle + \frac{a_{t+1}^2}{2}\|g_{t+1}\|^2.$$

Rearranging and taking expectation, we have

$$a_{t+1}\langle \hat{g}_{t+1}, v_t - x_\star\rangle = \mathbb{E}[a_{t+1}\langle g_{t+1}, v_t - x_\star\rangle \mid \mathcal{F}_t]$$

$$\leq D_t - \mathbb{E}[D_{t+1} \mid \mathcal{F}_t] + \frac{a_{t+1}^2}{2}\mathbb{E}[\|g_{t+1}\|^2 \mid \mathcal{F}_t]. \tag{26}$$

Moreover, by the second and third parts of (25),

$$\mathbb{E}[\|g_{t+1}\|^2 \mid \mathcal{F}_t] = \|\mathbb{E}[g_{t+1} \mid \mathcal{F}_t]\|^2 + \mathrm{Var}[g_{t+1} \mid \mathcal{F}_t] \leq \|\hat{g}_{t+1}\|^2 + \sigma^2\lambda_{t+1}^2\mathbb{E}[M_{t+1} \mid \mathcal{F}_t]. \tag{27}$$

By the definition $y_t$ and $A'_{t+1} = A_t + a'_{t+1}$, we have

$$a'_{t+1}v_t = A'_{t+1}y_t - A_t x_t = a'_{t+1}\hat{x}_{t+1} + A'_{t+1}(y_t - \hat{x}_{t+1}) - A_t(x_t - \hat{x}_{t+1}).$$

Therefore,

$$a'_{t+1}\langle \hat{g}_{t+1}, x_\star - v_t\rangle$$

$$= \langle \hat{g}_{t+1}, a'_{t+1}(x_\star - \hat{x}_{t+1}) + A'_{t+1}(\hat{x}_{t+1} - y_t) + A_t(x_t - \hat{x}_{t+1})\rangle$$

$$\overset{(i)}{\leq} a'_{t+1}[f(x_\star) - f(\hat{x}_{t+1})] + A'_{t+1}\langle \hat{g}_{t+1}, \hat{x}_{t+1} - y_t\rangle + A_t[f(x_t) - f(\hat{x}_{t+1})]$$

$$\overset{(ii)}{=} A_t E_t - A'_{t+1}[f(\hat{x}_{t+1}) - f(x_\star)] - \frac{A'_{t+1}}{\lambda_{t+1}}\|\hat{g}_{t+1}\|^2.$$

where we used $(i)$ the fact that $\hat{g}_{t+1} \in \partial f(\hat{x}_{t+1})$ and $(ii)$ that $A'_{t+1} = A_t + a'_{t+1}$ and $\hat{x}_{t+1} - y_t = -\hat{g}_{t+1}/\lambda_{t+1}$. To connect $f(\hat{x}_{t+1})$ to $f(\bar{x}_{t+1})$, we use the first part of (25), which gives

$$\mathbb{E}[f(\bar{x}_{t+1}) + \lambda_{t+1}M_{t+1} \mid \mathcal{F}_t] \leq f(\hat{x}_{t+1}) + \frac{\lambda_{t+1}}{2}\|\hat{x}_{t+1} - y_t\|^2 + \frac{\sigma^2}{2}\lambda_{t+1}\mathbb{E}[M_{t+1} \mid \mathcal{F}_t].$$

Substituting back and recalling that $\|\hat{x}_{t+1} - y_t\| = \|\hat{g}_{t+1}\|/\lambda_{t+1}$ and $f(\tilde{x}_{t+1}) \leq f(\bar{x}_{t+1})$ gives

$$A'_{t+1}\mathbb{E}[f(\tilde{x}_{t+1}) - f(x_\star) \mid \mathcal{F}_t] \leq$$

$$A_t E_t + a'_{t+1}\langle \hat{g}_{t+1}, v_t - x_\star\rangle - \left(1 - \frac{\sigma^2}{2}\right)A'_{t+1}\lambda_{t+1}\mathbb{E}[M_{t+1} \mid \mathcal{F}_t] - \frac{A'_{t+1}}{2\lambda_{t+1}}\|\hat{g}_{t+1}\|^2. \tag{28}$$

Let $\gamma_{t+1} := \min\left\{1, \frac{\lambda'_{t+1}}{\lambda_{t+1}}\right\}$ and note that this definition is consistent with $\gamma_{t+1}$ as defined in the algorithm and that, for any value of $\lambda'_{t+1}/\lambda_{t+1}$ we have $a_{t+1} = \gamma_{t+1}a'_{t+1}$, $A_{t+1} = (1 - \gamma_{t+1})A_t + \gamma_{t+1}A'_{t+1}$ and $x_{t+1} = \frac{(1-\gamma_{t+1})A_t}{A_{t+1}}x_t + \frac{\gamma_{t+1}A'_{t+1}}{A_{t+1}}\tilde{x}_{t+1}$. Therefore, by convexity,

$$f(x_{t+1}) \leq \frac{(1 - \gamma_{t+1})A_t}{A_{t+1}}f(x_t) + \frac{\gamma_{t+1}A'_{t+1}}{A_{t+1}}f(\tilde{x}_{t+1}).$$

Subtracting $f(x_\star)$, multiplying by $A_{t+1}$ and taking expectation, we have

$$\mathbb{E}[A_{t+1}E_{t+1} \mid \mathcal{F}_t] \leq (1 - \gamma_{t+1})E_t + \gamma_{t+1}A'_{t+1}\mathbb{E}[f(\tilde{x}_{t+1}) - f(x_\star) \mid \mathcal{F}_t]$$

$$\leq A_t E_t + a_{t+1}\langle \hat{g}_{t+1}, v_t - x_\star\rangle - \left(1 - \frac{\sigma^2}{2}\right)A'_{t+1}\gamma_{t+1}\lambda_{t+1}\mathbb{E}[M_{t+1} \mid \mathcal{F}_t] - \frac{\gamma_{t+1}A'_{t+1}}{2\lambda_{t+1}}\|\hat{g}_{t+1}\|^2. \tag{29}$$

where in the second inequality we substituted (28). Note that

$$a_{t+1}^2 = \gamma_{t+1}^2(a'_{t+1})^2 = \frac{\gamma_{t+1}^2 A'_{t+1}}{\lambda'_{t+1}} \leq \frac{\gamma_{t+1}A'_{t+1}}{\lambda_{t+1}}.$$

Substituting back into (26) and combining with (27) gives

$$a_{t+1} \langle \hat{g}_{t+1}, v_t - x_\star \rangle \le D_t - \mathbb{E}[D_{t+1} \mid \mathcal{F}_t] + \frac{\sigma^2 A'_{t+1} \gamma_{t+1} \lambda_{t+1}}{2} \mathbb{E}[M_{t+1} \mid \mathcal{F}_t] + \frac{\gamma_{t+1} A'_{t+1}}{2\lambda_{t+1}} \|\hat{g}_{t+1}\|^2.$$

Plugging the above bound on $a_{t+1} \langle \hat{g}_{t+1}, v_t - x_\star \rangle$ into (29), noting that $\gamma_{t+1}\lambda_{t+1} = \min\{\lambda'_{t+1}, \lambda_{t+1}\}$, and rearranging, we obtain

$$\mathbb{E}\big[A_{t+1}E_{t+1} + D_{t+1} + (1 - \sigma^2)A'_{t+1}\min\{\lambda'_{t+1}, \lambda_{t+1}\}M_{t+1} \mid \mathcal{F}_t\big] \le A_t E_t + D_t.$$

Iterating this bound and noting that $A_t \le A'_t$ for all $t$, we obtain

$$\mathbb{E}\left[A_T E_T + D_T + (1 - \sigma^2) \sum_{t \in \mathcal{S}_T^{>}} A_t \lambda'_t M_t\right] \le A_0 E_0 + D_0.$$

By our assumption that $x_\star$ is a minimizer of $f$, we have that $E_T$ is a nonegative random variable, and consequently the above display is a bound on the expectation of a nonnegative random variable. (This is the reason we require $x_\star$ to be a minimizer of $f$). Therefore, by Markov's inequality, the event

$$A_T E_T + D_T + (1 - \sigma^2) \sum_{t \in \mathcal{S}_T^{>}} A_t \lambda'_t M_t \le 3(A_0 E_0 + D_0)$$

holds with probability at least 2/3, implying (13), except with $A_0 E_0 + D_0$ multiplied by a factor of 3. Moreover, the growth bounds $\sqrt{A_T} \ge \frac{1}{2} \sum_{t \in \mathcal{S}_T^{\le}} 1/\sqrt{\lambda'_t}$ holds deterministically as a consequence of the update rule for $A_t$. These two facts suffice to establish the growth rate of $A_T$ precisely as we do in the proof of Theorem 1, thereby obtaining the same rate of convergence (up to a constant). □

## D   Proofs for Section 3

This section contains the analysis of our adaptive oracle implementations (Algorithms 2 and 3). We begin by quickly showing how an idealized regularized Newton step adaptively yields optimal movement bounds without need to know the degree or order of the Hessian Hölder continuity (Appendix D.1). Then, we prove movement bound and complexity guarantees for our second-order and first-order adaptive oracle implementations in Appendices D.2 and D.3, respectively. Finally, we provide auxiliary results used throughout the preceding proofs (Appendix D.4).

### D.1   A movement bound for the ideal Newton step

The following proposition shows how choosing the smallest $\lambda$ for which a $\lambda$-regularized Newton step satisfies the MS condition yields movement bounds adaptive to Hessian Hölder continuity.

**Proposition 2.** *For any $y \in \mathbb{R}^d$ and $\sigma \in (0, 1)$, let $\lambda^\star$ be the smallest $\lambda$ for which the $\lambda$-regularized Newton step $x_\lambda = y - [\nabla^2 f(y) + \lambda^\star I]^{-1}\nabla f(y)$ satisfies the MS condition $\|x_\lambda - (y - \frac{1}{\lambda}\nabla f(x))\| \le \sigma\|x_\lambda - y\|$. If $\nabla^2 f$ is $(H, \nu)$-Hölder continuous for any $\nu \in [0, 1]$, the triplet $(x_{\lambda^\star}, y, \lambda^\star)$ satisfies a $\left(1 + \nu, \left(\frac{H}{2\sigma}\right)^{1/(1+\nu)}\right)$-movement bound.*

*Proof.* Let

$$\tilde{x} = \operatorname*{argmin}_{x' \in \mathbb{R}^d}\left\{\tilde{f}_2(x'; y) + \frac{H}{2(2+\nu)\sigma}\|x' - y\|^{2+\nu}\right\} \quad \text{and} \quad \tilde{\lambda} = \frac{H}{2\sigma}\|x - y\|^\nu.$$

That is $(\tilde{x}, \tilde{\lambda}) = \mathcal{O}_{2, \nu\text{-reg}}(y)$ with parameter $M = H/\sigma$. Note that (a) $\tilde{x}, y$ and $\tilde{\lambda}$ satisfy the MS condition as explained in Section 3.1, and (b) $\tilde{x} = y - [\nabla^2 f(y) + \tilde{\lambda}I]^{-1}\nabla f(y)$. Therefore the minimal $\lambda^\star$ satisfying the MS condition must satisfy $\lambda^\star \le \tilde{\lambda}$ and

$$\|x^\star - y\| \overset{(\star)}{\ge} \|\tilde{x} - y\| = \left(\frac{2\sigma\tilde{\lambda}}{H}\right)^{1/\nu} \ge \left(\frac{2\sigma\lambda^\star}{H}\right)^{1/\nu},$$

where $(\star)$ is due to auxiliary Lemma 7. This yields the movement bound. □

We remark that the same proof and movement bound hold for a slightly more relaxed notion of $\lambda^\star$, namely the smallest for which the $x_\lambda$ satisfies the MS condition for all $\lambda \geq \lambda^*$. This is the actual notion of $\lambda^\star$ that we approximate in the next subsections, where we find $\lambda$ such that $x_\lambda$ satisfies the MS condition, but $x_{\lambda/2}$ does not satisfy it (and therefore $\lambda \geq \lambda^\star/2$).

### D.2 Analysis of Algorithm 2

**Theorem 2.** *Algorithm 2 with parameter $\sigma$ is a $\sigma$-MS oracle $\mathcal{O}_{\mathsf{aMSN}}$. For any $y \in \mathbb{R}^d$, computing $(x, \lambda) = \mathcal{O}_{\mathsf{aMSN}}(y)$ requires at most $2 + 2\log_2\big(1 + \big|\log_2 \frac{\lambda}{\lambda'}\big|\big)$ linear systems solutions. If LAZY is False or $\lambda > \lambda'$, and if $\nabla^2 f$ is $(H, \nu)$-Hölder in a ball of radius $2\|x - y\|$ around $y$, then $(x, y, \lambda)$ satisfy a $\big(1 + \nu, (2H/\sigma)^{1/(1+\nu)}/\sigma\big)$-movement bound.*

*Proof.* First, note that, by construction, the algorithm outputs a value of $\lambda$ for which CHECKMS$(\lambda; y, \sigma)$ evaluates to True, and is therefore a $\sigma$-MS oracle as per Definition 1.

Next, let us bound the total number of linear system solutions in the algorithm, noting it is equal to the number of calls to CHECKMS. The algorithm solves 1 linear system in line 1, and then solves $k^\star + 1$ linear systems in either the while-loop in line 5 or the while-loop in line 11. The algorithm then arrives at the while-loop in line 15 with two values $\lambda_{\mathrm{vld}}$ and $\lambda_{\mathrm{invld}}$ such that (a) CHECKMS$(\lambda_{\mathrm{vld}}; y, \sigma)$ is True and CHECKMS$(\lambda_{\mathrm{invld}}; y, \sigma)$ is False and (b) $\lambda_{\mathrm{vld}}/\lambda_{\mathrm{invld}} = 2^{2^{k^\star}}$. The while-loop maintains the invariant (a) while transforming $\lambda_{\mathrm{vld}}/\lambda_{\mathrm{invld}} \to \sqrt{\lambda_{\mathrm{vld}}/\lambda_{\mathrm{invld}}}$ at each iteration. After $j$ iterations of the while-loop, we have

$$\frac{\lambda_{\mathrm{vld}}}{\lambda_{\mathrm{invld}}} = \left(2^{2^{k^\star}}\right)^{2^{-j}} = 2^{2^{k^\star - j}}.$$

Therefore, after precisely $k^\star$ iterations we obtain $\lambda_{\mathrm{vld}} = 2\lambda_{\mathrm{invld}}$ and the loop terminates. Hence, the overall number of linear system solutions is $1 + (k^\star + 1) + k^\star = 2 + 2k^\star$, or just 1 in case the first CHECKMS is True and the algorithm is lazy.

To bound $k^\star$ in terms of the input $\lambda'$ and output $\lambda$, consider the values of $\lambda_{\mathrm{vld}}$ and $\lambda_{\mathrm{invld}}$ before entering the while-loop at line 15. First, note that $\lambda_{\mathrm{invld}} \leq \lambda \leq \lambda_{\mathrm{vld}}$. Second, assuming that $\lambda > \lambda'$ (i.e., the first CHECKMS fails) we have

$$\lambda_{\mathrm{vld}} = \lambda' \prod_{j=0}^{k^\star} 2^{2^j} = \lambda' \cdot 2^{\left(2^{k^\star+1}-1\right)}.$$

Combining these two facts, we have

$$2^{\left(2^{k^\star+1}-1\right)} \frac{\lambda'}{\lambda} \leq \frac{\lambda_{\mathrm{vld}}}{\lambda_{\mathrm{invld}}} = 2^{2^{k^\star}}.$$

Rearranging this inequality yields $k^\star \leq \log_2\big(1 + \log_2 \frac{\lambda}{\lambda'}\big)$ as claimed. The bound for $\lambda < \lambda'$ follows analogously.

We now turn to showing the movement bound assuming $\nabla^2 f$ is $(H, \nu)$-Hölder for $\nu \in [0, 1]$ within a ball of radius $2\|x - y\|$ around $y$, where $x = y - (\nabla^2 f(y) + \lambda I)^{-1}\nabla f(y)$ is the output of the algorithm. Let $\lambda_{1/2} = \lambda/2$ and $x_{1/2} = y - (\nabla^2 f(y) + \lambda_{1/2} I)^{-1}\nabla f(y)$. Let $\tilde{\lambda}$ and $\tilde{x} = y - (\nabla^2 f(y) + \tilde{\lambda} I)^{-1}\nabla f(y)$ satisfy $\tilde{\lambda} = \frac{H}{2\sigma}\|\tilde{x} - y\|^\nu$, i.e.,

$$\tilde{x} = \underset{x \in \mathbb{R}^d}{\mathrm{argmin}}\left\{\langle \nabla f(y), x - y\rangle + \frac{1}{2}\langle x - y, \nabla^2 f(y)(x - y)\rangle + \frac{H}{2(2 + \nu)\sigma}\|x - y\|^{2+\nu}\right\}.$$

It is immediate to see that for any $\lambda > \tilde{\lambda}$, CHECKMS$(\lambda; y, \sigma)$ must evaluate to True and thus the while loop in Line 11 is guaranteed to terminate.

We will now argue that $\lambda_{1/2} < \tilde{\lambda}$. First, note that $\lambda_{1/2}$ is the last value of $\lambda_{\mathrm{invld}}$ before the while loop at line 15 terminates. Therefore, CHECKMS$(\lambda_{1/2}; y, \sigma)$ must evaluate to False. Further, note that, by Lemma 7,

$$\|x_{1/2} - y\| \leq \frac{\lambda}{\lambda_{1/2}}\|x - y\| = 2\|x - y\|.$$

Using the local Hessian assumption of Hölder continuity along with $(\nabla^2 f(y) + \lambda_{1/2}I)x_{1/2} + \nabla f(y) = 0$, auxiliary Lemma 8 gives

$$\left\| x_{1/2} - \left( y - \frac{1}{\lambda_{1/2}} \nabla f(x_{1/2}) \right) \right\| \leq \frac{H}{2\lambda_{1/2}} \|x_{1/2} - y\|^{1+\nu} = \frac{\sigma\tilde{\lambda}}{\lambda_{1/2}\|\tilde{x} - y\|^\nu} \|x_{1/2} - y\|^{1+\nu},$$

with the final equality using the definition of $\tilde{\lambda}$. Assuming by contradiction that $\lambda_{1/2} \geq \tilde{\lambda}$, we have that

$$\left\| x_{1/2} - \left( y - \frac{1}{\lambda_{1/2}} \nabla f(x_{1/2}) \right) \right\| \leq \frac{\sigma\|x_{1/2} - y\|^{1+\nu}}{\|\tilde{x} - y\|^\nu} \leq \sigma\|x_{1/2} - y\|,$$

where the final inequality used Lemma 7 combined with $\lambda_{1/2} \geq \tilde{\lambda}$ to deduce that $\|x_{1/2} - y\| \leq \|\tilde{x} - y\|$. This implies that $\textsc{CheckMS}(\lambda_{1/2}; y, \sigma)$ is True, giving a contradiction. The $(1 + \nu, (2H/\sigma)^{1/(1+\nu)})$-movement bound follows from

$$\lambda = 2\lambda_{1/2} < 2\tilde{\lambda} = \frac{H\|\tilde{x} - y\|^\nu}{\sigma} \leq \frac{2^\nu H\|x - y\|^\nu}{\sigma^1},$$

with the final inequality using Lemma 7 combined with $\lambda/\tilde{\lambda} \leq 2$, which implies $\|\tilde{x} - y\| \leq 2\|x - y\|$. $\qquad\square$

We remark that the termination of the while loop in line 1 (when LAZY is False) is, strictly speaking, not guaranteed. For example, if the function $f$ is quadratic, CheckMS will always evaluate to True. However, it is also straightforward to verify that as long as CheckMS is True for a given $\lambda$, the corresponding regularized Newton step $x = y - (\nabla^2 f(y) + \lambda I)^{-1}\nabla f(y)$ has optimality gap bounded by $\lambda\|x - y\|^2$. Therefore, since the loop in line 1 decreases $\lambda$ at a double-exponential rate, if it fails to terminate after a small number of iterations then it means we have found an essentially optimal point. Put differently, if we seek an $\epsilon$ suboptimal point, we may stop the loop in line 1 after $O(\log\log(\frac{\lambda'_0 R^2}{\epsilon}))$ iterations. We account for this possibility in the complexity bound below.

**Corollary 3.** *Consider Algorithm 1 with initial point $x_0$, parameters $\alpha$ satisfying $1.1 \leq \alpha = O(1)$ and $\lambda'_0$, and $\sigma$-MS oracle $\mathcal{O}_{\mathsf{aMSN}}$ (with LAZY = True in all but the first iteration) with $\sigma \in (0.01, 0.99)$. For any $H, \epsilon > 0$, $\nu \in [0, 1]$ and any $x_\star \in \mathbb{R}^d$, if $f$ is convex with $(H, \nu)$-Hölder Hessian, the algorithm produces an iterate $x_T$ such that $f(x_T) \leq f(x_\star) + \epsilon$ using $T = O\left( \left( H\|x_0 - x_\star\|^{2+\nu}/\epsilon \right)^{2/(4+3\nu)} \right)$ Hessian evaluations and $O\left( T + \log\log\max\left\{ \frac{HR^\nu}{\lambda'_0}, \frac{\lambda'_0 R^2}{\epsilon} \right\} \right)$ linear system solutions, where $R$ is the distance between $x_0$ and $\arg\min_{x'} f(x')$.*

*Proof.* Throughout the proof, we let $T$ denote the index of the first iteration of Algorithm 1 for which $f(x_T) \leq f(x_\star) + \epsilon$. The bound on Hessian evaluation complexity follows immediately from the validity of the MS-approximate proximal oracle and movement bounds guaranteed in Theorem 2, and the iteration bound given by Theorem 1, noting that each call to Algorithm 2 requires only one Hessian computation.

To bound the total number of linear system solutions, we consider separately $(i)$ the iteration where $\lambda_t = \lambda'_t$, $(ii)$ the iterations $t \in [2, T-1]$ where $\lambda_t > \lambda'_t$ (i.e, those in $\mathcal{S}_T^>$), $(iii)$ the first iteration (note that $\lambda_{t+1} < \lambda'_{t+1}$ can only happen in the first iteration since we are using a lazy oracle), and $(iv)$ the last iteration.

Case $(i)$ is easy, because at iterations where $\lambda_{t+1} = \lambda'_{t+1}$ Algorithm 2 requires at most 2 linear system solves by Theorem 2, and thereofore all such iterations combined require $N_{(i)} = O(T) = O\left( \left( \frac{H\|x_0 - x_\star\|^{2+\nu}}{\epsilon} \right)^{2/(4+3\nu)} \right)$ linear system solves.

To handle case $(ii)$, which requires the most work, we use Theorem 2 to bound the total number of linear system solves contributed by these iterates by

$$N_{(ii)} \overset{(a)}{=} O\left(\sum_{t \in \mathcal{S}_{T-1}^{>}} \log_2\left(1 + \log_2 \frac{\lambda_t}{\lambda_t'}\right)\right) \overset{(b)}{=} O\left(\sum_{t \in \mathcal{S}_{T-1}^{>}} \log_2\left(1 + \log_2 \frac{H\|\tilde{x}_t - y_{t-1}\|^\nu}{\lambda_t'}\right)\right)$$

$$\overset{(c)}{=} O\left(\sum_{t \in \mathcal{S}_{T-1}^{>}} \left(H\frac{\|\tilde{x}_t - y_{t-1}\|^\nu}{\lambda_t'}\right)^{2/(4+3\nu)}\right) \overset{(d)}{=} O\left((H\|x_0 - x_\star\|^\nu A_{T-1})^{2/(4+3\nu)}\right),$$

which follows from $(a)$ the complexity bound in Theorem 2; $(b)$ the fact that a $(1+\nu, O(H^{1/(1+\nu)}))$-movement bound holds for every $t \in \mathcal{S}_T^{>}$, meaning that $\lambda = O(H)\|\tilde{x}_t - y_t\|^\nu$; $(c)$ by the inequalities

$$\log_2\left(1 + \log_2 z\right) \leq \log_2 z = c \log_2 z^{1/c} = O(z^{1/c})$$

for any $z \geq 1$ and fixed $c \geq 0$; and $(d)$ Lemma 9, using the assumption that $f(x_{T-1}) > f(x_\star)$. Moreover, note that

$$\epsilon < f(x_{T-1}) - f(x_\star) \leq \frac{\|x_0 - x_\star\|^2}{2A_{T-1}}$$

by eq. (9) in Proposition 1, which implies $A_{T-1} = O(\|x_0 - x_\star\|^2/\epsilon)$. Substituting back into the bound on $N_{(ii)}$, we obtain

$$N_{(ii)} = O\left((H\|x_0 - x_\star\|^\nu A_{T-1})^{2/(4+3\nu)}\right) = O\left(\left(\frac{H\|x_0 - x_\star\|^{2+\nu}}{\epsilon}\right)^{2/(4+3\nu)}\right).$$

Therefore, perhaps surprisingly, the worst-case double-logarithmic per-iteration linear system complexity amortizes to a constant.

To handle the last two edge cases, let $z_\star$ be the minimizer of $f$ closest to $x_0$, such that $\|x_0 - z_\star\| = R$. In case $(iii)$, i.e., the number of linear system solves in the first iteration. We consider separately the cases $\lambda_1 \geq \lambda_0'$ and $\lambda_1 < \lambda_0'$. In the former case, the movement bound guaranteed at the first iteration yields (noting that $y_0 = x_0$)

$$\frac{\lambda_1}{\lambda_0'} = O\left(\frac{H\|x_1 - x_0\|^\nu}{\lambda_0'}\right) = O\left(\frac{HR^\nu}{\lambda_0'}\right),$$

where the last transition follows from Lemma 5 and the assumption $f(x_1) \geq f(z_\star)$. In the latter case $(\lambda_1 < \lambda_0')$, Lemma 5 and $f(x_1) - f(z_\star) \geq f(x_1) - f(x_\star) \geq \epsilon$ gives

$$\frac{\lambda_0'}{\lambda_1} = O\left(\frac{\lambda_0' R^2}{f(x_1) - f(z_\star)}\right) = O\left(\frac{\lambda_0' R^2}{\epsilon}\right).$$

Therefore, the number of linear system solutions at the first iteration is

$$O\left(\log\left|\log\frac{\lambda_1}{\lambda_0'}\right|\right) = O\left(\log\log\max\left\{\frac{HR^\nu}{\lambda_0'}, \frac{\lambda_0' R^2}{\epsilon}\right\}\right).$$

Finally, we consider $(iv)$ the last iteration $t = T$. If $T \notin \mathcal{S}_T^{>}$ then there is nothing to consider, since it only contributes a single linear system solutions. If $T \in \mathcal{S}_T^{>}$, however, we cannot treat it as in case $(ii)$, since we are not guaranteed that $f(x_T) \geq f(x_\star)$. However, we *are* guaranteed that $f(x_T) \geq f(z_\star)$. Therefore, Lemma 9 allows us to conclude that

$$\frac{\lambda_T}{\lambda_T'} = O\left(\frac{H\|\tilde{x}_T - y_{T-1}\|^\nu}{\lambda_{T-1}'}\right) = O\left(\left[\sum_{t \in \mathcal{S}_T^{>}} \left(H\frac{\|\tilde{x}_t - y_{t-1}\|^\nu}{\lambda_t'}\right)^{\frac{2}{4+3\nu}}\right]^{\frac{4+3\nu}{2}}\right)$$

$$= O\left(HR^\nu A_T\right).$$

Noting that $A_T = O(A_{T-1})$ (see Lemma 10) and $A_{T-1} = O(R^2/\epsilon)$ (since $\epsilon < f(x_{T-1}) - f(x_\star) \leq f(x_{T-1}) - f(z_\star) \leq \frac{R^2}{2A_{T-1}}$) we conclude that $\frac{\lambda_T}{\lambda_T'} = O\left(\frac{HR^{2+\nu}}{\epsilon}\right)$. Since $\frac{HR^{2+\nu}}{\epsilon} = \frac{HR^\nu}{\lambda_0'} \cdot \frac{\lambda_0' R^2}{\epsilon} \leq \max\left\{\frac{HR^\nu}{\lambda_0'}, \frac{\lambda_0' R^2}{\epsilon}\right\}^2$, we conclude that the $O\left(\log\log\frac{\lambda_T}{\lambda_T'}\right)$ contribution of case $(iv)$ to the total number of linear systems is no greater than our bound for case $(iii)$. $\qquad\square$

We remark that departing from a normal bisection or doubling scheme we have used a "double-logarithmic scale" in the while loops starting at Line 5 and Line 11 in Algorithm 2. As shown in the proof above, this doesn't affect the complexity bounds shown for case $(ii)$, but gives better complexity bounds in the analysis of case $(iii)$ and $(iv)$. Eventually this allows us to only have an additive double-logarithmic term in the final complexity of linear system solves as stated in Corollary 3.

### D.3 Analysis of Algorithm 3

**Theorem 4.** *Algorithm 3 with parameter $\sigma$ is a $\sigma$-MS oracle $\mathcal{O}_{\mathsf{aMSN\text{-}fo}}$. For any $y \in \mathbb{R}^d$, computing $(x, \lambda) = \mathcal{O}_{\mathsf{aMSN\text{-}fo}}(y)$ requires at most $O\left(\sqrt{1 + \frac{\|\nabla^2 f(y)\|_{\mathrm{op}}}{\sigma \min\{\lambda, \lambda'\}}}\right)$ Hessian-vector product and $O\left(\left|\log \frac{\lambda}{\lambda'}\right|\right)$ gradient computations. If LAZY is False or $\lambda > \lambda'$, and if $\nabla^2 f$ is $(H, \nu)$-Hölder, then $(x, y, \lambda)$ satisfy a $\left(1 + \nu, (6H/\sigma)^{1/(1+\nu)}\right)$-movement bound.*

*Proof.* Clearly, Algorithm 3 can only return a pair $x, \lambda$ satisfying Definition 1. (At this point, we are not guaranteed that the algorithm ever returns. However, below we prove that the check in Line 5 succeeds for finite $\lambda$ as long as the Hessian is continuous).

To analyze the complexity of the algorithm, we begin by noting that it approximates a Newton step with MinRes at most $\left|\log_2 \frac{\lambda}{\lambda'}\right| + 1$ times: when $\lambda \geq \lambda'$ we approximate Newton steps for values for regularization parameters of the form $2^k \lambda'$ for $k = 0, 1, \ldots, \log_2 \frac{\lambda}{\lambda'}$; when $\lambda < \lambda'$ we instead consider regularization parameters of the form $2^{-k} \lambda'$ for $k = 0, 1, \ldots, \log_2 \frac{\lambda}{\lambda'} + 1$. These considerations immediately yield our claimed $O\left(\left|\log_2 \frac{\lambda}{\lambda'}\right|\right)$ bound on the number of gradients evaluated by the algorithm.

To bound the total Hessian-vector product complexity, suppose that Algorithm 3 attempts to approximate a Newton step with regularization parameter $\lambda_k$; we argue that the corresponding terminates in $O\left(\sqrt{\frac{\|\nabla^2 f(y)\|_{\mathrm{op}} + \lambda_k}{\lambda_k \sigma}}\right)$ Hessian-vector products. Let $A = \nabla^2 f(y) + \lambda_k I$, $b = -\nabla f(y)$ and $w^\star = A^{-1}b$. Lemma 6 guarantees that $\|r_t\| = O\left(\|A\|_{\mathrm{op}}\|w^\star\|/t^2\right)$. Consequently, after $T_\lambda = O\left(\sqrt{\frac{\|A\|_{\mathrm{op}}}{\lambda_k \sigma}}\right)$ steps we have $\|r_{T_\lambda}\| \leq \frac{\lambda_k \sigma}{4}\|w^\star\|$. Since $h(x) = \frac{1}{2}x^\top A x - b^\top x$ is $\lambda_k$-strongly-convex and $r_i = \nabla h(w_i)$, we have $\lambda_k\|w_{T_\lambda} - w^\star\| \leq \|r_{T_\lambda}\|$, and consequently $\|w_{T_\lambda} - w^\star\| \leq \frac{\sigma}{4}\|w^\star\| \leq \frac{1}{4}\|w^\star\|$. Since $\|w^\star\| - \|w_{T_\lambda}\| \leq \|w_{T_\lambda} - w^\star\|$ by the triangle inequality, we conclude that $\|w^\star\| \leq \frac{4}{3}\|w_{T_\lambda}\|$. Substituting back yields $\|r_{T_\lambda}\| \leq \frac{\lambda_k \sigma}{3}\|w_{T_\lambda}\|$, and consequently the while-loop must terminate in $T_\lambda = O\left(\sqrt{\frac{\|\nabla^2 f(y)\|_{\mathrm{op}} + \lambda_k}{\lambda_k \sigma}}\right)$ steps, with each step corresponding to a single Hessian-vector product.

Next, we argue that for very large $\lambda_k$ we do not need to compute any (new) Hessian-vector product, since the while-loop terminates in one step. More specifically that, $\lambda_k \geq \frac{4\|\nabla^2 f(y)\|_{\mathrm{op}}}{\sigma}$ the while-loop terminates after one step, i.e., $\|r_1\| \leq \frac{\lambda_k \sigma}{2}\|w_1\|$. To see this, first observe that (since $w_1$ has the smallest residual among all vectors $w$ proportional to $b$), we have $\|r_1\| \leq \|Ab/\lambda_k - b\| = \frac{1}{\lambda_k}\|\nabla^2 f(y)b\| \leq \frac{\|\nabla^2 f(y)\|_{\mathrm{op}}}{\lambda_k}\|b\| \leq \frac{\sigma}{4}\|b\|$. Moreover, since $w_1 = \frac{b^\top A b}{b^\top A^2 b}b$, we have $\|w_1\| \geq \frac{1}{\lambda_k + \|\nabla^2 f(y)\|_{\mathrm{op}}}\|b\| \geq \frac{4}{5\lambda_k}\|b\|$. Consequently, $\|r_1\| \leq \frac{5\lambda_k \sigma}{16}\|w_1\|$, meaning that the while-loop terminates after the first iterate. Therefore, for $\lambda_k \geq \frac{4\|\nabla^2 f(y)\|_{\mathrm{op}}}{\sigma}$ no Hessian-vector product computations are necessary (since the ones before the while-loop can be computed once for all $k$).

Let $\lambda_{\min}$ be the smallest value of $\lambda_k$ encountered by the algorithm, and note that $\lambda_{\min} \geq \min\{\lambda/2, \lambda'\}$, and that $\lambda_k = 2^k \lambda_{\min}$ for $k = 0, \ldots, O\left(\left|\log \frac{\lambda}{\lambda'}\right|\right)$. By the discussion above, we require Hessian-vector product computations only at the first $K = O\left(\log \frac{\|\nabla^2 f(y)\|_{\mathrm{op}}}{\sigma \lambda_{\min}}\right)$ iterations. Therefore the total number of Hessian-vector products is $O\left(\sum_{k=0}^{K} \sqrt{1 + \frac{\|\nabla^2 f(y)\|_{\mathrm{op}}}{\sigma 2^k \lambda_{\min}}}\right) = O\left(\sqrt{1 + \frac{\|\nabla^2 f(y)\|_{\mathrm{op}}}{\sigma \min\{\lambda, \lambda'\}}}\right)$, giving the claimed bound on Hessian-vector product count.

Next, we assume that $f$ is has an $(H, \nu)$-Hölder Hessian and argue that the algorithm's output $x, \lambda$ satisfies a movement bound (unless LAZY is True and $\lambda = \lambda'$). To do so, we first establish an upper bound on the returned $\lambda$. Let

$$\tilde{w} = \operatorname*{argmin}_v \left\{ v^\top \nabla f(y) + \frac{1}{2} v^\top \nabla^2 f(y) v + \frac{H}{(2+\nu)} \|v\|^{(2+\nu)\sigma} \right\}$$

and note that $\tilde{w} = -(\nabla^2 f(y) + \tilde{\lambda} I)^{-1} \nabla f(y)$ for $\tilde{\lambda} = \frac{H}{\sigma} \|\tilde{w}\|^\nu$. Let us show that the MS condition check in line 5 must succeed when for $\lambda_k \geq \tilde{\lambda}$. Let $w^\star = -(\nabla^2 f(y) + \lambda_k I)^{-1} \nabla f(y)$ denote the exact regularized Newton step corresponding to $\lambda_k$, let $w_i$ be the output of the corresponding MinRes run, and let

$$r_i = (\nabla^2 f(y) + \lambda_k I) w_i + \nabla f(y)$$

be the corresponding residual. Note that, by Lemma 8 we have

$$\left\| x - \left( y - \frac{1}{\lambda_k} \nabla f(x) \right) \right\| \leq \frac{1}{\lambda_k} \left( \|r_i\| + \frac{H}{2} \|w_i\|^{1+\nu} \right). \tag{30}$$

Moreover

$$\frac{H}{2} \|w_i\|^{1+\nu} = \frac{\sigma \tilde{\lambda}}{2\|\tilde{w}\|^\nu} \|w_i\|^{1+\nu} \leq \frac{\lambda_k \sigma}{2\|\tilde{w}\|^\nu} \|w_i\|^{1+\nu},$$

and

$$\|w_i\| \overset{(i)}{\leq} \|w^\star\| \overset{(ii)}{\leq} \|\tilde{w}\|$$

due to $(i)$ Lemma 6 and $(ii)$ Lemma 7 and the fact that $\lambda_k \geq \tilde{\lambda}$. Therefore, for $\nu \in [0, 1]$ we have $\frac{H}{2} \|w_i\|^{1+\nu} \leq \frac{\lambda_k \sigma}{2} \|w_i\|$, and $\|r_i\| \leq \frac{\lambda_k \sigma}{2} \|w_i\|$ admits an identical bound by the MinRes termination condition. Substituting back into (30), we conclude that

$$\left\| x - \left( y - \frac{1}{\lambda_k} \nabla f(x) \right) \right\| \leq \frac{1}{\lambda_k} \left( \|r_i\| + \frac{H}{2} \|w_i\|^{1+\nu} \right) \leq \sigma \|w_i\| = \sigma \|x - y\|$$

and consequently reaching $\lambda_k \geq \tilde{\lambda}$ ensures that the MS condition holds. However, the algorithm returns a value of $\lambda$ such that for $\lambda_k = \lambda/2$ the MS condition check fails, meaning that $\lambda/2 < \tilde{\lambda}$.

It remains to argue that when Algorithm 3 returns with $\lambda \leq 2\tilde{\lambda}$, an appropriate movement bound holds. To that end, recall that (by $\lambda$-strong convexity of the quadratic subproblem)

$$\|w^\star\| - \|w_i\| \leq \|w_i - w^\star\| \leq \frac{1}{\lambda} \|r_i\| \leq \frac{\sigma}{2} \|w_i\|$$

and consequently

$$\|w_i\| \geq \frac{2}{3} \|w^\star\|.$$

Moreover, since $\lambda \leq 2\tilde{\lambda}$, we have

$$\|w^\star\| \overset{(i)}{\geq} \frac{1}{2} \|\tilde{w}\| \overset{(ii)}{=} \frac{1}{2} \left( \frac{\sigma \tilde{\lambda}}{H} \right)^{\frac{1}{\nu}} \overset{(iii)}{\geq} \frac{1}{2} \left( \frac{\sigma \lambda}{2H} \right)^{\frac{1}{\nu}},$$

due to $(i)$ Lemma 7 and $\lambda \leq 2\tilde{\lambda}$, $(ii)$ the definition of $\tilde{\lambda}$ and $(iii)$ $\lambda \leq 2\tilde{\lambda}$ again. Combining the last two displays, we have

$$\|x - y\| = \|w_i\| \geq \frac{1}{3} \left( \frac{\sigma \lambda}{2H} \right)^{\frac{1}{\nu}} \geq \left( \frac{\lambda}{c^{1+\nu}} \right)^{1/\nu} \quad \text{for } c = (6H/\sigma)^{\frac{1}{1+\nu}},$$

as required. $\qquad \square$

**Corollary 5.** *Consider Algorithm 1 with initial point $x_0$, parameters $\alpha$ satisfying $1.1 \leq \alpha = O(1)$ and $\lambda_0'$, and $\sigma$-MS oracle $\mathcal{O}_{\text{aMSN-fo}}$ with LAZY set to True in all but the first iteration and $\sigma \in (0.01, 0.99)$. For any $L, H, \epsilon > 0$, $\nu \in [0, 1]$ and any $x_\star \in \mathbb{R}^d$, if $f$ is convex with $(H, \nu)$-Hölder Hessian and L-Lipschitz gradient, the algorithm produces an iterate $x_T$ such that $f(x_T) \leq f(x_\star) + \epsilon$ within $T = O\left( \left( \frac{H\|x_0 - x_\star\|^{2+\nu}}{\epsilon} \right)^{2/(4+3\nu)} \right)$ iterations and at most $O\left( \left( \frac{L\|x_0 - x_\star\|^2}{\epsilon} \right)^{1/2} + \sqrt{\frac{L}{\lambda_0'}} + \log \frac{\lambda_0'}{L} \right)$ gradient and Hessian-vector product evaluations.*

*Proof.* Throughout the proof, we let $T$ denote the index of the first iteration of Algorithm 1 for which $f(x_T) \leq f(x_\star) + \epsilon$; the claimed bound on total number of iterations is an immediate corollary of both Theorem 1 and Theorem 4.

We now bound the complexity of Algorithm 3 with an approach similar to the proof of Corollary 3. To do so, we categorize all iterations into the following two cases: $(i)$ $t > 1$ and (since LAZY is true) $\lambda_t \geq \lambda'_t$ or $(ii)$ the first iteration $t = 1$. Now using Theorem 4, we know that in case $(i)$ the number of Hessian-vector product and gradient evaluations is bounded by

$$
N_{(i)} = O\left( \sum_{t=2}^{\mathsf{T}} \left( \sqrt{1 + \frac{\|\nabla^2 f(y_t)\|_{\mathrm{op}}}{\sigma \lambda'_t}} + \log \frac{\lambda_t}{\lambda'_t} \right) \right) \overset{(a)}{=} O\left( \sum_{t=2}^{\mathsf{T}} \left( \sqrt{1 + \frac{L}{\lambda'_t}} + \log \frac{L}{\lambda'_t} \right) \right)
$$

$$
= O\left( \sum_{t=2}^{\mathsf{T}} \left( 1 + \sqrt{\frac{L}{\lambda'_t}} \right) \right) \overset{(b)}{=} O\left( T + \sqrt{LA_T} \right),
$$

where we use $(a)$ that $\|\nabla^2 f(y_t)\|_{\mathrm{op}} \leq L$ by the assumption of $L$-Lipschitz gradient, and that either $\lambda_t = \lambda'_t$ or $\lambda_t = O(L)$ by the movement bound guaranteed from Theorem 4 (since $L$-Lipschitz gradient means $(L, 0)$-Lipschitz Hessian), and $(b)$ the bound (16) from Lemma 1. Note that $T = O\left( \sqrt{\frac{L\|x_0 - x_\star\|^2}{\epsilon}} \right)$ due to the iteration count bound for $(L, 0)$-Hölder Hessian. Moreover, noting that $A_T = O(A_{T-1})$ by Lemma 10 and that $A_{T-1} = O(\|x_0 - x_\star\|^2/\epsilon)$ as argued in the proof of Corollary 3, we have $\sqrt{LA_T} = O\left( \sqrt{\frac{L\|x_0 - x_\star\|^2}{\epsilon}} \right)$ as well. Therefore,

$$
N_{(i)} = O\left( \sqrt{\frac{L\|x_0 - x_\star\|^2}{\epsilon}} \right).
$$

For case $(ii)$, Theorem 4 gives allows us to bound the number of first-order operations by

$$
N_{(ii)} = O\left( \sqrt{1 + \frac{\|\nabla^2 f(x_0)\|_{\mathrm{op}}}{\sigma \min(\lambda'_0, \lambda_1)}} + \left| \log \frac{\lambda_1}{\lambda'_0} \right| \right)
$$

$$
= O\left( \sqrt{1 + \frac{\|\nabla^2 f(x_0)\|_{\mathrm{op}}}{\lambda'_0} + \log \frac{\lambda_1}{\lambda'_0}} + \sqrt{1 + \frac{\|\nabla^2 f(x_0)\|_{\mathrm{op}}}{\lambda_1}} + \log \frac{\lambda'_0}{\lambda_1} \right)
$$

$$
= O\left( \sqrt{\frac{L}{\lambda'_0}} + \sqrt{\frac{L\|x_0 - x_\star\|^2}{\epsilon}} + \log \frac{\lambda'_0}{L} \right),
$$

where for the last equality we use $\log(\lambda'_0/\lambda_1) \leq \log(\lambda'_0/L) + \log(L/\lambda_1)$ and the bounds $\sqrt{\frac{L}{\lambda_1}} = O(\sqrt{LA_1}) = O\left( \sqrt{\frac{L\|x_0 - x_\star\|^2}{\epsilon}} \right)$ and $\lambda_1 = O(L)$ as in previous case. Summing up the $N_{(i)}$ and $N_{(ii)}$ yields the the claimed bound. $\square$

## D.4 Auxiliary results

Here, we list technical results invoked throughout the section.

**Lemma 5.** *Let $f : \mathbb{R}^d \to \mathbb{R}^d$ be convex, and suppose that for some $\sigma \in (0, 1)$, $x, y \in \mathbb{R}^d$ and $\lambda > 0$ the MS condition*

$$
\left\| x - \left( y - \tfrac{1}{\lambda} \nabla f(x) \right) \right\| \leq \sigma \|x - y\|
$$

*holds. Then, for any $x_\star \in \mathbb{R}^d$, we have*

$$
f(x) \leq f(x_\star) + \frac{\lambda}{2} \|x_\star - y\|^2 - \frac{\lambda(1 - \sigma^2)}{2} \|x - y\|^2.
$$

*Therefore*

$$
\lambda \geq \frac{2(f(x) - f(x_\star))}{\|x_\star - y\|^2} \quad \text{and, if } f(x) \geq f(x_\star), \ \|x - y\| \leq \frac{1}{\sqrt{1 - \sigma^2}} \|x_\star - y\|.
$$

*Proof.* Let $F(x') = f(x') + \frac{\lambda}{2}\|x' - y\|^2$. Since $F$ is $\lambda$-strongly convex, we have

$$f(x) + \frac{\lambda}{2}\|x - y\|^2 - f(x_\star) - \frac{\lambda}{2}\|x_\star - y\|^2 = F(x) - F(x_\star) \leq \frac{\|\nabla F(x)\|^2}{2\lambda}$$

for every $x_\star \in \mathbb{R}^d$. Moreover, the MS condition yields

$$\frac{\|\nabla F(x)\|^2}{2\lambda} = \frac{\lambda}{2}\left\|x - \left(y - \frac{1}{\lambda}\nabla f(x)\right)\right\|^2 \leq \frac{\lambda\sigma^2}{2}\|x - y\|^2.$$

The lemma follows by substituting back and rearranging. $\qquad\square$

**Lemma 6.** *Let $A \in \mathbb{R}^{d \times d}$ be a positive definite symmetric matrix, let $b \in \mathbb{R}^d$, and let $w^\star = A^{-1}b$. The iterates $\{w_t\}$ and residuals $\{r_t = Aw_t - b\}$ of the Conjugate Residuals/MinRes algorithm [42, 18] for minimizing $\|Aw - b\|$ satisfy*

1. *$\|w_t\|$ is non-decreasing in $t$ with $\|w_\infty\| = \|w^\star\|$,*

2. *$\|r_t\|$ is non-increasing in $t$ with $\|r_\infty\| = 0$,*

3. *$\|r_t\| = O\left(\frac{\|A\|_{\mathrm{op}}\|w^\star\|}{t^2}\right)$.*

*Proof.* The first two parts of the lemma are Theorems 2.3 and 2.4 of Fong and Saunders [18], respectively. To show the third part, we cite Lee et al. [28] which give a gradient method that, for any $L$-smooth convex function $h$ with minimizer $x_\star$, produces iterates $x_t$ such that $\|\nabla h(x_t)\| = O(L\|x_\star - x_0\|/t^2)$ [28, Corollary 1]. Applying this method to $h(x) = \frac{1}{2}x^\top Ax - b^\top x$ with $x_0 = 0$, which is convex and $\|A\|_{\mathrm{op}}$-smooth with minimizer $w^\star$, guarantees $\|Ax_t - b\| = \|\nabla h(x_t)\| = O(\|A\|_{\mathrm{op}}\|w^\star\|/t^2)$. Moreover, we note that $x_t$ is in the linear span of $\nabla h(0), \nabla h(x_1), \ldots \nabla h(x_{t-1})$ and consequently in the Krylov subspace $\mathrm{span}(b, Ab, \ldots, A^{t-1}b)$. Therefore $\|r_t\| \leq \|Ax_t - b\|$ by definition of the Conjugate Residuals/MinRes method. $\qquad\square$

**Lemma 7.** *Let $A \in \mathbb{R}^{d \times d}$ be positive semidefinite, let $b \in \mathbb{R}^d$ and let $\Delta(\lambda) = \|(A + \lambda I)^{-1}b\|$. Then, for any $\lambda_1 \leq \lambda_2$ we have*

$$\frac{\lambda_1}{\lambda_2}\Delta(\lambda_1) \leq \Delta(\lambda_2) \leq \Delta(\lambda_1).$$

*Proof.* This is an immediate consequence of

$$\frac{\lambda_1}{\lambda_2}(A + \lambda_2 I) \preceq A + \lambda_1 I \preceq A + \lambda_2 I$$

and the fact that if $0 \prec M_1 \preceq M_2$ and $M_1, M_2$ have the same eigenvectors, then $\|M_1^{-1}b\| \geq \|M_2^{-1}b\|$ for all $b$, since $M_1 \preceq M_2$ implies that the eigenvalues of $M_2$ majorize the eigenvalues of $M_1$. $\quad\square$

**Lemma 8.** *Let $f : \mathbb{R}^d \to \mathbb{R}$ and $x, y \in \mathbb{R}^d$, and suppose that $\nabla^2 f$ is $(H, \nu)$-Hölder continuous for some $\nu \in [0, 1]$ in a ball of radius $\|x - y\|$ around $y$. Then, for any $\lambda$,*

$$\left\|x - \left(y - \frac{1}{\lambda}\nabla f(x)\right)\right\| \leq \frac{1}{\lambda}\|(\nabla^2 f(y) + \lambda I)(x - y) + \nabla f(y)\| + \frac{H}{(1 + \nu)\lambda}\|x - y\|^{1+\nu}.$$

*Proof.* Let

$$\delta = \nabla f(x) - \nabla f(y) - \nabla^2 f(y)(x - y).$$

The local $H$-Hölder continuity of $\nabla^2 f$ around $y$ yields

$$\|\delta\| \leq \frac{H}{1 + \nu}\|x - y\|^{1+\nu}.$$

Moreover, for

$$r = (\nabla^2 f(y) + \lambda I)(x - y) + \nabla f(y)$$

algebraic manipulation yields

$$x - \left(y - \frac{1}{\lambda}\nabla f(x)\right) = \frac{1}{\lambda}(r + \delta),$$

and the lemma holds via the triangle inequality. $\qquad\square$

For the following lemma, recall the notation $\mathcal{S}_T^> = \{t \le T \mid \lambda_t > \lambda'_t\}$.

**Lemma 9.** *For every $T > 0$, $\nu \in [0, 1]$ and $x_\star \in \mathbb{R}^d$ such that $f(x_T) \ge f(x_\star)$, the iterates of Algorithm 1 with $\sigma \in (0.01, 0.99)$ and $\alpha \in (1.01, O(1))$ satisfy*

$$\sum_{t \in \mathcal{S}_T^>} \left( \frac{\|\tilde{x}_t - y\|^\nu}{\lambda'_t} \right)^{2/(4+3\nu)} = O\left( \|x_0 - x_\star\|^{2\nu/(4+3\nu)} A_T^{2/(4+3\nu)} \right).$$

*Proof.* The case $\nu = 0$ follows immediately from eq. (15) in Lemma 1. For $\nu \in (0, 1]$, define $u_t = \frac{\|\tilde{x}_t - y\|^2}{(\lambda'_t)^{2/\nu}} \mathbb{1}_{\{t \in \mathcal{S}_T^>\}}$ and $v_t = (\lambda'_t)^{1+2/\nu} A_t$. The reverse Hölder inequality gives, for any $q > 1$,

$$\left( \sum_{t \le T} u_t^{1/q} \right)^q \left( \sum_{t \le T} v_t^{-1/(q-1)} \right)^{-(q-1)} \le \sum_{t \le T} u_t v_t. \tag{31}$$

Substituting back the definitions of $u_t$ and $v_t$ we have

$$\sum_{t \le T} u_t v_t = \sum_{t \in \mathcal{S}_T^>} \|\tilde{x}_t - y\|^2 A_t \lambda'_t = O(\|x_0 - x_\star\|^2), \tag{32}$$

with the last transition due to eq. (9) in Proposition 1 and the fact that $f(x_T) \ge f(x_\star)$. Next, we substitute $q = \frac{4+3\nu}{\nu}$ and note that

$$\sum_{t \le T} v_t^{-\nu/(4+2\nu)} = \sum_{t \le T} \frac{1}{\sqrt{\lambda'_t} A_t^{\nu/(4+2\nu)}} = O\left( \sum_{t \le T} \frac{(\lambda'_t)^{-1/2}}{\left( \sum_{j \le t} (\lambda'_j)^{-1/2} \right)^{\nu/(2+\nu)}} \right)$$

with the last transition due to

$$\sqrt{A_{t'}} = \Omega\left( \sum_{t \le t'} \frac{1}{\sqrt{\lambda'_t}} \right) \tag{33}$$

for all $t'$ by eq. (16) in Lemma 1. Note that for every non-decreasing sequence $0 = B_0 \le B_1 \le B_2 \le \cdots \le B_T$ we have

$$\sum_{t \le T} \frac{B_t - B_{t-1}}{B_t^{\nu/(2+\nu)}} \le \sum_{t \le T} \frac{(B_t^{2/(2+\nu)} - B_{t-1}^{2/(2+\nu)})(B_t^{\nu/(2+\nu)} + B_{t-1}^{\nu/(2+\nu)})}{B_t^{\nu/(2+\nu)}}$$

$$\le 2 \sum_{t \le T} (B_t^{2/(2+\nu)} - B_{t-1}^{2/(2+\nu)}) = 2 B_T^{2/(2+\nu)}.$$

Substituting $B_t = \sum_{j \le t} (\lambda'_j)^{-1/2}$, we have

$$\sum_{t \le T} v_t^{-\nu/(4+2\nu)} = O\left( \left( \sum_{t \le T} \frac{1}{\sqrt{\lambda'_t}} \right)^{2/(2+\nu)} \right) = O(A_T^{1/(2+\nu)}),$$

with the final bound again using (33). This implies

$$\left( \sum_{t \le T} v_t^{-\nu/(4+2\nu)} \right)^{(4+2\nu)/\nu} = O(A_T^{2/\nu}). \tag{34}$$

Substituting $q = (4 + 3\nu)/\nu$ and the bounds (34) and (32) into (31) completes the proof. □

**Lemma 10.** *For every $T > 0$, the sequence $\{A_t\}$ in Algorithm 1 with $\alpha \in (1.01, O(1))$ satisfies $A_{t+1} = O(A_t)$.*

*Proof.* Note that

$$A'_{t+1} - A_t = a'_{t+1} = \sqrt{\frac{A'_{t+1}}{\lambda'_{t+1}}}$$

and therefore (since $A'_{t+1} > A_t$)

$$\sqrt{A'_{t+1}} \leq \sqrt{A_t} + \frac{1}{\sqrt{\lambda'_{t+1}}} \overset{(i)}{\leq} \sqrt{A_t} + O\left(\frac{1}{\sqrt{\lambda'_t}}\right) \overset{(ii)}{\leq} O(\sqrt{A_t})$$

due to $(i)$ $\lambda'_{t+1} \geq \lambda'_t/\alpha = \Omega(\lambda'_t)$ by the algorithm's construction and $(ii)$ $\frac{1}{\sqrt{\lambda'_t}} = O(\sqrt{A_t})$ by the last inequality in Lemma 1. The proof is complete by noting that $A_{t+1} \leq A'_{t+1}$. $\square$

## E  Experiments

This section provides the full details of the experiment we report in Section 4 (in Appendix E.1), as well as results of the additional experiments: algorithm comparison across additional datasets (Appendix E.2), parameter sensitivity of our algorithm (Appendix E.3), effect of changing the parameter $M$ in $\mathcal{O}_{\mathrm{cr}}$ (Appendix E.4), and the effect of momentum on the worst-case instance for Lipschitz-Hessian functions (Appendix E.5). Finally, we also demonstrate empirically the importance of the momentum damping mechanism in Algorithm 1 (Appendix E.6).

### E.1  Main experiment details

We report experiments for logistic regression objectives of the form

$$f(x) := \frac{1}{n} \sum_{i \in [n]} \log\left(1 + \exp\left(-c_i \phi_i^\top x\right)\right),$$

where each $\phi_i \in \mathbb{R}^d$ is a feature vector with a corresponding label $c_i \in \{-1, 1\}$.

**Implementation details.**  We now provide the key implementation details for all the algorithms considered in our experiments. For a complete description please refer to the Python implementations submitted with this manuscript.

- **Algorithm 1.** Our implementation of Algorithm 1 follows its pseudocode precisely. We keep $\lambda'_0 = 0.1$ throughout and set $\alpha = 2$ in all experiments except for those in Appendix E.3, where we test how changing it affects performance.

- **Algorithm 0 [32].** A direct implementation of the pseudocode of Algorithm 0 would be quite inefficient, since stating off the bisection with a large interval $[\lambda^\ell, \lambda^h]$ at each iterations will waste many oracle calls. Instead, we implement a *strong baseline* for our bisection-free algorithm by starting each bisection with a guess $\lambda^0_{t+1}$ determined by the previous iterations. We construct this guess using the scheme[8] for updating $\lambda'_t$ in Algorithm 1: if the previous final bisection output $\lambda_t$ and the previous initial bisection guess $\lambda^0_t$ satisfy $\lambda_t > \lambda^0_t$, we let $\lambda^0_{t+1} = 2\lambda^0$ and otherwise we set $\lambda^0_{t+1} = \frac{1}{2}\lambda^0_t$. We take $\lambda^0_1 = 0.1$.

  To construct a bisection interval out of the initial guess $\lambda^0_{t+1}$, we adopt a strategy similar to the ones used in Algorithms 2 and 3. To explain it, define the following terminology. Consider some $\lambda'_{t+1}$ and $\lambda_{t+1}$ computed by applying an MS oracle to $y_t$ and $\lambda'_{t+1}$, with $y_t$ computed from $\lambda'_{t+1}$ as in lines 5 and 8 of Algorithm 0. We say that $\lambda'_{t+1}$ is *valid* if $\lambda_{t+1} \in [\frac{1}{\rho}\lambda'_{t+1}, \lambda'_{t+1}]$, that $\lambda_{t+1}$ is *high* if $\lambda_{t+1} < \frac{1}{\rho}\lambda'_{t+1}$, and that $\lambda'_{t+1}$ is *low* if $\lambda_{t+1} > \lambda'_{t+1}$. If $\lambda^0_{t+1}$ is valid, we simply use it and there is no need for bisection. Otherwise, if it is low, we take $\lambda^\ell_{t+1} = \lambda^0_{t+1}$, and repeatedly double $\lambda^0_{t+1}$ until we find some $2^k\lambda^0_{t+1}$ that is either valid or high. In the former case we are again done, and in the latter case we set $\lambda^h_{t+1} = 2^k\lambda^0_{t+1}$ and continue with the bisection as described in Algorithm 0, except that (inspired by Algorithm 2) at each iteration we take $\lambda'_{t+1}$ to be the geometric mean of

---

[8]We also experimented with the heuristic $\lambda^0_{t+1} = \frac{1}{2}\lambda_t$, which performed slightly worse.

$\lambda_{t+1}^{\ell}$ and $\lambda_{t+1}^{h}$ rather than the arithmetic mean. The case that $\lambda_{t+1}^{0}$ is high is treated analogously, setting $\lambda_{t+1}^{h} = \lambda_{t+1}^{0}$ and repeatedly halving it until finding $2^{-k}\lambda_{t+1}^{0}$ that is either valid or low. Finally, we note that for $\sigma$-MS oracles with $\sigma > 0$ the bisection is only guaranteed to succeed when $\rho$ is sufficiently large. The precise value of $\rho$ depends on $\sigma$ and the order of the movement bound guaranteed by the oracle. Instead of attempting a precise calculation, we set $\rho = 4$.

- **Cubic-regularized Newton Method (CR) [37].** The method consists of simply iterating $x_{t+1} = \mathcal{O}_{\text{cr}}(x_t)$. For when the parameter $M$ in $\mathcal{O}_{\text{cr}}$ is set to $M = 0$, the method reduces to the classical **Newton's method** $x_{t+1} = -[\nabla^2 f(x_t)]^{-1}\nabla f(x_t)$.

- **Accelerated CR (ACR) [33].** We implement [33, Alg. 4.8] without changes.

- **Adaptive ACR [21].** We implement [21, Alg. 4] without changes.

- **Song et al. [41] heuristic.** Following a proposal in [41], we consider a version of Algorithm 0 that uses a single pre-specified sequence of $\lambda_t'$ without checking whether the resulting $\lambda_t$ is in the interval $[\frac{1}{\rho}\lambda_t', \lambda_t']$. We compute the sequence by setting $A_t = A_t' = \frac{1}{2HR}(t/3)^{7/2}$, and taking $\lambda_{t+1}' = \frac{A_{t+1}}{a_{t+1}^2} = \frac{A_{t+1}}{(A_{t+1}-A_t)^2}$. Here $H$ is an estimate of the function's Hessian Lipschitz constant (see below), and $R$ is an estimate of the Euclidean distance between $x_0$ from an optimal point. We obtain $R$ by using the default scikit-learn logistic regression solver [39]; it finds a far less accurate solution than the methods we consider but provides a reasonably accurate estimate of $R$.

- **Iterating $\mathcal{O}_{\text{aMSN}}$.** This scheme corresponds to simply iterating $x_{t+1}, \lambda_{t+1} = \mathcal{O}_{\text{aMSN}}(x_t, \lambda_t/2)$, with the initial $\lambda_1$ set to 0.1.

- **Iterating $\mathcal{O}_{\text{aMSN-fo}}$.** This scheme corresponds to simply iterating $x_{t+1}, \lambda_{t+1} = \mathcal{O}_{\text{aMSN-fo}}(x_t, \lambda_t/2)$, with the initial $\lambda_1$ set to 0.1.

- **Gradient descent (GD).** We iterate $x_{t+1} = x_t - \eta\nabla f(x_t)$ and choose the best value of $\eta$ from $\{3, 10, 30, 100, 1000, 3000\}$, making sure the best value is never on the edge of the grid, i.e., 3 and 3000 are never chosen.

- **Accelerated gradient descent (AGD) [38].** We implement the algorithm precisely as described in [38], and tune the step size $\eta$ as described for GD.

- **L-BFGS-B [10, 44].** We use the implementation available from SciPy [43], where we set all tolerance parameter to a very small value so that the algorithm only stops after exceeding the specified maximum number of iterations.

- **$\mathcal{O}_{\text{cr}}$.** To solve the problem (1) and implement $\mathcal{O}_{\text{cr}}$, we perform a bisection over $\lambda$ to solve for $\lambda$ that satisfies $\lambda \approx \frac{M}{2}\|[\nabla^2 f(y) + \lambda I]^{-1}\nabla f(y)\|$, and return $x = y - [\nabla^2 f(y) + \lambda I]^{-1}\nabla f(y)$. To ensure a high-quality solution to the implicit equation for $\lambda$, we stop the bisection only when $\frac{\lambda}{\frac{M}{2}\|[\nabla^2 f(y) + \lambda I]^{-1}\nabla f(y)\|} \in [1 - 10^{-5}, 1 + 10^{-5}]$. This results in a slow implementation of $\mathcal{O}_{\text{cr}}$ (requiring a lot of linear system solutions), but provides the ideal point of comparison since we measure complexity by number of Hessian evaluations. To ensure numerical stability, we also stop the bisection if the value of $\lambda$ falls below $\lambda_{\text{Newton}} = 10^{-10}$.

- **$\mathcal{O}_{\text{aMSN}}$.** Our implementation follows the pseudocode of Algorithm 2 precisely, except that, to ensure numerical stability, we stop the procedure if $\lambda$ falls below $\lambda_{\text{Newton}} = 10^{-10}$. When combining the oracle with Algorithm 1, we set the LAZY to be True in all iterations except the first, as in Corollary 3. In all other settings we set LAZY to be False.

- **$\mathcal{O}_{\text{aMSN-fo}}$.** Our implementation follows the pseudocode of Algorithm 3, except we only implement the case that LAZY =True, since doing otherwise appears less practical; it is not hard to extend Theorem 1 and Corollary 3 to provide similar guarantees even when the first iteration is lazy.

**Initialization.** We initialize all algorithms at the origin, i.e., with $x_0 = 0$.

**Estimating the Hessian Lipschitz constant.** For all algorithms that require an estimate for the Lipschitz constant $H$ of $\nabla^2 f$ (i.e., all the algorithms that use $\mathcal{O}_{\text{cr}}$), we set $H = \frac{1}{10}\bar{H}$, where $\|\frac{1}{n}\sum_{i=1}^{n}\phi_i\phi_i^{\top}\|_{\text{op}}\max_{i\in[n]}\|\phi_i\|$ is a conservative upper bound on the Lipschitz constant of $\nabla^2 f$ for logistic regression [41]. We explore the effect of varying the estimate $H$ in Appendix E.4. Note that value of $M$ given to $\mathcal{O}_{\text{cr}}$ is typically $2H$: for Algorithms 0 and 1 it is $H/\sigma$ and $\sigma = 2$, while CR and ACR also use $M = 2H$. We also use $H$ as the initial guess for Adaptive ACR.

**Datasets and preprocessing.** We compare our methods to other baselines using the following binary classification datasets:

- **a9a** ($n = 32,561$ and $d = 123$)

- **w8a** ($n = 49,749$ and $d = 300$)

- **splice** ($n = 1,000$ and $d = 60$)

- **synthetic** ($n = 500$ and $d = 200$).

The first three datasets are from LIBSVM [15], which is available under a BSD 3-Clause "New" or "Revised" license. The synthetic dataset is generated by sampling half of the data points from $\mathcal{N}_1(\mu_1, I)$ and the other half from $\mathcal{N}_1(\mu_2, I)$, where $\mu_1, \mu_2 \in \mathbb{R}^d$ are independent random vectors uniformly drawn from a sphere with radius $0.5$.

For all datasets we normalize the feature vectors, such that for every $i \in [n]$, each feature vector $\phi_i$ is a unit norm.

## E.2 Replicating Figure 1 with additional datasets

In Figure 2 we compare all the algorithms described above on logistic regression with the datasets: "a9a" (panels a-c), "w8a" (panels d-f), "splice" (panels g-i) and the synthetic dataset (panels j-l).

Among non-adaptive methods (panels a, d, g and j), Algorithm 1 outperforms the other non-adaptive methods while Algorithm 0 is consistently the second best-performing method.

Comparing adaptive methods (panels b, e, h and k), we see that our implementation of Algorithm 1 with the adaptive oracle $\mathcal{O}_{\text{aMSN}}$ converges faster than adaptive ACR and Algorithm 0 with $\mathcal{O}_{\text{aMSN}}$ for all datasets. However, our scheme that only iterates $\mathcal{O}_{\text{aMSN}}$ without momentum converges even faster, and Newton's method outperforms all second-order methods.

For first-order methods (panels c, f, i and l), iterating $\mathcal{O}_{\text{aMSN-fo}}$ scheme is comparable to L-BFGS-B on 2 out of 4 datasets and is faster than tuned AGD in 3 out of 4 datasets. On the synthetic dataset it is about twice slower than L-BFGS-B but still faster than tuned AGD, while on w8a it is about 50% slower than L-BFGS-B and tuned AGD, which perform comparably.

## E.3 Parameter sensitivity of Algorithm 1

We test the sensitivity of Algorithm 1 combined with our adaptive oracle (Algorithm 2 or Algorithm 3) to the parameters $\alpha$ and $\sigma$. Figure 3 shows that Algorithm 1 second-order oracle $\mathcal{O}_{\text{aMSN}}$ performs essentially the same for all $\alpha$ in the range $1.2$ to $8$, and that the oracle's performance is similar for $\sigma = 0.1$ and $\sigma = 0.25$, but slightly degrades for larger and smaller $\sigma$. Algorithm 1 combined with the first-order oracle $\mathcal{O}_{\text{aMSN-fo}}$ is a bit more sensitive to $\alpha$ (performing best for $\alpha = 1.2$), but is less sensitive to $\sigma$, showing similar performance for all $\sigma$ values except the very smallest $\sigma = 0.01$.

## E.4 Varying $M$ for $\mathcal{O}_{\text{cr}}$

In this section we test the performance of non-adaptive methods (i.e., the methods that use $\mathcal{O}_{\text{cr}}$) when changing the estimate of the function's Lipschitz constant $H$. In particular, we consider values of $H$ of the form $\beta \bar{H}$, where $\bar{H} = \|\frac{1}{n} \sum_{i=1}^{n} \phi_i \phi_i^\top\|_{\text{op}} \max_{i \in [n]} \|\phi_i\|$ is an upper bound on the Hessian Lipschitz constant and $\beta$ varies in $\{1, 10^{-1}, 10^{-2}, 10^{-3}, 10^{-4}, 10^{-5}, 10^{-6}, 10^{-7}, 10^{-8}\}$. (The experiments in Appendix E.2 correspond to $\beta = 0.1$). Figure 4 shows that our adaptive accelerated scheme (Algorithm 1 with the $\mathcal{O}_{\text{aMSN}}$) outperforms all non-adaptive methods with their optimal $H$ value, except for the CR method that has optimal $H \approx 0$ and therefore is almost equivalent to Newton's method.

## E.5 Performance on a worst-case instance

Having observed that our adaptive oracle $\mathcal{O}_{\text{aMSN}}$ performs better on logistic regression without the "acceleration" scheme in Algorithm 1, we now test whether Algorithm 1 demonstrably accelerates $\mathcal{O}_{\text{aMSN}}$ on a different, harder problem. In particular, we consider the worst case instance [3, 21, 16]

$f : \mathbb{R}^d \to \mathbb{R}$ given by

$$f(x) = \left| x^{(1)} - 1 \right|^3 + \sum_{i=2}^{d} \left| x^{(i)} - x^{(i-1)} \right|^3.$$

We note that, for $t < d$, the optimal rate of convergence for any of the algorithms we consider (which can only "discover" one coordinate of $f$ per iterations) is $O(t^{-2})$, or $O\left( \left\| x_0 - x_\star^{(t)} \right\|^3 t^{-3.5} \right)$ where $\left\| x_0 - x_\star^{(t)} \right\| = \Theta(\sqrt{t})$ is the distance between the initial point and the best solution with only $t$ non-zero coordinates.

In our experiments, we set $d = 3,000$ and compare the convergence rate of the following second-order methods: standard cubic regularized method (CR), its accelerated variant (ACR), Algorithm 1 with $\mathcal{O}_{cr}$, Algorithm 1 with the adaptive oracle $\mathcal{O}_{aMSN}$, and iterating the oracle $\mathcal{O}_{aMSN}$. For methods based on $\mathcal{O}_{cr}$ we estimate the Hessian Lipschitz constant to be $H = 10$. Figure 5 shows that the slope of the accelerated methods using $\mathcal{O}_{cr}$ (ACR and Algorithm 1) is sharper than the slope of the CR method, indicating a faster convergence rate due to the acceleration scheme. However, the convergence rate $\mathcal{O}_{aMSN}$ with and without the acceleration component is optimal. Therefore, even the worst-case instance for convex optimization with Lipschitz Hessian does not provide evidence that acceleration significantly benefits $\mathcal{O}_{aMSN}$.

### E.6 The importance of momentum damping in Algorithm 1

We compare our method (Algorithm 1 with $\mathcal{O}_{aMSN}$) to a variant of it that does not use the momentum damping mechanism. That is, we set $x_{t+1} = \tilde{x}_{t+1}$ and $a_{t+1} = a'_{t+1}$ regardless of the value of $\lambda_{t+1}$. As Figure 6 shows, without the momentum damping mechanism Algorithm 1 fails to converge on all the datasets we test ("a9a", "w8a", "splice", and the synthetic dataset).

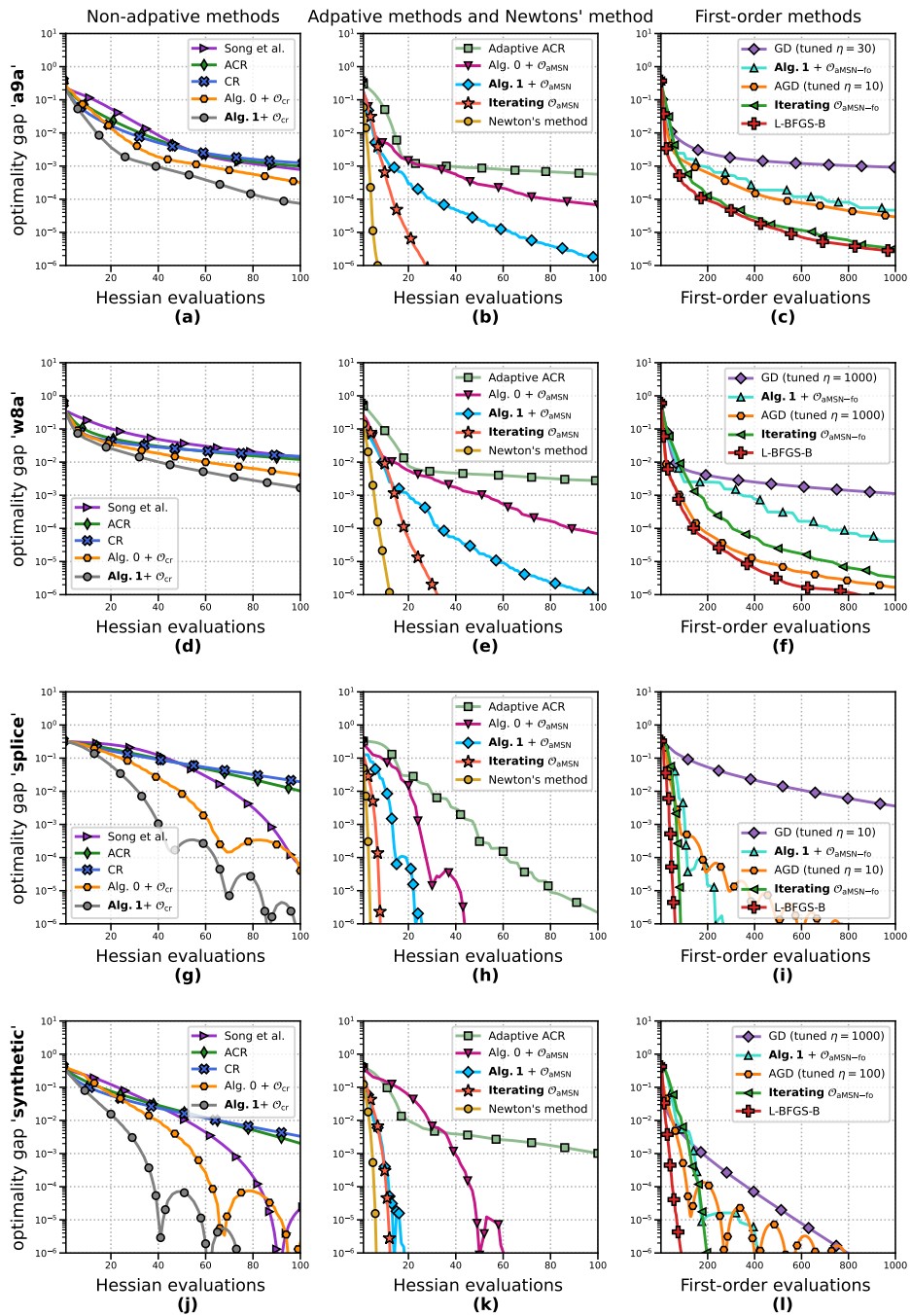

**Figure 2.** Empirical results on logistic regression with "a9a", "w8a", "splice" and a synthetic dataset. Boldface legend entries denote methods we contribute.

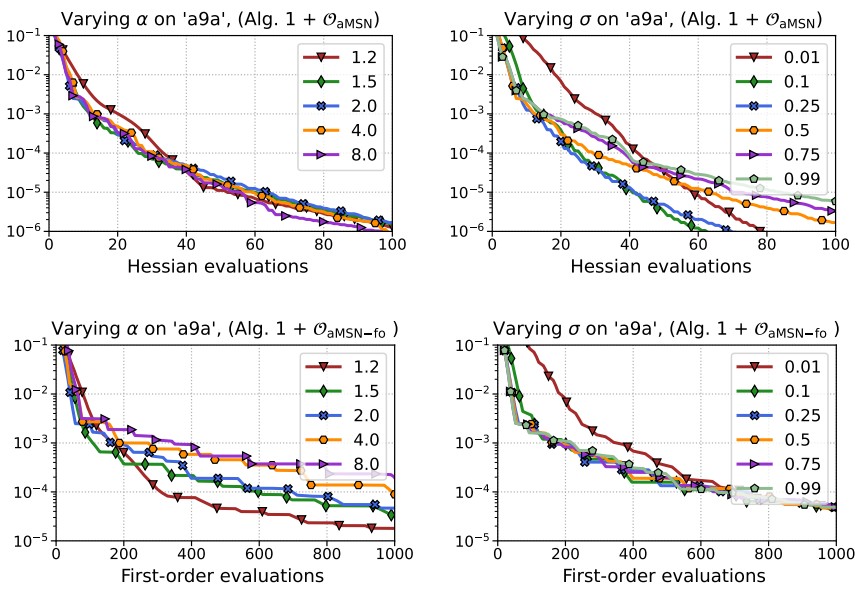

**Figure 3:** Testing the sensitivity of Algorithm 1 to the parameters $\alpha$ and $\sigma$ with the "a9a" dataset.

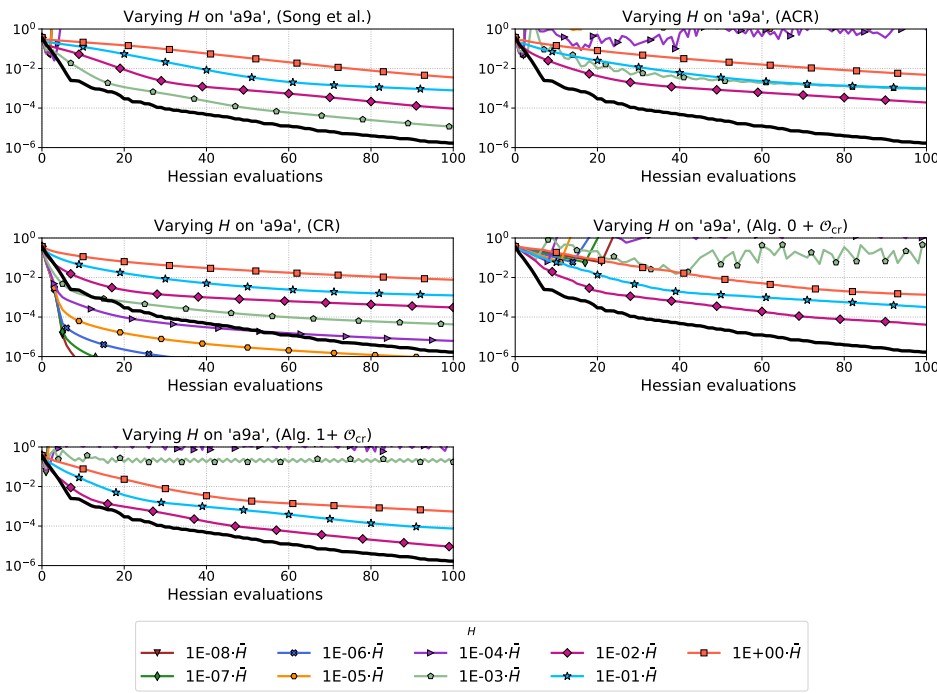

**Figure 4.** Varying the estimated Hessian Lipschitz constant $H$ non-adaptive methods. The thick black line corresponds our adaptive method (Algorithm 1 with $\mathcal{O}_{\mathsf{aMSN}}$).

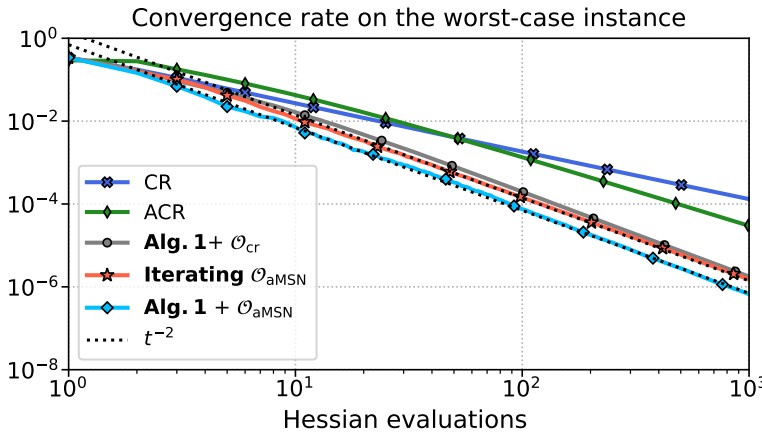

**Figure 5.** Empirical results on the worst case instance (the x-axis and y-axis are in logarithmic scale). Boldface legend entries denote methods we contribute.

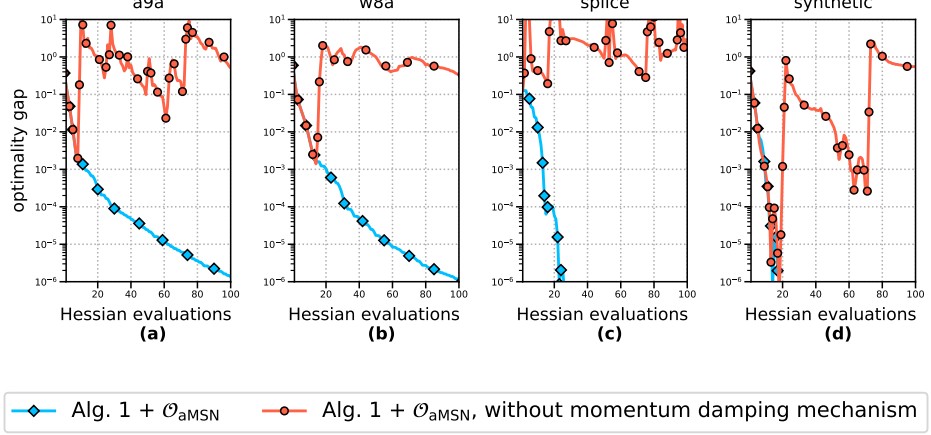

**Figure 6.** Algorithm 1 with (light blue line) and without (red line) the momentum damping mechanism. Title denotes the dataset name.