# OpenReview forum: "Optimal and Adaptive Monteiro-Svaiter Acceleration"
_NeurIPS.cc/2022/Conference — NeurIPS 2022 Accept_

### Official Review · Reviewer_RYyk · 2022-07-04

**Rating:** 8
**Confidence:** 4
**Soundness:** 4 excellent
**Presentation:** 4 excellent
**Contribution:** 4 excellent

**Summary:**

This work proposes a variant of the Monteiro-Svaiter (MS) acceleration framework that removes the logarithmic factor arises in its bisection procedure in each iteration. The authors then implement the MS oracle using several existing oracles and a new adaptive MS-Newton oracle. When using Taylor descent step to implement MS oracle, the resulting algorithm removes the logarithmic gap between the existing upper bounds and the lower bound for minimizing convex functions with Lipschitz pth order derivative, which resolves an important and challenging open problem in optimization theory. The authors also provide numerical evaluations which show some interesting observation regarding the effect of momentum for second-order methods, and point out several future directions.

**Questions:**

I am curious about the connection between the MS framework and the inexact high-order proximal points framework proposed in [1]. They have similar rate of convergence. Can the proposed MS framework serve as an upper-level process in the bi-level minimization framework? Is it possible to obtain lower-order methods that "overpass" traditional limits of the Complexity Theory similar to [1] under the MS framework?


[1] Inexact accelerated high-order proximal-point methods. Y Nesterov. Mathematical Programming, 2021.

**Limitations:**

The limitations and the potential negative societal impact have been adequately addressed by the authors.

**Strengths And Weaknesses:**

This work resolves the open problem of closing the gap between the upper bound and lower bound of minimizing convex functions with Lipschitz pth order derivative, which is a significant contribution. This problem is marked as an "open and challenging question in
Optimization Theory" by Nesterov in his 2018 textbook. The proof is highly technical and hard to verify. However, the authors provide proof sketches and numerical verifications to the proposed schemes, which increases my confidence in the correctness.

The authors also propose adaptive implementations of the MS-oracle for the second order case, and demonstrates the benefits through several numerical evaluations.

Overall, this is a significant, clearly presented and solid theoretical work. The content is pretty technical which seems not preventable for this research topic (constructing optimal methods). I strongly recommend to accept this work.

---

> ### Author Response · Authors · 2022-08-02
> **Author reply**
>
> Thank you for the positive review and insightful comments - we address them in detail below.
>
> > This problem is marked as an "open and challenging question in Optimization Theory" by Nesterov in his 2018 textbook.
>
> Thank you for pointing this out - it is satisfying to know that Nesterov noted this problem in his textbook. We will include this citation in the revised paper.
>
> > I am curious about the connection between the MS framework and the inexact high-order proximal points framework proposed in [1]. They have similar rate of convergence. Can the proposed MS framework serve as an upper-level process in the bi-level minimization framework?
>
> Thank you for pointing out [1], which is indeed a relevant reference that we have missed. We believe that the bi-level minimization framework provides a valuable additional perspective on optimal high-order acceleration. However, the main bottleneck for turning it into an implementable algorithm appears to lie in the fact that the framework assumes exact proximal point computations. That is, it does not provide a counterpart to the MS condition in Definition 1. Given such a counterpart, we believe it is likely that our techniques for removing bisection (namely, momentum damping and multiplicative $\lambda’$ updates) are likely to help in removing bisection from a bi-level-minization based method. This is an interesting topic for further research, and we will update our paper to discuss it.
>
> > Is it possible to obtain lower-order methods that "overpass" traditional limits of the Complexity Theory similar to [1] under the MS framework?
>
> This is an excellent question that we had not thought of before - thank you for bringing up! Indeed, we believe our developments yield improved complexity bounds for second-order methods for convex functions with Lipschitz *third* derivatives. In particular, Nesterov’s subproblem solution technique from the paper “Superfast second-order methods for unconstrained convex optimization” is an MS oracle implementation that (a) costs one Hessian evaluation and a logarithmic number of gradient evaluations and (b) satisfies an $(O(L_3), 3)$-movement bound when applied for convex functions with $L_3$ Lipschitz derivatives. Combining this oracle with our Algorithm 1 will immediately yield the rate of convergence $O(k^{-5})$, where $k$ is the number of Hessian evaluations. This is the optimal Hessian evaluation complexity (even given third derivatives access) and improves on the prior work “Near-Optimal Hyperfast Second-Order Method for Convex Optimization and its Sliding” by Kamzolov and Gasnikov by a logarithmic factor. We will be sure to mention this in the revised paper.
>
> These observations lead to the following question, which we find very interesting - can our adaptive second-order oracle (Algorithm 2) also yield the optimal $O(k^{-5})$ rate under third-order Lipschitz derivatives, without modification? We are currently unsure about the answer, and believe additional improvements might be required to enable this degree of adaptivity - this is a good topic for future work!

---

### Official Review · Reviewer_Zcbv · 2022-07-11

**Rating:** 7
**Confidence:** 2
**Soundness:** 3 good
**Presentation:** 3 good
**Contribution:** 3 good

**Summary:**

This paper presents a variant of the Monteiro-Svaiter (MS) acceleration framework that requires no bisection, but which maintains optimal complexity bounds (up to constant factors). The work by MS shows how to accelerate an oracle, and demonstrates the connection with the accelerated proximal point method, as well as establishing complexity guarantees. However, algorithms based upon this framework require the solution of an implicit equation to determine the dynamic regularization parameter, which typically requires the use of bisection, so that there is a 'gap' between the theoretical complexity and the practical behaviour.

This work 'bridges the gap' by developing an MS variant that requires no bisection, does not involve lots of parameters that need tuning, performs well in practice, and maintains the optimal complexity bounds. Moreover, they develop an adaptive alternative of the cubic regularization oracle that does not require tuning of parameter M (related to the Lipschitz constant of the Hessian), and obtains the optimal Hessian evaluation complexity for functions with order \nu Holder Hessian.

A first order implementation of the adaptive oracle is presented, which uses MinRes/Conjugate Residuals with an adaptive stopping criterion that comes from their analysis.

Finally, the authors also present numerical experiments in support of their work, and also show that momentum can be harmful for logistic regression.

**Questions:**

.

**Strengths And Weaknesses:**

I felt that the paper was well written, and the authors clearly identified how their work fitted in with the current literature, and also what the main contributions of the current work are. This was helpful in terms of readability.
I felt that the contribution (bridging the gap between optimal complexity vs practicality) was nice, and I think is a welcome addition to the field.

The paper is clearly, by its nature, quite technical. However, I felt that the authors were careful to include additional explanations to contextualize their work. I also felt the inclusion of Section 3.1 was really good.

---

> ### Author Response · Authors · 2022-08-02
> **Author reply**
>
> Thank you for the positive review. There is one point worth clarifying, which might positively impact your overall assessment: our paper is the first to obtain the optimal oracle complexity for convex optimization with Lipschitz or Holder Hessian. Due to the bisection, the complexity of prior works is greater than ours by a logarithmic factor. Therefore, our work not only bridges theory and practice - *it also improves the theory* by settling the complexity of basic optimization problems up to constant factors.

---

### Official Review · Reviewer_dAbm · 2022-07-13

**Rating:** 8
**Confidence:** 4
**Soundness:** 4 excellent
**Presentation:** 3 good
**Contribution:** 4 excellent

**Summary:**

In this paper, the authors present a new version of the Monteiro-Svaiter (MS)
acceleration framework. The main feature of this new version, compared to the
original MS framework, is that there is no need to (approximately) solve a
complicated nonlinear equation at every iteration that adds an extra logarithmic
factor to the total oracle complexity of the algorithm. As a result, the new
framework allows constructing optimal tensor methods of any order $p \geq 1$
(without any hidden logarithmic factors in the final complexity bound). In the
special case, that corresponds to the second-order method ($p = 2$), the authors
additionally propose a novel efficient procedure for solving the auxiliary
subproblem that arises at each iteration of the method. This procedure can be
seen as an alternative to the standard "line search" which is typically used for
an adaptive estimation of the Lipschitz (or Hölder) constant of the Hessian. In
the end, the authors obtain a new "universal" second-order method that requires
no knowledge of the problem parameters, and can be implemented using either an
exact linear system solver or an auxiliary conjugate gradient-type method.
Finally, the authors present several numerical experiments which confirm the
efficiency of their approach, and raise several interesting open questions.

**Questions:**

Major remarks:

1. Algorithm 2: Please clarify why the loops in lines 4 and 10 are guaranteed to
   terminate eventually.

1. Display between lines 895\-\-896: Please clarify the identity in the second
   line.

1. Lemma 3: 1) There is no need to assume that $r\_1, \\ldots, r\_k$ are integers.
   They can be any nonnegative real numbers. 2) You can assume that $\\beta \= 1$
   as it can always be "incorporated" into $r\_j$ (by defining $r\_j' \= \\beta
   r\_j$). 3) The proof can be considerably simplified. Indeed, denote $C\_i \=
   \\sum\_{j \= 1}^i B\_j r\_j$ for any $0 \\leq i \\leq k$ (with $C\_0 \= 0$). In this
   notation, our relation reads $B\_i^m \\geq C\_i$ (we assume that $\\beta \= 1$).
   Consequently, for any $0 \\leq i \\leq k$,
   $$
   C\_{i + 1} \- C\_i
   \\equiv
   B\_{i + 1} r\_{i + 1}
   \\geq
   C\_{i + 1}^{1 / m} r\_{i + 1},
   $$
   and hence
   $$
   r\_{i + 1}
   \\leq
   \\frac{C\_{i + 1} \- C\_i}{C\_{i + 1}^{1 / m}}
   \\leq
   \\int\_{C\_i}^{C\_{i + 1}} \\frac{d t}{t^{1 / m}}
   \=
   \\frac{m}{m \- 1} (C\_{i + 1}^{(m \- 1) / m} \- C\_i^{(m \- 1) / m})
   $$
   Summing up these inequalities and using $C\_0 \= 0$, we get, for any $1 \\leq i
   \\leq k$,
   $$
   C\_i^{(m \- 1) / m} \\geq \\frac{m \- 1}{m} \\sum\_{j \= 1}^i r\_j.
   $$
   Rearranging and using $B\_i \\geq C\_i^{1 / m}$, we obtain the claim.

Minor remarks:

1. Theorem 1: 1) Please clarify why the final convergence bound does not depend
   on $\\lambda\_0'$. 2) Is the formula for $C\_{\\alpha, s}$ correct? Why is it
   different from the one in lines 700\-\-701? 3) It would be better to add some
   remarks on the "optimal" value of $\\alpha$.

1. Line 278: "equivalent to". Not exactly, the constants are different.

1. How do you choose $\\lambda\_0'$ in the experiments (Section 4)?

1. Line 293: Perhaps you should divide by $6 \\sqrt{3}$ in the formula for the
   Lipschitz constant?

1. Proposition 1: 1) The letter $M$ is already used to denote the regularization
   coefficient in eq. (6), therefore it would be better to rename it. 2) Eq.
   (14): You can remove $A\_0 E\_0$ since $A\_0 \= 0$.

1. Lemma 1: Please clarify that the statement holds for **any** $T \\geq 1$ and
   that the index sets $Q\_T$ and the numbers $r\_{t, T}$ are consistent in the
   sense that $Q\_T \\subseteq Q\_{T + 1}$ and $r\_{t, T} \= r\_{t, T + 1}$ for all $T
   \\geq 1$ and all $t \\in Q\_T$. This is implicitly used later in Section C.3.

1. Proof of Lemma 1: It might be reasonable to write $n\_i$ instead of
   $\\operatorname{next}(\\tau\_i)$ to make the formulas more compact.

1. Line 668: Why $r\_1 \\geq \\frac{1}{2}$? Can it happen that
   $\\operatorname{next}(\\tau\_1) \= 1$ and then $r\_1 \= 0$?

1. Line 669: This is just the arithmetic\-geometric mean (AM\-GM) inequality.

1. Display between lines 670\-\-671: Please clarify what happens with the first
   term ($i \= 1$) in the second inequality. Should we define $r\_1 \=
   \\operatorname{next}(\\tau\_1) \- 1$ (without dividing by $2$)?

1. Section C.3: "The $s < \\infty$ case". It looks as if the case $s \= 1$ needs
   to be considered separately. Otherwise, it is not clear what
   $(\\lambda\_t')^{(s + 1) / (s \- 1)}$, $c^{2 s / (s \- 1)}$, etc. mean.

1. Please clarify that eqs. (27) and (28) hold for **any** $T \\geq 1$.

1. Line 688: Is it true that $r\_t \\geq 1$? According to line 668, we may only
   guarantee that $r\_t \\geq \\frac{1}{2}$.

1. Lemma 9: 1) One should add $\\lambda\_1 > 0$. 2) The claim in line 814 is not
   true in general as $0 \\preceq M\_1 \\preceq M\_2$ does not imply $M\_1^2 \\preceq
   M\_2^2$. Nevertheless, in this particular case, the implication does hold
   since the matrices $M\_1$ and $M\_2$ have the same orthonormal eigenbasis.

Typos:

1. Line 49: "term" $\\to$ "factor"?

1. Lines 51 and 242: "$\\mathcal{O}\_{\\mathrm{cr}}(1)$" $\\to$
   "$\\mathcal{O}\_{\\mathrm{cr}}$"?

1. Algorithm 1: 1) Line 9: Should be $A\_{t + 1} \= A\_{t + 1}'$. 2) Line 14:
   Should be $A\_{t + 1} \= A\_t + a\_{t + 1}$.

1. Algorithm 0: 1) Line 7: Should be $a\_{t + 1}'$ in the second fraction. 2)
   Line 10: Should be $A\_{t + 1} \= A\_{t + 1}'$.

1. Line 101: "$\\mathcal{X}$" $\\to$ "$\\mathbb{R}^d$"? The same applies to lines
   128, 135, ...

1. Line 118: "... very far **from** ...".

1. Footnote 1 (page 4): "... interval ... which always". Remove "which".

1. Line 156: There should be "$\\geq$" in the definition of $S\_T^{\\geq}$.

1. Line 216: $\\lambda$ should be in the right\-hand side, not the left\-hand side.

1. Algorithm 2, line 10: "$\\lambda\_{\\mathrm{vld}}$" $\\to$
   "$\\lambda\_{\\mathrm{invld}}$".

1. Lines 248, 272: What does $\\alpha \= (1.1, O(1))$ mean?

1. Eq. (16): There should be $v\_t \- x\_\*$ and $M\_{t + 1}$. The same applies to
   the display between lines 635\-\-636.

1. Line 630: There should be $\\lambda\_{t + 1}'$.

1. Eq. (17): There should be "$\\leq$" in the second line.

1. Display in lines 635\-\-636: Forgotten $[f(\\tilde{x}\_{t + 1}) \- f^\*]$ at the
   end of the first line.

1. Eq. (18): Should be $M\_{t + 1}$.

1. Line 660: Should be $k \\in (\\operatorname{next}(\\tau\_i), \\tau\_{i + 1}]$
   (including $\\tau\_{i + 1}$).

1. Line 664: 1) "we **have**". 2) Should be
   $\\lambda\_{\\operatorname{next}(\\tau\_i)}' \= \\ldots$ instead of "$\\leq$"?

1. Eq. (26): Rename "$i$" $\\to$ "$j$" in the second sum.

1. Display between lines 691\-\-692: 1) There should be $B\_T$ instead of $B\_t$ in
   the first identity. 2) "for all $t \\in Q\_T$" $\\to$ "for all $T \\geq 1$".

1. Lemma 10: There should be $H / (1 + \\nu)$ instead of $H / 2$ in the displays
   between lines 816\-\-817 and 818\-\-819.

1. Display between lines 895\-\-896: There should be "$\\log\_2$" everywhere.

1. Line 897: "holds **for** every".

1. Display between lines 900\-\-901: Should be $x\_0 \- x\_\*$.

**Strengths And Weaknesses:**

This is an excellent work that has at least two significant contributions.

First, the authors solve an important open problem of constructing an optimal
tensor method that does not have any extra logarithmic factors in its complexity
estimate. Their solution is quite elegant in that their method is essentially
the original MS algorithm with a couple of small (but nontrivial) modifications.
Specifically, instead of having an inner loop searching for an (approximate)
solution to the nonlinear equation, they simply make one step (of the basic
method) and then check if a certain condition is satisfied: if yes, they keep
the result and go to the next iteration; if no, they take the average of the
"unsuccessful" new point and the current one, and also go the next iteration. In
addition to that, they have a very simple adaptation procedure for the "guess"
of the solution to the nonlinear equation: it is either multiplied by a constant
if it happens to be too small, or divided by a constant if it happens to be too
large, but, in any case, the corresponding adjustment happens only once at each
iteration and does not require an auxiliary inner loop. Overall, the method is
very interesting and definitely requires further attention.

It is also important that the authors do not stop after obtaining a method that
is theoretically optimal but requires the knowledge of certain problem
parameters (such as the Lipschitz constant). Instead, they develop an adaptive
variant of their scheme for the special case of the second-order oracle.
Essentially, they propose a new variant of the "line search" procedure which is
quite different (and also simpler and more efficient) than the standard one that
is typically used in second-order methods. Personally, I find this new
"line search" very interesting and believe it could be of use in a broader
context.

In my opinion, the main part of the paper is clearly written. However, the
quality of the supplementary material is slightly worse.

Please see my detailed remarks in the next section.

---

> ### Author Response · Authors · 2022-08-02
> **Author reply**
>
> Thank you for the exceptionally detailed reading of our paper and positive remarks on its contribution. We will carefully address all of your careful and constructive feedback in our revision, which will greatly improve our paper. Below, we address your comments and questions in detail.
>
> **Major remarks**
>
> 1. The while loop in line 10 is guaranteed to terminate because, whenever the Hessian (or gradient) is continuous in a neighborhood of the the query point $y$, any $\lambda$ larger than some $\tilde{\lambda}<\infty$ is guaranteed to satisfy the MS condition. We argue this in lines 870-872.
>
>     The termination of the while loop in line 4 (when Lazy is False) is, strictly speaking, not guaranteed. For example, if the function $f$ is quadratic, MSCheck will always be True. However, it is also straightforward to verify that as long as MSCheck is True for a given $\lambda$, the corresponding regularized Newton step $x=y - (\nabla^2 f(y) + \lambda I)^{-1} \nabla f(y)$ has optimality gap bounded by $\lambda \Vert x-y\Vert ^2$. Therefore, since the loop in line 4 decreases $\lambda$ at a double-exponential rate, if it fails to terminate after a small number of iterations then it means we have found an essentially optimal point. Put differently, if we seek an $\epsilon$ suboptimal point, we may stop the loop in line 4 after $O(\log \log( \frac{\lambda_0’ R^2}{\epsilon}))$ iterations. This possibility is accounted for by the complexity bound in Corollary 3.
>
>     We agree that these termination issues are not sufficiently clear in our current writeup, and will revise it to clarify them.
>
> 2. The equality (rather than inequality) symbol is just notational inconsistency which we will fix in the revision. The bound holds because, for any $z>1$ and constant $c$ $$\log (1+\log z) \le \log(z) = c \log (z^{1/c}) \le O(z^{1/c}).$$
>
> 3. Thank you for the great observations and suggestions regarding Lemma 3! We agree with all of them and will incorporate these improvements and simplifications in our revision.
>
> **Minor remarks**
>
> 1. We answer each sub-question in turn:
>      1) The convergence rate in Theorem 1 does not depend on $\lambda_0’$ because of the way we set $\lambda_1 = \lambda’_1$ in line 2, and because we assume a movement holds for iteration $t=1$.
>
>     2) We use $\ln(\alpha) = \Omega(\min\{1, \alpha-1\})$ and $s+1/(3s+1) \in [1/2,1/3]$ to simplify the expression. We will update the proof to include this additional step.
>
>     3)  Thank you for the suggestion - we will include such discussion in the revision. See also Figure 6 (left column) for an empirical study of this question, showing that $\alpha=2$ is a reasonable choice.
>
> 2. Good catch - we will clarify this point.
>
> 3. We keep $\lambda_0’=0.1$ throughout (see line 1015) - the parameter has little to no effect in practice.
>
> 4. Note that $1/6\sqrt{3} \approx 0.1$, so that taking $M=0.2\bar{H}$ approximately corresponds to taking 2 times the upper bound on the Lipschitz constant, which is what the theory calls for in most methods. We will clarify this in the revision. See also Figure 7 for experiments on the effect of varying $M$.
>
> 5. We agree with both of your comments - we will rename $M$ and remove the $A_0$ term in the revision.
>
> 6. In the revision, we will use a different letter for $T$ here, and show this consistency argument which is indeed important for C.3. Note that $r_{t,T}=r_{t,T+1}$ may not hold in general, but it does hold in the sense that $r_{t, T_1} = r_{t,T_2}$ if $t< T_1< T_2$, $t, T_1,T_2\in\mathcal{Q}_T$, which suffices for using the recursive argument for C.3.
>
> 7. Thanks for the suggestion! We will make that change in the revision.
>
> 8. Yes, indeed it can happen. This claim that $r_t\ge1/2$ for all $ti\in\mathcal{Q}_T$ is  wrong and we will instead only state $r_t\ge0$ for all $t\in\mathcal{Q}_T$ (which will not affect the correctness of later derivations due to the point 13 below).
>
> 9. Correct - we will use this simpler explanation in the revision.
>
> 10. The first term comes purely from the first term (when $i=1$) in the second summation term by using $next(\tau_1)-\tau_1\ge \frac{1}{2}*[next(\tau_1)-\tau_1]$ - this is why we need a coefficient of ¼  instead of ½ in front in the lower bound obtained.
>
> 11. Good point - we will treat this case more carefully in the revision. Note however that s=1 essentially corresponds to the analysis of Nesterov acceleration.
>
> 12. We will clarify this in the revision.
>
> 13. You are correct. For this reason, we need to prove Lemmas 3 and 4 for the case when r_i are non-negative, and sum_i r_i >0. Our original proofs of both lemmas work for this more general case and so does the simpler proof you are suggesting (instead of the original Lemma 3).
>
> 14. This is a great catch! We have also realized this after submitting the paper and we will correct this point in the revision.
>
> **Typos**: thank you so much for spotting all of these typos - we will correct them carefully.

---

> > ### Comment · Reviewer_dAbm · 2022-08-09
> > **After rebuttal**
> >
> > Thanks for your response. Please make sure to include all these comments into
> > the revised version of the paper.
> >
> > I just have one additional comment regarding Major remark 2 and your answer to
> > it. If I did not miss anything, there should be the binary logarithm $\\log\_2$
> > everywhere instead of the natural logarithm $\\ln$. Therefore, the inequality
> > should be $\\log\_2(1 + t) \\leq t / \\ln 2$ (instead of $\\log\_2(1 + t) \\leq
> > t$). But most probably this additional constant factor should not be a problem.
> > In any case, what I find a bit puzzling is that, in the end, you apply the bound
> > $\\log t \\leq t$. Isn't it too crude? During my original reading, I had an
> > impression that lines 895--896 are exactly the reason why it is important to
> > have the "double-logarithmic scale" in Algorithm 2. Is it true, or is there any
> > other reason? I would appreciate if you could clarify it.

---

> > > ### Author Response · Authors · 2022-08-09
> > > **re: After rebuttal**
> > >
> > > Since the base of the logarithm only changes constant factors, we omit it when using big-O notation. You are correct that binary logarithms naturally show up in the analysis of Algorithms 2 and 3 - we will clarify this in the revision.
> > >
> > > It is true that in lines 895-896 we just “throw away” the outer logarithmic term; a normal bisection or doubling scheme would also give the same bound, which is precisely what happens in the proof of Corollary 5 (lines 991-992). The utility of using a “double-logarithmic scale” in Algorithm 2 manifests in the additive double-logarithmic term in Corollary 3, which stems from cases (iii) and (iv) in the proof. A normal doubling/bisection scheme would have given us an additive logarithmic term instead. This is admittedly only a small difference, but still one worth noting in our opinion.
> > >
> > > Thank you again for all your valuable feedback, which we will make sure to thoroughly address in the revision.

---

### Official Review · Reviewer_RLWS · 2022-07-17

**Rating:** 5
**Confidence:** 4
**Soundness:** 3 good
**Presentation:** 2 fair
**Contribution:** 2 fair

**Summary:**

The paper introduces a new variant of Monteiro-Svaiter (MS) framework which serves as a meta-algorithm to accelerate (inexact) backward optimization method in convex optimization, like proximal point algorithm, cubic regularization, etc. The proposed variant removes the bisection step in the original MS method, improving the theoretical complexity by a logarithmic term so that the obtained bound matches the optimal bounds up to constant factors. The proposed method does not require any parameter tuning in a way that the regularization parameter $\lambda$ is adaptively adjusted, without requiring any knowledge of the Lipschitz/Hölder constant and order in advance. The paper shows that the algorithm achieves optimal complexity bound and provide a detailed discussion on the use case of cubic regularization. Experiments are conducted to validate the theoretical guarantees.

**Questions:**

The paper provides detailed discussion on how to improve MS framework with the cubic regularization method and further abstract it to a more generalized framework. However, some steps in the abstraction is not very clear to me.

In Definition 1/Algorithm 1, the paper introduces a novel oracle called MS oracle which takes input of a couple of iterate and regularization parameter ($y, \lambda'$)  and output another couple $(x, \lambda)$ such that
$$ \Vert x - (y - \frac{1}{\lambda} \nabla f(x)) \Vert \le \sigma \Vert x- y \Vert $$
However, what is a bit intriguing to me is that the oracle does not seem to depend on the $\lambda' $. Conceptually, I understand that we want to perform some line search type method from $\lambda'$ to get $\lambda$. However, without any restriction on $\lambda$, the oracle seems meaningless as the obtained $\lambda$ could be arbitrary. More precisely, take the example of proximal point algorithm, for any $\lambda$, when we solve the subproblem
$$F(z) := f(z) + \frac{\lambda}{2} \Vert z - y \Vert^2 $$
to sufficient accuracy, the solution will satisfies the MS oracle. In other words, take a very small $\lambda = 10^{-10} \lambda'$  and solve the above $F$ up to very high accuracy is essentially solving the original problem. So I think an important step in Algorithm 1 should be discussing how to select $\lambda$ with respect to $\lambda'$. Otherwise the complexity bound could be vacuous or non-meaningful.

Another question I have is how the algorithm achieves no parameter tuning. Upon my understanding (correct me if I am wrong), the parameters $s$ and $c$ in the ($s$, $c$) movement bound serves as some sort of approximate Holder order. However, I am not able to understand how the algorithm adaptively select them, it seems like they are kept constant along the way.

Minor questions:
In equation right below line 191, it is claimed that the MS oracle is satisfied for Taylor descent step. It seems to me this only hold  for $p+\nu =2 $, otherwise there would be a power in the (x-y) term.

Literature:
I believe the following literature on accelerated proximal point algorithms are related (as special case $\lambda' =\lambda$)
1.  Inexact and accelerated proximal point algorithms.  Salzo and Villa
2. Catalyst acceleration for first-order convex optimization: from theory to practice. Lin et al

**Limitations:**

The contribution of the paper is mostly theoretical where might have limited practical impact

**Strengths And Weaknesses:**

Strength: Theoretical improvement on existing MS framework
Weakness: very hard to follow, the presentation need to be improved. More clarification on several key technical component is required.

---

> ### Author Response · Authors · 2022-08-02
> **Author reply**
>
> Thank you for your comments, which we address in detail below - we would be happy to answer any additional questions. We hope that you will re-evaluate the contribution and strength of our paper based on our responses.
>
> > the presentation need to be improved
>
> While Monteiro-Svaiter acceleration is highly technical, it is a powerful optimization technique that we hope our work will make more accessible. We are keen to improve the readability of our paper and would appreciate concrete suggestions.
>
> > Algorithm 1 should be discussing how to select $\lambda$ with respect to $\lambda′$. Otherwise the complexity bound could be vacuous or non-meaningful
>
> Algorithm 1 does indeed work with *any* oracle satisfying Definition 1. While this includes exact proximal oracles with very small $\lambda$ (as you point out), implementing such oracles would indeed require many gradient / Hessian evaluations. However, there exist other, cheap-to-implement oracles that also satisfy Definition 1 - we account for the implementation cost of various oracles in Section 3. We obtain different complexity bounds by then multiplying the iteration count bound from Theorem 1 with different oracle implementation complexities - please see Table 1 and Corollaries 3 and 5.
>
> To further clarify this point, it might help to note that for many efficient oracles the value of $\lambda$ is not chosen prior to the oracle call, but is rather *implicitly* determined from the oracle output. For example, for the cubic-regularized Newton step in Eq. (1), if $y$ is the oracle query and $x$ is its output, then $\lambda = 0.5 M \Vert x-y\Vert $. While lines 19-37 in the introduction, Definition 1, and Section 3.1 were written in an attempt to highlight the implicit nature of $\lambda$, we welcome suggestions for improvements.
>
> Finally, we note that oracles with implicitly-determined $\lambda$ values are not new to our work - they appear in all prior work on MS acceleration (see Algorithm 0). The novel aspect of Algorithm 1 is the more efficient use of these oracles, which decreases the number of times they need to be invoked by a logarithmic factor.
>
>
> > Another question I have is how the algorithm achieves no parameter tuning. Upon my understanding (correct me if I am wrong), the parameters $s$ and $c$ in the $(s, c)$ movement bound serves as some sort of approximate Holder order. However, I am not able to understand how the algorithm adaptively select them, it seems like they are kept constant along the way.
>
> The movement bound parameters depend on a combination of the oracle structure (e.g., gradient step vs. Newton step) and the objective function structure (e.g., Lipschitz continuity of the Hessian) - please see Section 3.1 for examples. In practice, the movement bound parameters are not known in advance. Algorithm 1 is *agnostic* to these parameters. *Adaptivity* is achieved via carefully designing new oracles - see Algorithm 2 and 3 in Sections 3.2 and 3.3, respectively.
>
> > Minor questions: In equation right below line 191, it is claimed that the MS oracle is satisfied for Taylor descent step. It seems to me this only hold for $p+\nu=2$, otherwise there would be a power in the (x-y) term.
>
> The claim holds for any $p+\nu \ge 2$ - please see lines 193-195 for explanation of how we get from the RHS of the equation below line 191 to the term $\sigma \Vert x-y\Vert $ in the MS condition in Definition 1.
>
> > Literature: I believe the following literature on accelerated proximal point algorithms are related
>
> Thank you for these relevant references. We already cite Salzo and Villa (see reference [33]), and we will add a citation of Lin et al as well.
>
> > The contribution of the paper is mostly theoretical where might have limited practical impact
>
> We agree that the primary contribution of the paper is to solve an enduring theoretical open problem in convex optimization. However, especially compared to prior work on Monteiro-Svaiter acceleration, we are also fairly optimistic about the practical impact of our work: our experiments indicate that our algorithms outperform prior theoretically-certified accelerated methods, without any parameter tuning. While Newton’s method appears to be empirically superior for logistic regression, we believe that (a) this finding is important by itself and warrants publication, and (b) our development still constitutes progress toward algorithms that are both more efficient and theoretically sound. Please see Section 1.2 for additional discussion of the limitations of our work.

---

### Meta-Review · Area_Chair_i2GG · 2022-08-23

**Recommendation:** Accept
**Confidence:** Certain

**Metareview:**

The paper proposes a variant of MS acceleration that requires no bisection. This in turn can be used to accelerate the cubic regularization method and other proximal based methods, matching previously established lower bounds. The specialized second order variant requires no knowledge of the lipschitz constant of the Hessian by using what can be considered as a new type of line-search. Furthermore, the level of writing and contributions was enough to motivate the reviewers to examine the paper in depth, including the supplementary material. Finally, the expert reviewers were very impressed with contributions of the paper, and believe this will have repercussions outside of the immediate targeted applications.

**Award:**

Yes

---

### Decision · Program_Chairs · 2022-09-14

Accept